**Cite this article:** van der Bles AM, van der Linden S, Freeman ALJ, Mitchell J, Galvao AB, Zaval L, Spiegelhalter DJ. 2019 Communicating uncertainty about facts, numbers and science. *R. Soc. open sci.* **6**: 181870.

psychology/statistics

uncertainty communication, epistemic uncertainty, economic statistics, IPCC, grade

**Author for correspondence:**
Anne Marthe van der Bles
e-mail: amv46@cam.ac.uk

# Communicating uncertainty about facts, numbers and science

Anne Marthe van der Bles[1,2], Sander van der Linden[1,2], Alexandra L. J. Freeman[1], James Mitchell[3], Ana B. Galvao[3], Lisa Zaval[4] and David J. Spiegelhalter[1]

[1]Winton Centre for Risk and Evidence Communication, Department of Pure Mathematics and Mathematical Statistics, and [2]Cambridge Social Decision-Making Lab, Department of Psychology, University of Cambridge, Cambridge, UK
[3]Warwick Business School, University of Warwick, Coventry, UK
[4]Department of Psychology, Columbia University, New York, NY, USA

 AMvdB, 0000-0002-7953-9425; SvdL, 0000-0002-0269-1744; ALJF, 0000-0002-4115-161X; JM, 0000-0003-0532-4568; ABG, 0000-0003-3263-9450; DJS, 0000-0001-9350-6745

Uncertainty is an inherent part of knowledge, and yet in an era of contested expertise, many shy away from openly communicating their uncertainty about what they know, fearful of their audience's reaction. But what effect does communication of such epistemic uncertainty have? Empirical research is widely scattered across many disciplines. This interdisciplinary review structures and summarizes current practice and research across domains, combining a statistical and psychological perspective. This informs a framework for uncertainty communication in which we identify three objects of uncertainty—facts, numbers and science—and two levels of uncertainty: direct and indirect. An examination of current practices provides a scale of nine expressions of direct uncertainty. We discuss attempts to codify indirect uncertainty in terms of quality of the underlying evidence. We review the limited literature about the effects of communicating epistemic uncertainty on cognition, affect, trust and decision-making. While there is some evidence that communicating epistemic uncertainty does not necessarily affect audiences negatively, impact can vary between individuals and communication formats. Case studies in economic statistics and climate change illustrate our framework in action. We conclude with advice to guide both communicators and future researchers in this important but so far rather neglected field.

# 1. Communicating uncertainty about facts, numbers and science

Uncertainty: a situation in which something is not known, or something that is not known or certain (Cambridge Dictionary) [1]

Uncertainty is all-pervasive in the world, and we regularly communicate this in everyday life. We might say we are uncertain when we are unable to predict the future, we cannot decide what to do, there is ambiguity about what something means, we are ignorant of what has happened or simply for a general feeling of doubt or unease. The broad definition above from the Cambridge dictionary reflects these myriad ways the term 'uncertainty' is used in normal speech.

In the scientific context, a large literature has focused on what is frequently termed 'aleatory uncertainty' due to the fundamental indeterminacy or randomness in the world, often couched in terms of luck or chance. This generally relates to future events, which we *can't know* for certain. This form of uncertainty is an essential part of the assessment, communication and management of both quantifiable and unquantifiable future risks, and prominent examples include uncertain economic forecasts, climate change models and actuarial survival curves.

By contrast, our focus in this paper is uncertainties about facts, numbers and science due to limited knowledge or ignorance—so-called epistemic uncertainty. Epistemic uncertainty generally, but not always, concerns past or present phenomena that we currently *don't know* but could, at least in theory, know or establish.[1] Such epistemic uncertainty is an integral part of every stage of the scientific process: from the assumptions we have, the observations we note, to the extrapolations and the generalizations that we make. This means that all knowledge on which decisions and policies are based—from medical evidence to government statistics—is shrouded with epistemic uncertainty of different types and degrees.

Risk assessment and communication about possible future events are well-established academic and professional disciplines. Apart from the pure aleatory uncertainty of, say, roulette, the assessment of future risks generally also contains a strong element of epistemic uncertainty, in that further knowledge would revise our predictions: see the later example of climate change. However, there has been comparatively little study of communicating 'pure' epistemic uncertainty, even though failure to do so clearly can seriously compromise decisions (see box 1).

Recent claims that we are living in a 'post-truth' society [7] do not seem encouraging for scientists and policy makers to feel able to communicate their uncertainty openly. Surveys suggest declining levels of trust in governments and institutions [8–10], although trust in scientists apparently remains high in both the UK and USA [11,12]. Anecdotal experience suggests a tacit assumption among many scientists and policy makers that communicating uncertainty might have negative consequences, such as signalling incompetence, encouraging critics and decreasing trust (e.g. [13]). By contrast, an alternative view as proposed, for example, by the philosopher O'Neill [14] is that such transparency might build rather than undermine trust in authorities.

In order to know which of these conflicting claims hold, empirical evidence on the effects of communicating uncertainty about facts, numbers and science needs to be collected and reviewed. This process faces two major challenges. First, the existing empirical research on the effects of communicating epistemic uncertainty is limited. Second, 'communicating epistemic uncertainty' can mean many different things. It can be a graph of a probability distribution of the historic global temperature change, a range around an estimate of the number of tigers in India, or a statement about the uncertainty arising from poor-quality evidence, such as a contaminated DNA test in a criminal court. All these variations may influence how the communication of uncertainty affects people.

In this paper, we present a cohesive framework that aims to provide clarity and structure to the issues surrounding such communication. It combines a statistical approach to quantifying uncertainty with a psychological perspective that stresses the importance of the effects of communication on the audience, and is informed by both a review of empirical studies on these effects and examples of real-world uncertainty communication from a range of fields. Our aim is to provide guidance on how best to communicate uncertainty honestly and transparently without losing trust and credibility, to the benefit of everyone who subsequently uses the information to form an opinion or make a decision.

[1]We may, for example, have epistemic uncertainty about future events that have no randomness attached to them but that we currently do not know (for example, presents that we might receive on our birthday that have already been bought: there is no aleatory uncertainty, only uncertainty caused by our lack of information, which will be updated when our birthday arrives). In this paper, we do not consider concepts that are not even theoretically knowable, such as non-identifiable parameters in statistical models, knowledge about counterfactual events or the existence of God. We refer the reader to Manski [2] for a discussion of 'nonrefutable' and 'refutable' (or testable) assumptions in econometrics.

> **Box 1.** The importance of uncertainty communication: the tale of the 'dodgy dossier'.
>
> On 24 September 2002, the British government published a document entitled 'Iraq's Weapons of Mass Destruction: The Assessment of the British Government' [3]. It included claims about Iraq having programmes to develop weapons of mass destruction and nuclear ambitions, and provided a 'case for war'. After the 2003 invasion of Iraq, however, the Iraq Survey Group found no active weapons of mass destruction and no efforts to restart a nuclear programme.
>
> Given these obvious gaps between the document and subsequent findings in reality, an independent investigation (the Butler Review) was set up in 2004. The Butler Review concluded that although there was no deliberate distortion in the report, expressions of uncertainty in the intelligence, present in the original non-public assessments, were removed or not made clear enough in the public report.
>
> > 'We believe that it was a serious weakness that the JIC's warnings on the limitations of the intelligence underlying some of its judgements were not made sufficiently clear in the dossier'. [4, p. 82 and p. 114]
>
> In the USA, it was the Intelligence Community's October 2002 National Intelligence Estimate (NIE) called 'Iraq's Continuing Programs for Weapons of Mass Destruction' [5] that was the analogous document pre-invasion. A US Senate Select Committee investigation was even more critical of it than the Butler Review was in the UK, but its second conclusion was similar:
>
> > 'Conclusion 2. The Intelligence Community did not accurately or adequately explain to policymakers the uncertainties behind the judgments in the October 2002 National Intelligence Estimate'. [6, p. 16]
>
> The removal of considerable expressions of uncertainty from both documents had a dramatic effect on the opinions of the public and governments, and in the UK at least the removal of the uncertainties was considered key to paving the way to war.

## 1.1. A framework for communicating epistemic uncertainty

In contrast to the numerous attempts at generic taxonomies of uncertainty, the framework proposed in this paper is specifically geared to the task of communication: a comparison with other proposals is made in the next section. Based on Lasswell's venerable model of communication [15], our framework addresses who communicates what, in what form, to whom and to what effect while acknowledging the relevant context as part of the characteristics of the audience. This framework for uncertainty communication is displayed in figure 1.

The first two factors in our framework relate to *who* is communicating (briefly covered in §2):

— the *people assessing the uncertainty*, who will generally be 'experts' of some kind, such as individual scientists, scientific groups such as the Intergovernmental Panel on Climate Change (IPCC), or official bodies such as national statistical organizations. These are essentially the 'owners' of the uncertainty.
— the *people doing the communication*, who may include technical experts, communication professionals and journalists, often acting on behalf of institutions.

Factors related to *what* is being communicated are (§3):

— the *object* about which there is uncertainty, in terms of facts, numbers or scientific models and hypotheses
— the *source* of the uncertainty, as in the reasons for the lack of knowledge
— the *level* of the uncertainty communicated: from direct uncertainty about a fact, to the indirect uncertainty or lack of confidence in the underlying science
— the *magnitude* of the uncertainty, from a small lack of precision to a substantial degree of ignorance.

Factors relating to the *form* of the communication (§4):

— the *expression of the uncertainty*, such as a full probability distribution or just a brief mention that uncertainty exists
— the *format* of the uncertainty communication, in terms of numbers, visualizations or verbal statements
— the *medium* of the communication, such as print, online, broadcast or verbal conversation.

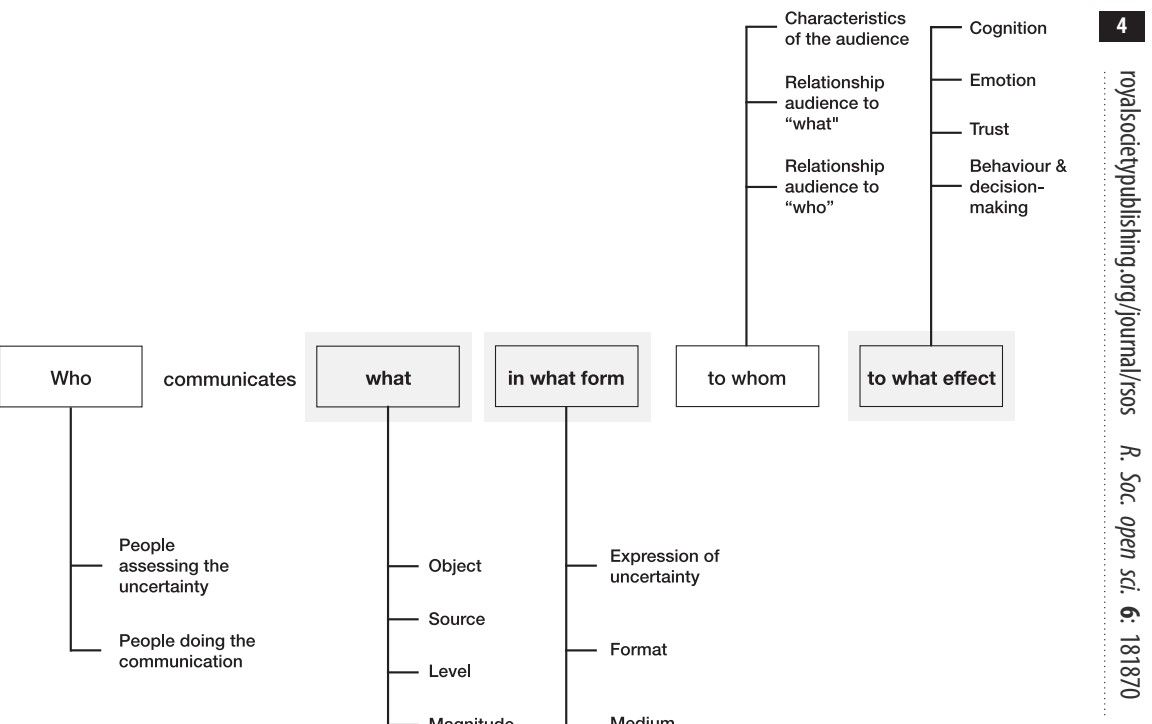

**Figure 1.** Basic deconstruction of the communication of epistemic uncertainty based on the Lasswell model of communication [15]. Our emphases in this paper—*what*, *in what form* and *to what effect*, are indicated in bold.

Factors relating to *whom* is being communicated to (briefly covered in §5):

— the *characteristics of the audiences*, for example, in terms of their varying levels of numeracy and (graphical) literacy, their expertise and knowledge of the field
— the *relationship of the audience to what is being communicated*, such as whether the topic is contested or emotionally laden for them
— the *relationship of the audience to the people doing the communication*, including perceived credibility and whether there is trust or distrust between audience and communicators.

Finally, factors relating to *what effect* the communication has on the audience (§6):

— the effect of communication on the audience's *cognition, emotion, trust*, and *behaviour and decision-making*.

The first three sections of this paper follow the list above, briefly describing the *who* before concentrating on the *what* and the *form* of the communication. We illustrate current practice in uncertainty communication in a variety of domains including forensics, environmental health risks, public health, conservation biology, history and military intelligence. In the last two sections, we review the current, rather limited, academic literature evaluating the psychological effect of uncertainty communication—including visual, verbal and numerical formats—and what is known about the moderating effects of audience characteristics. The focus of this paper is on clarifying and structuring *what* is being communicated and in *what form*, and reviewing what we know about its *effects*. Only brief comments are provided about the *who* and *to whom* components.

Next, two case studies are presented: one in the field of climate change and one in the field of official economic statistics. These serve to illustrate how our framework of approaching uncertainty communication might be used to analyse current real-world graphics and messages, and inform future research and development of more evidence-based communications. The final discussion summarizes our contribution and provides key points for both communicators and researchers of communication.

A worthy eventual goal would be empirically based guidance for a communicator on the likely forms, levels and prominence of uncertainty communication that would suit their audience and aims. This study is intended to make a start towards that aim and we summarize our conclusions (so far) for communicators in box 5.

## 1.2. Other frameworks for uncertainty

Many taxonomies of uncertainty have been made in a range of disciplines, often being concerned with 'deeper' uncertainties inherent in any formal models that have been constructed as ways of representing our scientific understanding of the world around us. For example, in the context of integrated assessment models for climate change, Walker *et al*. [16] separated uncertainty about the context, the structure of the model itself, the outcomes considered and the weights or values being assigned to outcomes, while van Asselt & Rotmans [17] deconstruct 'source' to list five sources of uncertainty due to variability and seven sources of uncertainty due to limited knowledge. Morgan *et al*. [18] emphasize numerical expression of uncertainty, including placing probabilities on alternative models, while in contrast Kandlikar *et al*. [19] proposed a qualitative scale of confidence in the underlying science, based on the degree of expert agreement and quality of underlying evidence (this corresponds to our 'indirect' level of uncertainty, as outlined in §3.3: see also the Case Study 2 on climate change before the Discussion).

Within medicine, Han [20] characterizes uncertainty in clinical decision-making in terms of probability of future uncertain outcomes, ambiguity about what those probabilities are and complexity of the problem. In a general scientific context, Wynne [21] considers 'indeterminacy' to mean the uncertainty about what scientific knowledge fits the current situation, and 'ignorance' as when we don't know what we don't know about the completeness and validity of our knowledge, which by definition escapes recognition. Under the generic banner of 'incertitude', Stirling [22] uses the term *ambiguity* for when there is doubt about outcomes, and *ignorance* when both probabilities and outcomes cannot be confidently specified. Funtowicz & Ravetz's [23] NUSAP scheme for reporting numbers emphasizes the 'pedigree' (the P in NUSAP), again corresponding to our 'indirect' level of uncertainty, reflecting the quality of the underlying evidence.

In spite of all this activity, no consensus has emerged as to a general framework, perhaps due to the wide variety of contexts and tasks being considered, and the complexity of many of the proposals. Our structure, with its more restricted aim of communicating epistemic uncertainty, attempts to be a pragmatic cross-disciplinary compromise between applicability and generality. The individual elements of it are those factors which we believe (either through direct empirical evidence or suggestive evidence from other fields) could affect the communication of uncertainty and thus should be considered individually.

# 2. Who is communicating?

Following the structure given in figure 1, we note briefly the importance of identifying *who* is communicating uncertainty. The people assessing and communicating uncertainty are many and varied, from specialists assessing evidence to communication officers or the media. They might be the same people doing both, or might be different people intimately involved—or not—in each other's task. Communicators may intend to have very different effects on their audiences, from strategically deployed uncertainty (also known as 'merchants of doubt') to transparent informativeness. For example, in the report on the document 'Iraq's Weapons of Mass Destruction: The Assessment of the British Government' [3] discussed in box 1 it was noted that the differences in uncertainty communication were in part because: 'The Government wanted a document on which it could draw in its advocacy of its policy. The JIC sought to offer a dispassionate assessment of intelligence and other material…' ([4] para 327).

As will be commented on further in the *to whom* section, assessors and communicators of uncertainty might have an existing relationship with the audience they are communicating to, which might be characterized by trust or distrust. A review of the literature on source credibility falls outside the scope of this paper, but we do want to raise the point of considering who is assessing and communicating uncertainty, their goals for communication and their relationship with the audience. These factors influence the choice of communication form and the effects of communication.

# 3. What is being communicated?

## 3.1. The object of uncertainty

Perhaps the first crucial question is: what are we uncertain about? Our specific focus is on residual epistemic uncertainty following scientific analysis, which will generally mean constructing a *model* for

---

**Box 2.** When we admit we do not know all the possibilities.

Donald Rumsfeld's famous discourse on the importance of 'unknown unknowns' highlighted the need to consider possibilities that cannot be currently identified [24]. While usually used as a motivation for developing resilient strategies for dealing with unforeseen future events, sometimes termed 'black swans', the idea can also apply to epistemic uncertainty about possible explanations or facts when it takes the form of a 'none of the above' category, meaning an eventuality that cannot currently be given a label. Examples might include a perpetrator of a crime who is not on the list of suspects, or a scientific mechanism that has not yet been formulated. It will generally be challenging to place a probability on this 'other' category.

 The humility to admit the possibility of being wrong is sometimes known as Cromwell's Law, after Oliver Cromwell's celebrated plea in the face of the Church of Scotland's obstinacy: 'I beseech you, in the bowels of Christ, think it possible you may be mistaken' [25, p. 18].

---

whatever is being studied, in the sense of a formal representation of available knowledge that contains certain assumptions about the values of potential variables, the process by which they are observed, and the way in which they interact.

 As previously emphasized, in contrast to the existing encompassing taxonomies our more restricted focus is on communicating epistemic uncertainty about facts, quantities and scientific hypotheses.

1. *Facts:* These can be formally considered as *categorical variables* that are (at least theoretically) directly verifiable, for example, whether or not the midsummer arctic ice-sheet has reduced in size over the last decade, or whether the number of homicides has increased in the last year; or one of a number of possibilities, such as who committed a particular crime. It is important that one category might be 'none of the above' (see box 2).

2. *Numbers:* These are *continuous variables* that describe the world. They may, at least in principle, be directly observable, or they may be theoretical constructs which are used as parameters within a model of the world. Examples of the former are the number of tigers in India, the current proportion of unemployed, or the growth in Gross Domestic Product (GDP) in the UK last year. Objects such as these which are being quantified always need to be carefully defined. This is clear when the object is an artificial construct such as GDP, but the definition of 'unemployed' also rests on changing convention, and even a 'tiger' needs unambiguous definition.
   Other quantities may be parameters of scientific models that cannot be directly observed but are only estimated within a scientific modelling framework, such as the size of risks associated with carcinogens, the average treatment effect of a drug, or the percentage of anthropogenic influence on global temperature over the last century—such parameters are often denoted by Greek letters such as $\theta$.

3. *Scientific hypotheses:* These are theories about how the world works, expressed as structural models of the relationship between variables, such as whether a particular exposure is carcinogenic, or the form of the dose–response relationship between ionizing radiation and harm. We will generally be uncertain about the most appropriate assumptions in a mathematical representation of the world. Remembering statistician George Box's adage that 'all models are wrong', but some are 'useful' [26, p. 792], we should in principle distinguish between the uncertainty about the adequacy of a model to represent the world (Does my map include all existing islands?), and uncertainty about the world itself (Does this island actually exist?). However, in practice, the lines between these often get blurred: the Higgs Boson cannot be directly observed, and so its existence is inferred as a component of a model that may, in future, be superseded. Scientific models and hypotheses are, like parameters, not directly observable 'things', but working assumptions.

 To illustrate these different objects of uncertainty, suppose you are asked to flip a coin – you flip it and cover it up immediately without seeing it. You now need to communicate your uncertainty about what the coin shows. In an idealized world, the answer is straightforward: your uncertainty about the *fact* of whether the coin shows heads (Object 1) is expressed by your probability[2] of $\frac{1}{2}$. This is a classic example of communicating uncertainty through the mathematical language of probability.

---

[2]Note that this is a probability in the Bayesian sense, expressing personal epistemic uncertainty rather than randomness.

But the real world can be more complicated, and not so readily quantifiable. Even fair coins may not be exactly balanced, and so there is inevitably a small element of uncertainty around the *number* $\frac{1}{2}$ (Object 2). This should be negligible provided the coin was flipped and not spun on its edge—a spun US penny coin is reported to land heads-up only around 20% of the time [27]. But additional knowledge might alter this probability: for example, if you know that the coin was heads-up before it was flipped, this changes the probability that it lands heads-up to around 51%.

Further, if you suspect the person who gave you the coin was a trickster, then the coin might even be two-headed and the probability of a head becomes one. So your confidence in the scientific *model* for the coin (Object 3) is vital, and this will depend on the evidence available about the situation—something not readily reduced to a numerical expression.[3]

## 3.2. Sources of uncertainty

A wide range of reasons for scientific uncertainty can be identified, including:

(1)  variability within a sampled population or repeated measures leading to, for example, statistical margins-of-error
(2)  computational or systematic inadequacies of measurement
(3)  limited knowledge and ignorance about underlying processes, and
(4)  expert disagreement.

The source may affect the response to uncertainty; it is an empirically researchable question whether, for example, difficulty in measurement versus expert disagreement as sources of uncertainty have different effects on an audience.

Different sources of uncertainty can lead to different forms of communication. For example, when assessing the number of migrants to a country in a preceding year, the impact of sampling variation due to survey design may be quantifiable and therefore communicated as a confidence interval. And in econometrics, partial identification is able to use the available (perhaps incomplete) data to communicate bounds around statistics or parameters of interest, by considering a weaker set of assumptions than required for *point* identification [2,28]. However, the uncertainty due to non-representative samples or inaccurate responses may be more difficult to quantify than the sampling variation (and yet possibly be of a greater magnitude) and so may need to be expressed in a different way.

## 3.3. The level of uncertainty

A vital consideration in communication is what we have termed the level of uncertainty: whether the uncertainty is directly about the object, or a form of indirect 'meta-uncertainty'—how sure we are about the underlying evidence upon which our assessments are based. This differs from the common distinction made between situations where probabilities are, or are not, assumed known. In the context of uncertainty quantification, the former is known as first-order uncertainty and the latter second-order uncertainty, often expressed as a probability distribution over first-order probability distributions or alternative models. An alternative categorization derives from Knight [29] and Keynes [30], who distinguish quantifiable risks from deeper (unquantifiable) uncertainties.

In contrast to both these approaches, we have observed that the major division in practical examples of communication comes between statements about uncertainty around the object of interest, which may or may not comprise precise first-order probabilities, and a 'meta-level' reflection on the adequacy of evidence upon which to make any judgement whatever. We therefore consider that, when communicating, it is most appropriate to distinguish two fundamental levels of uncertainty:

*Direct* uncertainty about the fact, number or scientific hypothesis. This can be communicated either in absolute quantitative terms, say a probability distribution or confidence interval, or expressed relative to alternatives, such as likelihood ratios, or given an approximate quantitative form, verbal summary and so on.
*Indirect* uncertainty in terms of the quality of the underlying knowledge that forms a basis for any claims about the fact, number or hypothesis. This will generally be communicated as a list of caveats about the underlying sources of evidence, possibly amalgamated into a qualitative or ordered categorical scale.

---

[3]However, Bayesian researchers perform 'Bayesian model averaging' which places subjective probabilities on the correctness of alternative, candidate scientific models; see the Technical appendix for further discussion.

**Box 3.** The expression of levels of uncertainty in legal reasoning.

Consider an archetypal criminal legal case in which the impact of a specific item of evidence on the possible guilt of a suspect is being considered.

*Direct uncertainty* concerns the *absolute* probability of guilt, and the *relative* 'probative value' given to an item of evidence for or against guilt of this particular suspect.

*Indirect uncertainty* would be reflected in the credibility to be given to an individual's testimony concerning this item of evidence.

In this context, these uncertainties are usually communicated in verbal terms: for example, direct absolute uncertainty may be expressed as 'beyond reasonable doubt', direct relative uncertainty may be communicated by saying some forensic evidence 'supports' or 'is consistent with' the guilt of the accused, while the indirect quality of the background knowledge might be introduced in cross-examination by querying the competence of the forensic expert or their access to appropriate data.

These ideas can be given a formal mathematical expression that may help understanding. Let $G$ and $I$ represent the uncertain facts of the guilt or innocence of the accused, and $d$ represent the specific item of forensic evidence being considered, for example, a footprint or DNA. Bayes theorem provides the appropriate formal structure for taking into account forensic evidence, and can be written as $\frac{p(G|d)}{p(I|d)} = \frac{p(d|G)}{p(d|I)} \times \frac{p(G)}{p(I)}$.

Here $p(G|d)$ represents the absolute probability that the suspect is guilty, and $p(I|d) = 1 - p(G|d)$ the probability that they are innocent (although such quantifications would not normally be allowed in a legal trial). This is communication of direct, absolute uncertainty.

$p(d|G)/p(d|I)$ is the 'likelihood ratio', which expresses the relative support given to Guilt over Innocence by the item of evidence. In DNA evidence, this would typically be the inverse of the 'random-match probability', the chance that the DNA would be found on a randomly chosen member of other possible culprits, typically of the order of more than 1 in 10 million. Note that this would *not* mean there was a 1 in 10 million chance that the suspect was innocent—this error in interpretation is known as the 'prosecutor's fallacy'. Likelihood ratios are, therefore, expressions of relative uncertainty and commonly communicated in bands, so that a likelihood ratio between 1000 and 10 000 would be interpreted as 'strong support' for the guilt of the suspect [31]. Likelihood ratios could be multiplied together for independent items of forensic evidence to provide an overall level of support of the evidence for guilt: this is currently not permitted in UK courts.

Finally, indirect uncertainty can be expressed as the confidence in the claim of '10 million', which would be based on the quality and size of the database relevant to this case, and other factors such as potential contamination.

This division neither matches the traditional split into first/second-order nor quantified/unquantified uncertainty. Direct uncertainty may be assessed through modelling or through expert judgement, involving aspects of both first- and second-order uncertainty, and may be quantified to a greater or lesser extent, whereas indirect uncertainty is a reflexive summary of our confidence in the models or the experts.[4] An example of a system designed to communicate indirect uncertainty is the GRADE system of summarizing overall quality of evidence, which we discuss further in §4.

Box 3 demonstrates the difference between direct and indirect uncertainty within a legal context where we hope the distinction between the two levels is particularly clear.

## 3.4. The magnitude of the uncertainty

It seems intuitive that the magnitude of uncertainty being communicated would likely influence the audience's response to it—it could indeed be seen as one of the commonest goals of uncertainty

---

[4]If we feel we 'know' the probabilities (pure first-order uncertainty), for example, when we have an unbiased coin, then in a sense there is no indirect uncertainty, since there are no caveats except for our assumptions. But as soon as assumptions are expressed, there is the possibility of someone else questioning them, and so *they* may have caveats. This reinforces the fact that epistemic uncertainty is always subjective and depends on the knowledge and judgements of the people assessing the uncertainty.

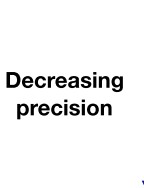

**Decreasing precision**

i.  A full explicit probability distribution
ii.  A summary of a distribution
iii.  A rounded number, range or an order-of-magnitude assessment
iv.  A predefined categorisation of uncertainty
v.  A qualifying verbal statement
vi.  A list of possibilities or scenarios
vii.  Informally mentioning the existence of uncertainty
viii.  No mention of uncertainty
ix.  Explicit denial that uncertainty exists

**Figure 2.** Alternative expressions for communicating direct uncertainty about a fact, number or scientific hypothesis.

communication. However, it is often not explicitly drawn out as an important variable in empirical work (see §6 where this is discussed).

# 4. In what form is the uncertainty communicated?

## 4.1. Expressions of uncertainty

Each of the different kinds of uncertainty discussed in §3 can be expressed in a wide range of forms, and these forms may affect the effects of uncertainty communication. In this section, we consider the space created by the different dimensions that we have used to define uncertainty and how it can be filled by different expressions.

### 4.1.1. Direct uncertainty (absolute expressions)

Direct uncertainty about a fact, number or scientific hypothesis is the type of uncertainty which can be the most precisely expressed and therefore lends itself to the widest possible range of forms of expression. In figure 2, we list these forms, in order of their decreasing precision (capability of expressing detail of magnitude).

Expressions at the top of the list can be considered as Donald Rumsfeld's 'known unknowns' [24], whereas his 'unknown unknowns' would fall under expression vii, in which uncertainty is acknowledged without being able to provide a list of possibilities.

In order to explore whether each in this list of nine expressions of absolute, direct uncertainty could be applied to all three objects of uncertainty in our framework - categorical or binary *facts*, continuous variables (*numbers*) and *models* - we set out to find real examples of each in use. The results of our search are shown in table 1. We were not able to find examples for each cell in the table, illustrating where some usages are rare at best. However, our intention was both to test the comprehensiveness of our framework and to illustrate it to help others identify how it can be applied. We fully admit that some of the entries are ambiguous: for example, as we shall see in box 4, the IARC's claim of a 'probable carcinogen' is more an indirect summary of the quality of evidence for carcinogenicity, rather than a direct expression of probability and so may not belong in the table at all.

### 4.1.2. Direct uncertainty (relative expressions)

Relative uncertainty about competing hypotheses or values for a measure can also be expressed in different forms. Verbal comparisons include statements of the form 'A is more likely than B', while numerical expressions include likelihood ratios for comparing facts and scientific hypotheses, likelihood functions for relative support for different numbers, and comparative measures of model adequacy such as the Akaike Information Criterion [61] or Bayesian Information Criterion [62]: formal definitions are provided in the Technical appendix on statistical approaches to communicating epistemic uncertainty. *P*-values are a measure of conflict between data and a hypothesis, and are certainly not direct expressions of a probability of hypotheses. However, as described in the Technical appendix, in many circumstances they correspond to a specific confidence interval for a numerical parameter.

### 4.1.3. Indirect uncertainty (quality of underlying evidence)

Methods for communicating the quality of the underlying evidence do not give quantitative information about absolute values or facts, but summarize the subjective confidence we have in any claim.

**Table 1.** Exploring examples of real-world use of each of the nine possible expressions of direct, absolute uncertainty about each of the three possible objects of uncertainty: facts (categorical variables), numbers (continuous variables) or hypotheses (models).

| expression | object = potentially observable facts, categorical and binary measures | object = numbers (i.e. continuous variables) either directly measurable or constructed | object = models and hypotheses |
|---|---|---|---|
| (1) a full explicit probability distribution, communicated numerically or visually | *domain: history.* In forensic analysis of the skeleton found underneath a car park in Leicester in 2012 [32], researchers claimed a 96% probability that the individual had blue eyes and 77% probability that he had fair hair. Combining the forensic evidence using the method of likelihood ratios outlined in box 2, it was concluded that the probability that the skeleton is that of Richard III lies between 0.999994 and 0.9999999. This was deemed sufficient to warrant a full burial in Leicester Cathedral. | *domain: public health.* Full posterior probability distributions for the uncertain prevalance of Hepatitis C in England are provided graphically by Harris et al. [33]. | *domain: biology.* Posterior probabilities of alternative phylogenies (evolutionary pathways) are produced by software, e.g. MrBAYES [34]. |
| (2) summary of a distribution communicated numerically or visually e.g. 95% confidence intervals, error bars, margins of error, fan charts | | *domain: history.* Using household survey methods, the victims of violence in the Iraq war have been estimated as 601 027 deaths up to June 2006 (95% confidence interval of 426 369–793 663) [35]. These figures are contested, and there is further uncertainty due to disagreement between sources, for example, a different survey estimated 151 000 deaths due to violence (95% uncertainty range, 104 000–223 000) [36]. | |

(Continued.)

**Table 1.** (*Continued.*)

| expression | object = potentially observable facts, categorical and binary measures | object = numbers (i.e. continuous variables) either directly measurable or constructed | object = models and hypotheses |
|---|---|---|---|
| (3) a rounded figure, range or an order-of-magnitude assessment *e.g. number between x and y, up to x (without information about the underlying distribution).* | | *domain: conservation biology.* From Global Wild Tiger Population Status, April 2016 document [37]: 'In 2014, India undertook its largest, most intensive and systematic national tiger population survey. The survey included new areas and more intensive sampling. The survey estimated the population to range between 1945 to 2491 with a mean estimate of 2226 tigers'. | |
| (4) a predefined categorization. | *domain: climate change.* From the 2013 IPCC summary for policy makers of Working Group 1 (The Physical Science Basis): 'It is likely that the rate of global mean sea level rise has continued to increase since the early 20th century'. Likely is defined as 66–100% likelihood. [38] | | *domain: public health.* The International Agency for Research on Cancer has classified RF fields as 'possibly carcinogenic to humans', based on limited evidence of a possible increase in risk for brain tumours among cell phone users, and inadequate evidence for other types of cancer. This is one of a set of predefined categories expressing certainty of carcinogenicity (box 4) [39]. |

(*Continued.*)

**Table 1.** (Continued.)

| expression | object = potentially observable facts, categorical and binary measures | object = numbers (i.e. continuous variables) either directly measurable or constructed | object = models and hypotheses |
|---|---|---|---|
| (5) qualifying verbal statements applied to a number or hypothesis. e.g. *around x, roughly x, very likely x, probably x.* e.g. *not very likely that.. likely that.. if not defined more formally.* | *domain: history*. A quote from an essay on the trail of Jeanne d'Arc by Pierre Champion: 'he had studied theology at Paris for eight years, and that the provincial chapter had designated him to "read the Bible". It is, therefore, not very likely that he could have been master of theology by 1431, at least at the University of Paris'. [40]  *domain: politics*. From an MSNBC interview with Senator Jeff Merkeley: 'Q: You're saying it looks like some Americans helped the Russians and the bigger question is just whether they were affiliated with Donald Trump or not? A: Yes, I'm saying it is very likely — it's very likely and we need to get to the bottom of who was involved here'. [41] | *domain: biology*. From a Science News article, titled 'How Much Did the Dodo Really Weigh': 'Andrew Kitchener set about trying to figure out what a dodo would have looked like. [. . .] Kitchener eventually concluded that the dodo was a much slimmer bird than artists made it look, probably in the range of 10.5 to 17.5 kilograms'. [42] | |

**Table 1.** (*Continued.*)

| expression | object = potentially observable facts, categorical and binary measures | object = numbers (i.e. continuous variables) either directly measurable or constructed | object = models and hypotheses |
|---|---|---|---|
| (6) list of possibilities<br><br>*e.g. it is x, y or z.* | *domain: health.* From a fact sheet on Abnormal Prenatal Cell-free DNA Screening Results by the National Society of Genetics Counsellors: 'An abnormal result may indicate an affected fetus, but can also represent a false positive result in an unaffected pregnancy, confined placental mosaicism, placental and fetal mosaicism, a vanishing twin, an unrecognized maternal condition or other unknown biological occurrence'. [43]<br><br>*domain: legal epidemiology.* 'In Barker v. Corrs Lord Hoffman had specifically considered the situation where the Claimant suffered lung cancer that might have been caused by exposure to asbestos or by other carcinogenic matter but might also been caused by smoking and it could not be proved which was more likely to be the causative agent'. [44] | *domain: intelligence.* Barack Obama in the Channel 4 television documentary 'Bin Laden: Shoot to Kill' (2011): 'Some of our intelligence officers thought that it was only a 40 or 30% chance that Bin Laden was in the compound. Others thought that it was as high as 80 or 90%. At the conclusion of a fairly lengthy discussion where everybody gave their assessments I said: this is basically 50−50'. [45] | *domain: biology.* From a National Geographic article on Dinosaur Extinction: 'Scientists tend to huddle around one of two hypotheses that may explain the Cretaceous extinction: an extraterrestrial impact, such as an asteroid or comet, or a massive bout of volcanism'. [46] |

*(Continued.)*

**Table 1.** (Continued.)

| expression | object = potentially observable facts, categorical and binary measures | object = numbers (i.e. continuous variables) either directly measurable or constructed | object = models and hypotheses |
|---|---|---|---|
| (7) humility: mentioning uncertainty *statements about the possibility of being wrong, the fact that uncertainty exists, unknown unknowns, etc.* | *domain: law.* From a report in Computing: 'Leading legal experts disagree about whether the EU's General Data Protection Regulation (GDPR) is in fact already in force in the UK. . . . Speaking at a recent Computing event, Bridget Kenyon, head of security at University College London, explained that the GDPR is already in force, in her opinion. "Actually GDPR is in force now, but what's not in place yet is the penalties", said Kenyon. 'So if there's a breach now, the ICO could hold on to it and give you the penalties in May 2018', she argued. Computing queried both the ICO itself, and several legal experts on the veracity of this claim, and found conflicting opinions, suggesting a degree of uncertainty rules in the industry'. [47] | *domain: law (on film).* A quote from the film 12 Angry Men: 'Nine of us now seem to feel that the defendant is innocent, but we're just gambling on probabilities. We may be wrong. We may be trying to return a guilty man to the community'. [48] *domain: law.* From the document 'Findings of facts and reasons' in the case The judicial authority in Sweden v. Julian Paul Assange: 'He does not agree that he was informed that she had made a decision to arrest Mr Assange, and believes he was not told until 30th September. I cannot be sure when he was informed of the arrest in absentia'. [49] *domain: forensics.* The court permitted the expert to testify that 'in my opinion, the DNA profiling evidence provides support for the view that some of the DNA recovered was from Ashley Thomas, but I am unable to quantify the level of this support'. [50] | *domain: physics.* From a book chapter on the evolution of Quantum Field Theory by Gerard 't Hooft: 'At first sight, quantum chromodynamics (QCD) seems to be an exception: the theory is renormalizable, and by using lattice simulations one can address its infrared behaviour. Here, however, we have to keep in mind that mathematical proofs for the internal consistency of this theory are still lacking. Most of us believe without doubt that the theory will work fine under all circumstance, with unlimited precision in principle, and we have good reasons for this belief, but we cannot be sure'. [51] |

**Table 1.** (Continued.)

| expression | object = potentially observable facts, categorical and binary measures | object = numbers (i.e. continuous variables) either directly measurable or constructed | object = models and hypotheses |
|---|---|---|---|
| (8) no mention of uncertainty | *domain: forensics.* Extract from the judgement R v. Deen, 1994 [52]: 'Q: So the likelihood of this being any other man but Andrew Deen is one in 3 million? A: Yes Q: What is your conclusion? A: My conclusion is that the semen has originated from Andrew Deen. Q: Are you sure of that? A: Yes' [N.B. This is a classic case of the 'prosecutor's fallacy' and the expert witness is drawing an incorrect conclusion from the evidence.] | *domain: economics.* The Office for National Statistics Statistical Bulletin, UK labour market: October 2017, reports the unemployment figures: 'For June to August 2017, there were 1.44 million unemployed people, 52 000 fewer than for March to May 2017 and 215 000 fewer than for a year earlier'. [53] *domain: health.* Q-Risk cardiovascular risk calculator [54]: 'Your risk of having a heart attack or stroke within the next 10 years is 12.3%'. | *domain: climate change.* From the American Association for the Advancement of Science Board Statement on Climate Change: 'The scientific evidence is clear: global climate change caused by human activities is occurring now, and it is a growing threat to society'. [55] |
| (9) explicit denial uncertainty exists | *domain: politics.* From a speech by US vice president Dick Cheney to the Veterans of Foreign Wars (VFW) national convention in Nashville, Tennessee, on 26 August 2002: 'Simply stated, there is no doubt that Saddam Hussein now has weapons of mass destruction'. [56] *domain: legal.* From the judgement in the case Regina v. Pendleton (on Appeal from the Court of Appeal (Criminal Division)): 'We have no doubt that the conviction was safe'. [57] | | *domain: biology.* 'The statement that organisms have descended with modifications from common ancestors—the historical reality of evolution—is not a theory. It is a fact, as fully as the fact of the Earth's revolution about the sun'. [58] *domain: physics.* University of California, Berkeley physicist Daniel McKinsey in an interview with CBC: 'It's certainly there. We know dark matter exists' [59] |

**Box 4.** Ways that institutions try to simplify uncertainty communication—and the problems that can arise as a result.

When institutions or regulatory bodies have to communicate uncertainty, they often attempt a simplified rule-based classification, which can easily be followed by all members of the organization. However, devising such a system without acknowledging the potential for confusion has led to problems.

For example, the International Agency for Research on Cancer (IARC) has a long-standing series of monographs assessing the carcinogenicity of exposure to various potential mutagens. For different items of evidence, a scale for the quality of the research (indirect level) is combined with the apparent strength of evidence (a direct, relative level), leading to classifications such as 'sufficient evidence of carcinogenicity in humans' and 'evidence suggesting lack of carcinogenicity in animals'. An algorithm then combines these assessments for different strands of evidence to finally classify different agents on the direct, four-category scale for scientific hypotheses mentioned in table 1: 'Carcinogenic to humans', 'Probably carcinogenic to humans', 'not classifiable', 'Probably not carcinogenic to humans' [39].

However, this scale does not give any numerical interpretation to 'probably', and gives no information about the size of any carcinogenic effect, leading to considerable confusion in public communication. For example, processed meats and cigarettes are placed in the same category— 'Carcinogenic to humans'—not because they are equally carcinogenic, but because the evidence around each is judged equally suggestive of a link.

Somewhat similarly, the American College of Medical Genetics and Genomics uses a set of judgemental rules to classify genetic variants in terms of their likelihood of being pathogenic, proposing that 'the terms 'likely pathogenic' and 'likely benign' be used to mean greater than 90% certainty of a variant either being disease causing or benign to provide laboratories with a common, albeit arbitrary, definition [60]'. But there is no firm empirical, numerical basis for 'certainty' to be determined and no indication to a patient regarding how possessing the 'pathogenic' variant might affect them (in terms of likelihood or severity of any effect). Patients who are given the information that they have been found to have a 'likely pathogenic' variant are therefore no better informed about the possible consequences for them.

In order to attempt to *assess* indirect uncertainty, a number of fields have established checklists to try to assess the quality of evidence in as objective a way as possible. These may relate to either an *individual claim*, such as the CONSORT system, for determining the characteristics of the claims resulting from a randomized controlled trial [63], and the Maryland Scale of Scientific Methods, for determining the strength of a crime prevention study [64], or the *totality of evidence*, attempting to take into account the quality, quantity and consistency of multiple studies to give an overall assessment of the confidence we can have in a particular assertion; see [65,66] for reviews. These tools provide the basis for systems that attempt to *communicate* overall quality of evidence (although the distinction between methods of assessment and methods of communication of indirect uncertainty is rarely made).

Many methods of communicating indirect uncertainty have been developed in different fields. Limitations in the underlying evidence might be summarized by qualitative verbal caveats, or an ordered set of categories (which may be communicated numerically, graphically or verbally). For example, the GRADE Working Group has established a scale for communicating the quality of the evidence underlying claims about the effects of medical interventions, which ranges from 'Very low quality', graphically represented as a single plus symbol and/or circle, to 'High Quality', graphically represented as 4 plus symbols and/or circles [67]. Other examples are the 'padlock' ratings used by the UK's Educational Endowment Foundation [68] (figure 3), or the US National Intelligence Council's recommendation that intelligence analysts provide a qualitative assessment of analytic confidence on a high/medium/low scale 'based on the scope and quality of information supporting our judgments' (p. 5 [69]). In effect, such ordered scales provide a form of 'star-rating' for the conclusions.

These broad categorical ratings are used when the impact of poorer quality evidence is difficult to quantify. One issue with such broad categorical ratings or verbal descriptions (e.g. 'high quality') is that their meaning is in part dependent on the context of their use: at what threshold evidence is classified as high quality or low quality might depend on the research field or topic. The audience, especially if they are non-experts, might not be aware of this. In addition, research has shown that there is considerable variation in people's interpretation of verbal probability and uncertainty words

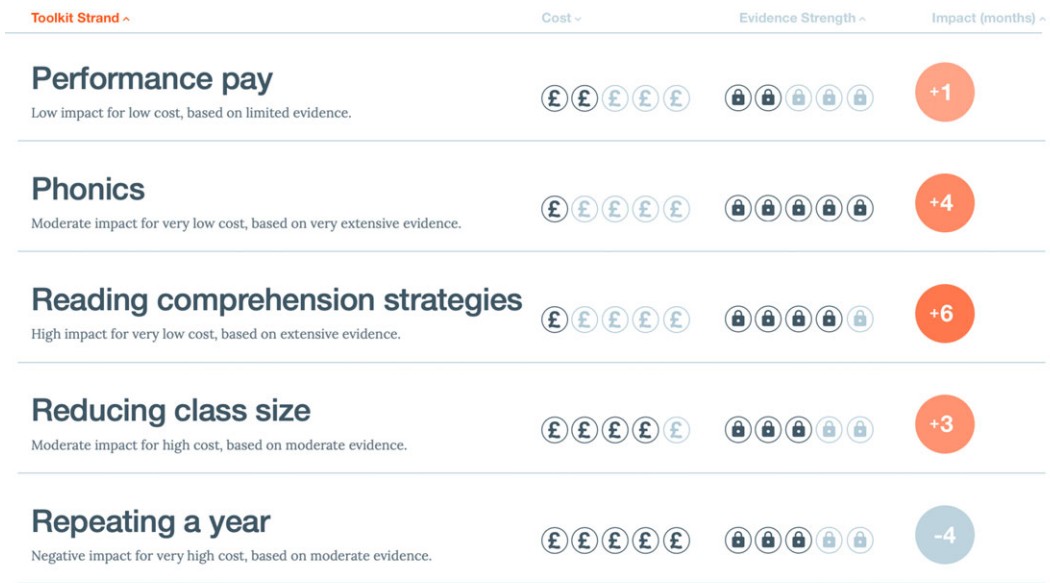

**Figure 3.** The Education Endowment Foundation's summary of five educational interventions, in terms of cost, evidence strength and impact measured in months of educational advancement. 'Evidence strength' is a summary of the quality of evidence (indirect uncertainty) underlying the estimates of impact on an ordered categorical scale, analogous to a 'star-rating'.

such as 'likely' [70–73]. There might be a similar variability in what people interpret 'high quality' or 'low quality' to mean, which might make such broad categorical ratings or verbal descriptions less effective. However, it might be hoped that, with additional knowledge or judgement, some caveats could contribute to a direct, quantitative expression of uncertainty: for example, by widening a confidence interval due to the potential systematic bias in a survey.

In practice, both direct and indirect uncertainties are often expressed simultaneously, as demonstrated by the following Cochrane systematic review:

'We found that giving immunotherapy, mainly vaccine-based (aiming to activate the host immune system to induce human immune response to tumour-specific antigens), after surgery or radiotherapy did not, on average, make people live longer'. 'We found a small, but not statistically significant, improvement in OS (HR 0.94, 95% CI 0.83 to 1.06; $P = 0.35$), . . . ; high-quality evidence)' [74]

In this example, the number of primary interest is the hazard ratio (HR)—the proportional change in overall survival (OS) for people given immunotherapy. The HR is estimated to be 0.94, corresponding to a 6% reduction in the risk of dying in a fixed time period, and the direct, absolute uncertainty around this figure is communicated as a 95% confidence interval (0.83–1.06). This is a 'ii' on our scale of methods of expressions for communicating direct, absolute uncertainty—a summary of a distribution for the true value.

The $p$-value (0.35) expresses the weak evidence that the true value of the HR is different from 1 (i.e. that those given immunotherapy really did live longer than those who were not given this therapy). Formally, this says there is a 35% chance of having observed at least the 6% relative change in survival if there were actually no effect of the immunotherapy (and all the other modelling assumptions are correct)—an effect not considered to be statistically significant (when the alpha level is set at the conventional 0.05). This $p$-value can be translated to an absolute expression: it means that a 65% confidence interval for the true effect just excludes 1.

The quality of the evidence behind these direct claims is expressed through the GRADE scale, with 'high-quality' and the symbolic 4 '+' (figure 4) meaning that we as readers can put good faith in both the confidence interval and the $p$-value.

This amount of information could potentially be overwhelming, and difficult to illustrate graphically and interpret, so organizations have (apparently without recourse to empirical testing) sought less comprehensive forms of uncertainty communication. These may try to conflate the different levels of uncertainty to try to simplify the message, but box 4 shows this has clear potential for confusion. We cite these examples as a useful warning to practitioners considering constructing a 'simplified' method of communicating the uncertainties in their field.

| outcomes | anticipated absolute effects* (95% CI) | | relative effect (95% CI) | no. participants (studies) | quality of the evidence (GRADE) | comments |
|---|---|---|---|---|---|---|
| | assumed risk with surgical treatment only (control group) | corresponding risk with immunotherapy plus surgery (experimental group) | | | | |
| overall survival<br><br>duration of follow-up: varied between studies (the median follow-up time ranged from 37.7 months to 70 months) | the median overall survival time ranged across control groups from 22.3 to 60.2 months | the median overall survival time ranged across experimental groups from 25.6 to 62.0 months | HR 0.94 (0.83 – 1.06) | 3693<br><br>(3 RCTs) | ⊕⊕⊕⊕<br>HIGH | |

**Figure 4.** A Cochrane 'summary of findings' table illustrating both direct (confidence interval) and indirect (GRADE scale) levels of uncertainty [74].

Methods have been proposed for turning indirect into direct uncertainty. In the context of a meta-analysis of healthcare interventions, Turner *et al.* [75] demonstrate that experts can take caveats about lower-quality studies and express their impact in terms of subjective probability distributions of potential biases. When these are added to the nominal confidence intervals, the intervals appropriately widen and the heterogeneity of the studies are explained. These techniques have been tried in a variety of applications [76,77] and show promise, although they do require acceptance of quantified expert judgement.

## 4.2. Format and medium of uncertainty communication

The other important aspects of the 'how' in our framework of uncertainty communication (figure 1) are the format and the medium. Uncertainty can be expressed in one (or a combination) of three different formats: visual, numerical and/or verbal. The appropriate format in part depends on the medium of communication, which might be written and printed official reports, online websites, smart phone applications, print media, television, or spoken in person or on the radio. We therefore consider these two aspects of format and medium together. However, these different formats have the potential to carry different levels of information and therefore choosing one is not simply a design choice—it can influence the type of expression of uncertainty available and its potential effect on the audience. Expressions i–iv in §4.1 are predominantly numerical or visual expressions; expressions v-ix are predominantly verbal (and less precise).

Whereas numerical (numbers) and verbal (words) communication are relatively constrained in their design, there are a variety of ways to communicate uncertainty visually. Examples of common ways to visualize epistemic uncertainty around a number, expressed as an estimate with a range ('i' or 'ii' in our scale), are presented in figure 5. Error bars are widely used in scientific and other publications to illustrate the bounds of a confidence interval, but provide no indication of the underlying distribution of the number. Other visualizations attempt to give an (approximate) idea of this underlying distribution: for example, diamonds, which are often used when considering treatment effects in a medical meta-analysis, or violin plots, which are designed to give a more accurate idea of the underlying distribution. Fan plots are designed to show the bounds of several different confidence intervals (often coloured to emphasize the changing probability density going further from the point) and are used, for example, by the Bank of England when communicating past and forecasted future GDP estimates. Finally, density strips are the most accurate representation of the underlying probability distribution around the point estimate.

Such visualizations have primarily been explored within the context of future risks, and Spiegelhalter *et al.* [78] reviewed different types of visualizations of uncertainty about the future, such as bar charts, icon arrays, fan charts or probability distributions. By contrast, MacEachren *et al.* [79] reviewed different types of visualization of epistemic uncertainty in spatial data such as maps or medical imaging: various attributes of the colours and lines used to construct a map may be varied to illustrate uncertainty [79], while colour saturation, crispness and opacity, as well as the addition of specific indicators (glyphs) may give uncertainty information (such as the IPCC's use of the '+' sign on its climate maps). One main conclusion from both reviews is that whereas a wide variety of types of graphics have been developed to communicate probabilities, there is limited empirical evidence of how alternative formats may influence audience understanding and response.

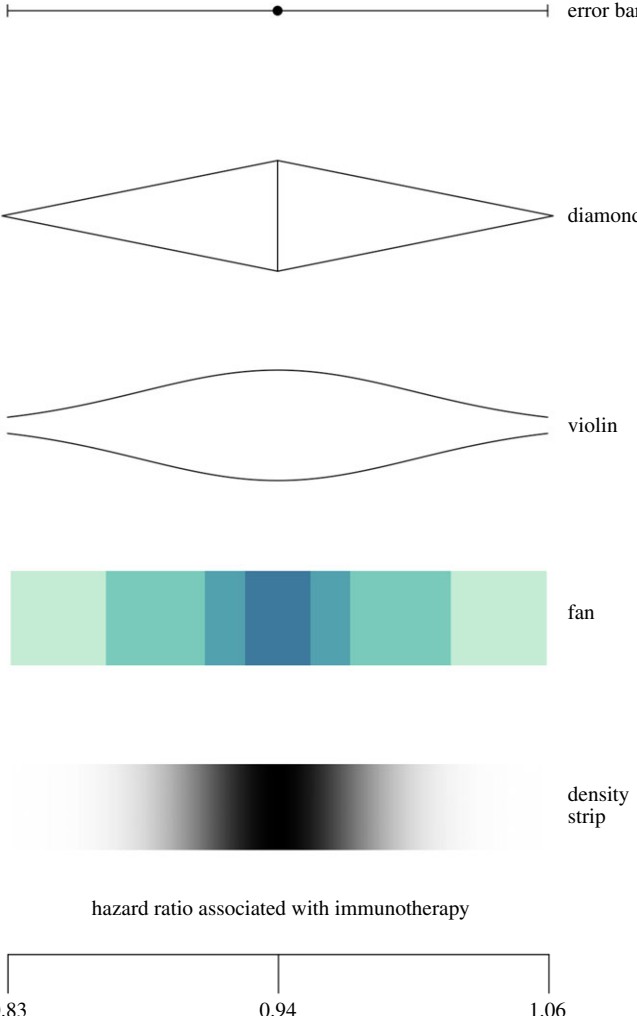

**Figure 5.** Common expressions of uncertainty around numbers, illustrated using the immunotherapy example in figure 4: an (i) error bar; (ii) diamond; (iii) violin plot; (iv) fan plot and (v) density strip.

# 5. Communicated to whom?

The goal of communication is to affect an audience in some way: to inform, motivate, instruct or influence people. The effects of uncertainty communication depend not only on the aspects discussed so far, such as the object of uncertainty and the format of communication, but also on the characteristics of the target audience and on the relationship between the audience and the communicator, the topic or source of the uncertainty. Important differences between individuals, such as their level of expertise, prior attitudes [80], numeracy skills [80,81], education level [82] or optimism [83,84], might mean that the same communication of uncertainty affects people differently. For example, people's interpretation of information can be shaped by situations in which a topic is contested or has become politicized; or by the situational context in which the information exchange takes place (e.g. under high stress). To illustrate, studies show that people selectively seek out information that is consistent with their prior beliefs and sometimes process it more fluently than information that is inconsistent with their prior beliefs, phenomena variously described as motivated cognition and confirmation bias [85–88]. Through these processes, the audience's pre-existing beliefs or attitudes towards the communicator, topic or object of uncertainty might influence or change the effects of uncertainty communication.

As a case in point, Dieckmann *et al*. [80] found that when participants judged uncertainty communicated as a range around the predicted average global surface temperature increase, people who indicated more climate change acceptance were more likely to perceive a normal distribution or a distribution in which higher values were more likely. By contrast, people who indicated less climate change acceptance were more likely to perceive a uniform distribution or a distribution in which lower values were more likely [80]. In addition, people's prior beliefs about the source of uncertainty

for a certain topic might influence the effects of uncertainty communication. Indeed, for some topics or in some decision settings, people might expect uncertainty (for example, during weather forecasts [89]), whereas in others, they might be less welcoming of uncertainty information [90,91].

Unfortunately, there is very little systematic empirical work studying these effects on the communication of epistemic uncertainty. In §6, we highlight where these factors have been part of the studies, and we will examine the important issue of credibility and trust in more detail in §6.3. The key point is that it is important for effective communication to know your audience.

# 6. Communicated to what effect?

The fifth and final section of our framework (figure 1) concerns the psychological effects of communicating uncertainty. Evaluating these effects is important, as this can help establish whether the communication of uncertainty has achieved its intended goal and whether it might have had any unintended consequences.

We reviewed what is known about the impact of communicating epistemic uncertainty on human cognition (understanding), emotion (feeling), trust and decision-making. We did this by searching the literature for empirical research in psychology, judgement and decision-making, and related disciplines. This review informed the construction of our framework; and here we use the framework in turn to structure the reporting of the findings of the review.

Before reporting those findings, we should explain that it is important to distinguish epistemic or scientific uncertainty from the subjective psychological experience of uncertainty—the feeling which might be the *result* of an ambiguous communication. Psychological uncertainty is a human experience, usually defined as an aversive psychological state in which an individual lacks information. In other words, it describes the subjective feeling of 'not knowing' [92,93]. The psychological experience of uncertainty has been extensively investigated: the fact that people are averse to ambiguous information has been referred to as 'one of the most robust phenomena in the decision-making literature' [94, p. 1]. This is not the subject of our reviewing; we focus on uncertainty that is the property of a fact, number, or model that is being communicated.

Second, we limit the scope of our review to the psychological effects of communicating *epistemic* uncertainty as defined in our introduction. This follows our argument that it is important to conceptually distinguish *aleatory* uncertainty (unknowns due to inherent indeterminacy or randomness) from epistemic uncertainty due to our lack of knowledge about a past or present fact or number (which often could, in principle, be known). In turn, of course, epistemic uncertainty about the past or present may or may not influence a future event. We expect that there may be important differences in the psychological impact of communicating aleatory versus epistemic uncertainty. In fact, Fox & Ülkümen [95] allude to this distinction by recalling one of the most difficult decisions Barack Obama had to make during his presidency. In deciding whether or not to launch an attack against a compound that was suspected to house Osama Bin Laden, he faced two qualitatively different forms of uncertainty. The first concerns uncertainty about a measurable fact (either Bin Laden resided at the compound or he did not) but the second type of uncertainty revolved around possible futures: is the mission going to be successful or not? Fox and Ülkümen make a compelling argument that judgement under uncertainty is indeed likely to invoke a conscious or unconscious attribution to epistemic and aleatory forms of uncertainty. For example, people seem to express psychological differences in these two forms of uncertainty in natural language: whereas pure epistemic uncertainty is often expressed as the degree of confidence in one's knowledge about a fact, aleatory uncertainty is more likely to be expressed in probabilities associated with future outcomes [96].

We agree with Fox & Ülkümen that most researchers continue to treat uncertainty as a 'unitary' construct [95]. At present, the existing research we have reviewed has predominantly investigated reactions to aleatory uncertainty, or has conflated the two kinds. For example, although epistemic uncertainty may be part of an ambiguous experimental situation (e.g. not knowing the exact probability distribution of a gambling task), ambiguity aversion is often—but not exclusively—about people's aversion to using this information for making decisions about future event with a random component. This is not our focus, but we recognize that the two types of uncertainty can interact and sometimes one may qualify the other. Accordingly, we will sometimes draw on relevant work about ambiguity to inform our discussion of the effects of epistemic uncertainty, because few existing empirical studies have clearly made this distinction.

Thirdly, here we are only considering one level of uncertainty: direct uncertainty. Empirical work on the effects of communicating indirect uncertainty (quality of evidence) deserves a separate treatment elsewhere.

We structure our narrative review according to the expressions of direct uncertainty we identified in §4.1, ranging from expression (i) full distributions to (ix) denying uncertainty. As far as we know, expressions viii (not mentioning uncertainty) and ix (denying uncertainty) have not been explicitly studied, and are left to the Discussion. Where conclusions from studies appear to be drawn across multiple forms of expression, we have conflated those different categories under the same headings below for clarity.

We also note that this literature is particularly widely scattered and that the review is not meant to be systematic. We have cited all relevant studies that we have found, and aimed to cover a broad array of studies representative of findings in the field. Because we cannot offer an in-depth review of each single study, we have instead opted to highlight particularly insightful relevant studies in more detail for illustrative purposes throughout.

## 6.1. The effect of communicating uncertainty on cognition

The term cognition is generally used to describe mental processes or actions that make up 'thinking', such as the processing, interpretation and retrieval of information, including attention, memory, judgement and evaluation. In this section, we discuss psychological research that has investigated how various expressions of uncertainty influence people's interpretation and understanding of information.

### 6.1.1. Expressions i and ii: full and summaries of distributions

Some research has explored how people interpret error bars and related visualizations of mean and error in graphs. For example, Correll & Gleicher [97] examined the extent to which people could make accurate statistical inferences about information presented in bar charts with error bars, compared to alternative representations of mean and error, for example violin plots and gradient plots (e.g. figure 4). Bar charts can lead to biased interpretations: they suffer from 'within-the-bar-bias', where values within the bar are seen as more likely than values outside the bar, and error bars do little to correct this bias [98]. Gradient plots and violin plots however are visually symmetric and continuous, and indeed Correll & Gleicher's results showed that these can help people to make judgements that are more in line with statistical expectations (i.e. the actual probability of a certain value).

### 6.1.2. Expression iii: ranges

Early research by Johnson & Slovic [99] examined the presentation of uncertainty through range estimates. Their hypothetical scenario involved an example that described the likelihood of getting cancer from drinking water that contains a particular chemical (1 in 1 000 000). Although this number was presented as the most likely estimate, the message noted that the true risk could be as low as zero or as high as 10 in 1 000 000. Using a convenience sample of the American public, Johnson & Slovic's research suggested a potential upward bias, where about half of respondents thought that the highest number is the correct figure (although the other half of respondents disagreed; for comparative evidence, see also [91,100]).

Johnson & Slovic's [99] research suggests that people's interpretation of the distribution underlying a range can differ. Research by Dieckmann *et al.* [81] showed similar results for ranges representing future uncertainty: when presented with just a numerical range, most people indicated that they perceived either a uniform or normal distribution, but some perceived a u-shaped distribution. They found that providing more information, such as a best estimate within the range or including a picture of a normal density plot, led more people to indicate that they perceived a normal distribution.

In subsequent research, Dieckmann *et al.* [80] showed that people's interpretation of numerical ranges can be influenced by motivated reasoning. Motivated reasoning is a cognitive shortcut where people interpret information in a way that is consistent with a predetermined conclusion [101]. In general, this appears to lead people to perceive uncertain information as having more variance [86], which facilitates biased assimilation. For example, in the study of Dieckmann *et al.* [80], people's prior beliefs about global warming changed their interpretation of the distribution of a numerical range in a way that was congruent with their personal beliefs. Yet, such biases can be corrected. For example, Dieckmann *et al.* [80] showed that opportunistic tendencies to interpret uncertainty in a motivated manner were reduced by pointing out how to correctly interpret the uncertainty in question [102].

Overall, this research suggests that depending on their exact presentation, numerical ranges can be interpreted differently by different people. In general, graphics may help people recognize uncertainty; but it is important to choose an appropriate type of graphic to convey uncertainty properly. Some information about the distribution seems important for people to interpret ranges more accurately.

### 6.1.3. Expressions iv and v: predefined categorizations and qualifying verbal statements

A sizeable body of research has focused on determining how people interpret various verbal expressions of uncertainty and how consistent such interpretations are. Interpretation is mostly determined by assigning probability percentages to verbal uncertainty expressions or by rank ordering them. Several literature reviews have suggested that whereas individuals are internally consistent, there is substantial variability between individuals in their interpretation of uncertainty expressions [70–73]. This suggests that whereas one person would interpret a word such as 'likely' similarly across different encounters (for example, meaning 'at least 50% chance'), another person could have a very different interpretation of what 'likely' means (for example, 'at least 75%').

Other studies have even contested the degree of internal consistency in people's interpretation of verbal probability expressions. For example, verbal expressions of probability can be heavily dependent on the psychological context [103,104]. A few studies have offered partial explanations for such context-effects, such as the perceived base rate of an event [105]. It is important to note that although people often seem to have a (potentially strategic) preference for verbal expressions of probability [104], these cognitive inconsistencies are problematic for communication [103].

This has been pointed out most clearly in the domain of climate change. Budescu et al. [71] examined people's interpretation of verbal probability expressions used in reports from the IPCC, which covered both statements with aleatory and epistemic uncertainty. They found that people consistently misinterpret the intended meaning of IPCC's uncertainty expressions, mostly indicating probabilities that were less extreme than IPCC's guidelines prescribe. For example, whereas the IPCC intends the expression 'very likely' to indicate a probability of 90% or higher, the typical or median response from participants was approximately 65–75%. Furthermore, there were large differences between individuals in their interpretation of the expressions, which were found to be associated with prior beliefs about climate change. Indeed, several studies have observed that qualifiers such as 'most' or a 'majority' in statements of fact, such as 'the majority of climate scientists have concluded that human-caused climate change is happening', are interpreted at around 60%, whereas the communicator typically intends to convey scientific agreement between 90 and 100% [106–109]. Importantly, such inconsistencies can be reduced by including numeric information alongside or in place of verbal probability expressions [71,107,110].

Other interpretation issues remain. For example, Teigen et al. [111] found that when people are presented with a histogram showing the actual frequencies of occurrence of quantifiable events, such as the battery life of a laptop or how long it takes to post a letter from Norway to the USA, and are then asked to choose a value that represents 'unlikely' or 'improbable', people consistently choose values that have a near 0% frequency of occurrence—as opposed to picking low values that have actually occurred in the sample. For example, when battery life is shown to range between 2.5 and 4.5 h, people think that 5 or 6 h are better examples of 'improbable' duration than those that actually occur in 10% of cases. This 'extremity' effect is thought to be influenced by framing, where negative (unlikely) or positive (likely) verbal frames focus people on the (non-)occurrence of an event [112,113].

In sum, this research suggests that there is considerable variation in how verbal expressions of uncertainty are interpreted, both between different people and within the same person in different contexts. One solution to decrease variability in interpretation could be to supplement verbal expressions with numeric information: research has demonstrated that this can be effective in increasing alignment between people's interpretation and the intended meaning of words in a predefined categorization (e.g. [110]). However, recent research on expressions of future (including aleatory) uncertainty indicates that including numeric information may not reduce the extremity effect [114].

### 6.1.4. Expressions vi and vii: listing possibilities and mentioning uncertainty

Another form of verbal uncertainty communication involves simply listing various possibilities or mentioning uncertainty through, for example, caveats. Such general statements about scientific uncertainty can cover both uncertainty about the object in question (e.g. fact, quantity and model) as well as the strength or quality of underlying evidence or science. Corbett & Durfee [115] examined the

effects of communicating uncertainty in news stories about climate science. They focused on mentioning disagreement or controversy among scientists through a lack of context in which to interpret the meaning of the findings. Their results showed that news stories that included expert disagreement increased people's perceptions of uncertainty about climate science, whereas news stories that included context decreased perceptions of uncertainty.

These findings appear to be quite general. Indeed, when quantifiable facts, such as the scientific consensus on climate change and vaccine safety, are contested implicitly through caveats or expert disagreement, perceived uncertainty about the science typically increases [116–118]. Yet, exceptions do exist. For example, research in the context of emerging technologies found that verbally highlighting broad scientific uncertainty about nanotechnology did not meaningfully change more general beliefs about the uncertainty of scientific evidence [119].

## 6.2. The effect of communicating uncertainty on emotions and affective reactions

A large literature in psychology illustrates that people process uncertain information in two qualitatively different ways: Slovic *et al.* [120], for example, differentiate between processing risk 'as analysis' versus risk 'as feelings'. Dual-process theories in psychology commonly describe two systems of thinking, with one system being more 'analytic', following rules of logic, probability and deliberation, whereas the other is more associative, fast, intuitive and affective [121–125]. Although the functional, anatomical and psychological differences between these two systems have not gone without criticism (e.g. [126]), it is important to consider the impact of uncertainty communication on people's affective reactions, given that emotional responses are often dominant in processing risk information [120,124,127,128]. In addition, people's emotions and affective reactions exert an important influence on decision-making [129–131]. Comparatively, there is less work on affect than on cognition in the context of epistemic uncertainty communication specifically. We could not find any prior studies that have explored the emotional effects of full or summary probability distributions (expressions i and ii) or predefined categories and qualifying statements (expressions iv and v).

### 6.2.1. Expression iii: ranges

The effects of communicating uncertainty on people's emotions have been studied to some extent in the medical domain. For example, in a number of focus groups with 48 American adults, Han *et al.* [83] found that numerical risk ranges (e.g. 'your risk of colon cancer is somewhere between 5% and 13%') elicited more worry from most people than a point estimate ('…9%'), but for others they noted the opposite effect or indifference between formats. In a series of follow-up studies, Han *et al.* [84] presented people with either a point estimate or a numerical range without a point estimate, and did this either in text or in a visual format (a bar graph from 0 to 100%). The results indicated that compared to no uncertainty, presenting uncertainty as a numerical range in text increased people's reported levels of worry about developing colon cancer and perceived risk, but in a visual format uncertainty decreased worry and risk. A follow-up experiment comparing a range being presented in text or in combined visual-text formats did not yield any significant differences in worry and perceived risk between the text and visual formats. This research suggests that epistemic uncertainty might influence emotional responses, but that the exact form of communication matters and that more research is needed to gain a better understanding of how people's affective reactions are shaped by the communication of scientific uncertainty. This work also found some potential influences of individual differences. They found that people high in dispositional optimism reported less cancer-related worry after being shown a cancer risk estimate with uncertainty communicated as a range compared to just the point estimate.

### 6.2.2. Expression vi: listing possibilities

Van Dijk & Zeelenberg [132] examined the impact of communicating uncertainty as a list of possibilities about facts on people's affective responses. In the first of two experiments, they asked people to imagine participating in a game in which they won a prize: a prize that was either certain (two conditions: definitely winning a CD versus definitely a dinner) or a prize that was uncertain (they were told to imagine that they won either a CD or a dinner). Participants who were presented with an uncertain prize reported to be less happy, less satisfied and felt less good than students in either of the certain prize conditions. In the second experiment, participants were asked to imagine winning one of two

certain prizes or the uncertain prize in a lottery, but subsequently to imagine they had lost their lottery ticket and would not be able to claim the prize. In this case, participants who were presented with an uncertain prize reported feeling less unpleasant, disappointed and bad than participants in both certain prize conditions. This research suggests that communicating uncertainty as a list of possibilities about a fact (which prize people won) can either dampen or heighten people's emotional responses to this fact.

### 6.2.3. Expression vii: mentioning uncertainty as caveats

Jensen *et al*. [133–135] studied the effects of communicating scientific uncertainty through verbal statements addressing caveats or limitations (versus a generic statement presenting low uncertainty) about cancer research in news stories. They examined people's affective responses, specifically cancer fatalism, 'a disposition defined by feelings of angst and nihilism' [135] and nutritional backlash, which is described as a range of negative feelings such as fear, guilt, worry and anger about dietary recommendations [134,135]. The results across these studies were inconsistent: their first research [134] found a decrease in cancer fatalism and a marginally significant decrease in nutritional backlash for people who had read news stories that contained higher levels of scientific uncertainty. Follow-up research [135], however, did not find an effect of uncertainty on cancer fatalism and nutritional backlash, but did find a decrease in fatalism and nutritional backlash for people who had read news stories that depicted disclosure (in which uncertainty statements were attributed to the key researchers covered in the news story) rather than expert disagreement.

Overall, it appears that more research is needed in order to gain a better understanding of the effects of epistemic uncertainty about science, facts and numbers on people's affective and emotional reactions. Furthermore, research is often unclear as to whether 'emotion' is meant to tap into a fast evaluative judgement of a stimulus (affect), or whether discrete emotions are of interest (e.g. fear, worry), and how these are or should be measured (e.g. a physiological response to uncertainty versus self-report). In short, the limited research described above reports inconsistent results: it appears that communicating uncertainty can have an impact on people's emotions, but that the nature of the impact might be dependent on how emotions are defined and measured as well as how uncertainty interacts with other characteristics of the communication.

## 6.3. The effect of communicating uncertainty on trust and credibility

People's relationship with trust is asymmetrical: it takes a long time to forge but can be destroyed in an instant [136]. At a generic level, there are some near-universal aspects of human social cognition that assist people in determining whom and what information to trust. Two of these basic dimensions include 'competence' and 'warmth' [137]. Affect and cognition fuse together here in establishing trust. In order to be perceived as credible, both 'cold' expertise is required (knowledgeability) as well as a perceived motivation to be sincere and truthful (warmth), that is, a feeling of trust [138,139].

Although scientists and researchers generally score high on perceived competence and expertise, they are often perceived to lack warmth, a key component of which is 'trustworthiness' [140]. More generally, a decline in public trust of regulators and industry has been observed [141]. To remedy this relationship, Fiske & Dupree [140] suggest that rather than trying to persuade, warmth can be gained by openly discussing and sharing scientific information. Yet, whether greater transparency in the communication of uncertainty will enhance credibility and public trust remains an open empirical question [142]. On the one hand, presenting information as certain (when it is not) is misleading and can damage and undermine public trust. Thus, emphasizing uncertainty may help signal transparency and honesty. On the other hand, explicitly conveying scientific uncertainty may be used as a tool to politicize science [143], or to undermine the perceived competence of the communicator as people tend to use precision as a cue for judging expertise [144]. Research into how the communication of scientific uncertainty impacts trust and credibility is very sparse, and we found examples from only three forms of expression of uncertainty.

### 6.3.1. Expression ii and iii: summaries of distributions and ranges

Early research by Johnson & Slovic [145] found that a discussion of range estimates in the context of environmental health risks signalled government honesty for a majority of their sample. However, in a similar follow-up study, results were more mixed where equal numbers agreed and disagreed about

perceived honesty [99]. Moreover, about a third felt that a range discussion made the government seem less competent, and about 40% of the sample did not think the government was telling the truth. A later study, including a re-analysis of this earlier research by Johnson [146], suggests that communicating uncertainty through range estimates revealed mixed results, signalling honesty and competence for sizeable portions of participants across studies (25–49%) but also dishonesty and incompetence for non-negligible minorities (8–20%).

Similarly, in a series of small focus groups, Schapira *et al.* [82] explored responses to a line graph that visualized an estimate of breast cancer mortality relative risk reduction of mammography screening. The representation of uncertainty through a confidence interval led to confusion and decreased trust in the information for women in the less-educated groups. Women in the higher-educated groups were 'more accepting' of such scientific uncertainty and in general indicated that the confidence interval should be presented to patients, so that all information is available for decision-making. Similar results have been found for individual differences in numeracy [147].

Yet Lipkus *et al.* [148] found no effect of uncertainty communication: in a pre-post design, participants found a point estimate just as credible and trustworthy as a range of risks about breast cancer. In the context of individualized colorectal cancer risk, Han and colleagues [84] also found no main effect of uncertainty (confidence interval versus point estimate) or format (visual versus text) on perceived credibility, which included a measure of trust. Similarly, Kuhn [149], using a relatively small student sample, evaluated four communication formats (point estimate, verbal uncertainty, numerical range and biased range) across five (aleatory and epistemic) environmental hazards and found no main effect of uncertainty on trust in science or government.

By contrast, van der Bles *et al.* [150] distinguish between trust in the numbers and trust in the messenger, as they find that introducing uncertainty about a range of numbers (e.g. the number of tigers in India, unemployment in the UK, global temperature rise) reduces trust in the number but not necessarily in the messenger. This also varies by format (or precision), with much greater reduction in trust in the number for verbal/less precise expressions of uncertainty (e.g. 'somewhat higher or lower') than for numerical expressions (e.g. point estimate with range).

### 6.3.2. Expression vii: mentioning uncertainty

Some research has studied the effect of mentioning uncertainty, typically a combination of uncertainty about an object and uncertainty about the underlying quality or strength of evidence, on trust and credibility. Jensen *et al.* studied the effect of uncertainty in news stories on trust in the medical profession [134], and on trust in and expertise of journalists and scientists [133]. Jensen [133] found that people viewed both scientists and journalists as more trustworthy when they had read news stories presenting higher levels of uncertainty that were attributed to the primary scientists (disclosure, compared to expert disagreement). There were no effects on the perceived expertise of scientists and journalists. Jensen *et al.* [134] found no effect of uncertainty on people's trust in the medical profession, but they did find that people expressed increased trust when they read articles in which the uncertainty was mentioned by unaffiliated researchers (expert disagreement) rather than statements from scientists whose research was covered in the article.

Yet, other work shows that when uncertainty is introduced through a conflict between experts (e.g. half of experts say that studies show a link between aluminium and Alzheimer's disease whereas the other half deny such a link), source credibility and trustworthiness tend to decline compared to when all experts agree there is uncertainty [151]. Similarly, Löfstedt [152] observes that in the context of the Swedish acrylamide scare, public disagreements between epidemiologists and toxicologists over the link between acrylamide and cancer led to public distrust in scientists. To some extent, uncertainty created through divergent scientific perspectives may be dependent on context. For example, Jensen & Hurley [153] found that uncertainty about the health effects of dioxin (a possible carcinogen) increased the credibility and trustworthiness of scientists, whereas the opposite pattern was found for conflicting stories about wolf reintroduction in the USA.

Wiedemann & Schütz [154] evaluated the effects of disclosing uncertainty in the context of health risks from exposure to electromagnetic fields. In one experimental condition, scientists verbally acknowledged that 'substantial uncertainties exist as to whether current protection from electrosmog is sufficient'. Such verbal disclosure of uncertainty did not undermine public trust in health protection. On the other hand, verbal uncertainty expressed through caveats and limitations (e.g. 'perhaps', 'maybe', 'possibly') in economic news has been associated with lower public confidence in the economy [155,156].

In sum, until more research is conducted, it is difficult to make firm conclusions about these mixed findings across domains about the way and extent to which communicating uncertainty affects the perceived credibility of and trust in both the message and the communicator.

## 6.4. The effect of communicating uncertainty on behaviour and decision-making

For many communicators, the most important aspect of communicating uncertainty is its effect on people's behaviour and decision-making. This is particularly relevant in the context of decision-making under uncertainty, for example, in medical or policy-relevant contexts. Although we recognize the large literature on ambiguity aversion [94,157], to our knowledge there has been no systematic empirical investigation of the effect of *epistemic* uncertainty communication on people's behaviour and decision-making. Nonetheless, somewhat scattered results do exist and there have been some broadly relevant studies. For example, Tversky & Shafir [158] illustrate how the introduction of uncertainty can influence decision-making. In a hypothetical decision-making scenario involving a vacation package, students who were uncertain about a verifiable fact (e.g. whether or not they passed a qualifying exam), were much less likely to book a trip (compared to students who were certain), and even willing to pay a small sum of money to postpone their decision. Although hypothetical, these results suggest that communicating uncertainty may lead people to postpone their decision-making in some contexts.

By contrast, much more research exists when it comes to *aleatory* uncertainty about the future. For example, Morton *et al.* [159] find that describing the uncertainty around future climate change impacts (a point estimate versus point estimate with a range) influences people's intention to act pro-environmentally, depending on how the impacts are framed (positive versus negative). As another example, Joslyn & LeClerc [160] studied the effect of uncertainty in weather forecasts on (hypothetical) decision-making by asking participants to assume the role of a road maintenance company considering whether to pay to salt the roads. They found that uncertainty communication increased decision quality and trust in the forecast. In particular, participants in the uncertainty condition took appropriate precautionary action and withheld unnecessary action more often than participants in the deterministic forecast condition. Similarly, Driver *et al.* [161] conducted experiments to investigate whether people can make better investment choices if presented with visual rather than verbal descriptions of investment uncertainty. Specifically, the authors found that representing financial disclosure (a risk-return indicator) pictorially helped people better rank funds according to their risk and return profile and better assess their suitability when making financial decisions.

It therefore seems clear that uncertainty has the potential to influence decision-making across domains, from medical decision-making to consumer and environmental behaviour, but more systematic research is needed to investigate explicitly how epistemic uncertainty influences human behaviour and decision-making. As Raftery recommended in the context of probabilistic forecasting [162], this will involve research that bears in mind the diversity and types of audiences; and the psychological impact that the presentation of uncertainty has on its audience.

## 6.5. Conclusions about the psychological effects of communicating uncertainty

Although the scattered evidence available suggests that communicating direct epistemic uncertainty does affect people's cognition, emotion, trust, and behaviour and decision-making, little has been done within a systematic framework—identifying the aspects of the communication that are being manipulated and therefore delineating their precise effects. Even within a framework, such as the one we have suggested, being systematic is very difficult: formats are inevitably correlated with the *precision* of the expression, with numbers having the most potential to convey the most precise information and verbal expressions the least, and so any attempt to vary format will often also vary the expression. The content is essentially different.

Audience reactions to different *magnitudes* of uncertainty would also seem an important cognitive outcome for communication of epistemic uncertainty. However, there appear to be few empirical studies investigating this phenomenon: initial work in the context of a (fictitious) news article [150] showed no change in the public's trust in either the number communicated or the communicators of the uncertainty when different magnitudes of uncertainty were communicated as a numeric range. It may well be that uncertainty needs to be put into context for a non-specialist audience in order for its magnitude to be of any relevance. Without specialist knowledge of a subject, it may be impossible for an audience to judge whether a given magnitude of uncertainty is important or not. It could also be

that an audience that is not basing a decision on a particular fact, number or hypothesis does not have enough vested interest to be discriminatory and therefore may have a more binary approach to judging whether information is 'certain or not'.

Considering the literature on the psychological effects of different *expressions* of uncertainty, however, suggests several interesting preliminary findings. We can be relatively confident that there is substantial individual variability in how people understand uncertainty through verbal qualifying statements or predefined distributions. This can lead to a gap between how people understand the communication of uncertainty and the actual intention of the communicator. Accordingly, the appropriate use of graphical visualizations and numerical uncertainty may aid in ensuring a correct understanding and comprehension of uncertainty. Yet, although people's understanding of numbers seems more consistent, there is still scope for variability in interpretation, for example, when it comes to different interpretations of distributions underlying numerical ranges. This might lead to different psychological and behavioural outcomes, depending on whether people interpret the lower, midpoint, or upper values to be the more likely 'true' value.

The limited research that has investigated the effects of epistemic uncertainty communication on emotions has found mixed results, which suggests that (epistemic) uncertainty does not *always* have a negative effect on people's affective states. This is an important preliminary conclusion, given the often-cited concern that people are generally averse toward any kind of uncertainty [92,159]. What's more, uncertainty about the future can interfere with people's basic psychological needs for control and predictability [163], whereas epistemic uncertainty about the past and present may not always be subject to the same concerns. People also make forecasting errors about how much they (dis)like uncertainty [164]; and so more research is needed to evaluate how the presence of uncertainty about facts and numbers interacts with people's emotional and affective dispositions toward the issue.

Similarly, several smaller scale studies have revealed conflicting information about whether explicitly acknowledging scientific uncertainty—either numerically or verbally—enhances or undermines the extent to which people trust or find the information credible. It also remains unclear how the effect of uncertainty on trust interacts with communication format or characteristics of the communicator. The preliminary conclusion that explicitly acknowledging uncertainty does not always lead to an inherent decrease in public trust or credibility, though, is worth noting, and suggests an important potential avenue for future research.

Finally, if it is the case that some audiences are insensitive to changes in the magnitude of the uncertainty being communicated, then it could be said that the communication is not delivering the intended message. This could have important consequences for decisions made by audiences based on the communication (as outlined in box 1). It therefore seems critical that research on the communication of uncertainty explicitly investigates the effect of manipulating the magnitude of uncertainty. For work with a directly practical outcome (such as the transmission of an important message to decision-makers), this will be a useful indication of the success of the message. For work with a research focus, it will help elucidate different audiences' relationship with uncertainty, and how context and different forms of expression affect the perceptions of magnitude.

# 7. Case studies in communicating uncertainty

Although different fields and organizations have taken different approaches to the problem of communicating uncertainty, our framework has revealed considerable commonalities. Here we examine two important areas in more detail: official economic statistics and climate change. The latter domain, in particular, is often associated with making predictions about an uncertain future, but we focus on the way in which it expresses epistemic uncertainty. We illustrate how both domains fit into the structure established above, and hopefully provide insights for others wanting to communicate uncertainty.

## 7.1. Case study 1: official economic statistics

### 7.1.1. What epistemic uncertainties are there?

*Objects of uncertainty:* Measurable historical economic variables such as real gross domestic product (GDP), inflation and employment.

*Sources of uncertainty:* Statistical offices typically provide estimates of economic variables using surveys that are subject to both sampling and non-sampling errors. Manski [165] re-interprets these

sources of data uncertainty as 'transitory' and 'permanent'. 'Transitory' statistical uncertainty arises because data collection takes time, with 'early' data releases revised over time as new information arrives. 'Permanent' statistical uncertainty does not diminish over time and arises due to the limited data available (e.g. sampling uncertainty due to a finite sample), and/or data quality (e.g. survey non-response).

Statistical offices are under pressure, by policymakers and other users of the data, to produce timely estimates. But this induces a trade-off with data reliability, since timely data rely more strongly on incomplete surveys. For example, until July 2018, the UK Office for National Statistics (ONS) produced a 'preliminary' estimate of GDP based on about 44% of the sample around 27 days after the end of the reference quarter. But since July 2018, they provide their 'first' estimate of GDP around 40 days after the end of the month/quarter, based on about 60% of the sample. This 'first' estimate of GDP should therefore be expected to be revised as more sample data become available. But the ONS hopes that its new publication model will deliver less uncertain estimates of economic growth, given that, because of pushing publication back by two weeks, they are based on a higher sampling percentage.

*Levels of uncertainty:* Non-sampling errors generally give rise to caveats about the quality of the underlying evidence (indirect uncertainty), while sampling errors may be quantified as direct margins of error on the quantity of interest (direct uncertainty).

### 7.1.2. In what form are the uncertainties communicated?

*Expressions and formats of uncertainty:* table 2 shows, using examples of UK and US practice for GDP, inflation and employment, how national statistical offices provide an incomplete expression of the uncertainty associated with their data according to our proposed scale for 'numbers'. They use verbal, numeric and (much less commonly) visual formats of communication.

Table 2 distinguishes if and how statistical offices communicate data uncertainty in the 'headline' press releases, that typically form the focus of the media when disseminating the data release more widely, from what is said in 'smaller print' (including lower down an often long press release) and/or perhaps in separate technical reports or online. As table 2 shows, data estimates for these three economic variables are all reported as point estimates in the headline data release, even though textual and—perhaps more so in the USA than the UK—quantitative acknowledgements of the uncertainties associated with the data do follow elsewhere, but arguably with limited prominence. For example, while the BLS do publish, both numerically and visually, margins of error on US unemployment estimates these are not found in the headline releases. 90% confidence interval graphs are found only on a pulldown menu on a webpage accessible from the press release.

Manski [170] has similarly documented the 'incredible certitude' in official economic statistics, given that they are commonly reported as point estimates without adequate attention paid to uncertainties.

For some economic variables, such as employment, it is easier to quantify sampling uncertainties; but for others, such as GDP, it is understandably more challenging. For example, to quote the ONS: 'The estimate of GDP … is currently constructed from a wide variety of data sources, some of which are not based on random samples or do not have published sampling and non-sampling errors available. As such it is very difficult to measure both error aspects and their impact on GDP. While development work continues in this area, like all other G7 national statistical institutes, we don't publish a measure of the sampling error or non-sampling error associated with GDP' [171].

This rather limited communication runs counter to the longstanding awareness of economic statisticians of the importance of quantifying and communicating the uncertainties associated with such economic statistics; Morgenstern [172] assessed the accuracy of economic data and argued for the provision of error estimates in official statistics 60 years ago. But, as Manski [165] concludes in a rare paper on this topic in a leading academic economics journal, economic data continue to be communicated often with little upfront indication of their uncertainties.

While table 2 provides evidence of limited quantification and communication of data uncertainty by statistical offices, independent users of data have provided their own uncertainty estimates. A prominent example is the Bank of England's 'fan chart' for GDP growth, as shown in figure 6. As well as indicating the uncertainty associated with their forecasts, this figure provides a quantitative visual indication of the uncertainty associated with historical and recent official ONS estimates of GDP; see [174] for further analysis. More generally, many statistical offices and central banks now publish and maintain 'real-time' macroeconomic databases (e.g. [166,175]) to reflect the fact that many economic statistics, like GDP, are revised over time. These real-time data are commonly used by researchers in

**Table 2.** Cross-country and cross-variable measurement and communication of uncertainty about economic statistics.

| | economic variables | | |
| --- | --- | --- | --- |
| | GDP | inflation | employment |
| economic concept: | the value of all final goods and services produced within a country in a given period of time | the rate of increase in prices for goods and services | total number of people employed in a country |
| geographic source: | United Kingdom, Office for National Statistics (ONS) | | |
| economic statistics: | gross domestic product, chained volume measure | consumer Prices Index (CPI) 12-month rate | number of people aged 16–64 at work |
| source of uncertainty: | revisions from updated sample information, correction of errors, benchmarking, updated base period for constant price estimates and methodological changes [166]. | price index weighting methodology and survey design. | sampling error and revisions. |
| uncertainty communication in headline press release: | *v. single number with qualifying verbal statements* 'UK gross domestic product was estimated to have increased by 0.4% in Quarter 3 2017, a similar rate of growth to the previous two quarters'. | *viii. no mention of uncertainty* 'The Consumer Prices Index including owner occupiers' housing costs (CPIH) 12-month inflation rate was 2.8% in October 2017, unchanged from September 2017'. | *viii. no mention of uncertainty* 'There were 32.06 million people in work, 14 000 fewer than for April to June 2017 but 279 000 more than for a year earlier'. |
| uncertainty communication in supporting documentation: | *v. single number with qualifying verbal statements* The estimate is subject to revision as more data become available, but the revisions are typically small between the preliminary and third estimates of GDP, with no upward or downward bias to these revisions'. (ONS website). Revisions are not quantified; but references to real-time data sources and studies are provided in the ONS website. | *v. single number with qualifying verbal statements* The release includes a section on 'Quality and methodology'. But this does not explicitly mention uncertainty due to sampling and/or non-sampling errors. | *iii. a range or an order-of-magnitude assessment* The release includes a section on 'Quality and methodology'. This discusses and provides uncertainty estimates such as 95% confidence intervals due to sampling errors [167]. |

(*Continued.*)

**Table 2.** (*Continued.*)

| | economic variables | | |
|---|---|---|---|
| | GDP | inflation | employment |
| geographic source: | United States, Bureau of Economic Analysis (BEA) and Bureau of Labor Statistics (BLS) | | |
| economic statistics: | real gross domestic product | per cent changes in Consumer Price Index for All Urban Consumers | total non-farm payroll employment |
| source of uncertainty: | revisions from updated sample information, updated base period for constant price estimates and methodological changes [168]. | sampling error and revisions | sampling error and revisions |
| uncertainty communication in headline press release: | *viii. no mention of uncertainty* 'Real gross domestic product increased at an annual rate of 3.0 percent in the third quarter of 2017'. | *viii. no mention of uncertainty* 'The Consumer Price Index for All Urban Consumers (CPI-U) rose 0.1 percent in October on a seasonally adjusted basis, the U.S. Bureau of Labor Statistic reported today'. | *viii. no mention of uncertainty* 'Total nonfarm payroll employment rose by 228000 in November (2017), and the unemployment rate was unchanged at 4.1 percent, the U.S. Bureau of Labor Statistics reported today'. |
| uncertainty communication in supporting documentation: | *v. single number with qualifying verbal statements* 'The Bureau emphasized that the third-quarter advance estimate released today is based on source data that are incomplete or subject to further revision by the source agency (see "Source Data for the Advance Estimate" on page 2)'. Average data revision estimates are also provided in this press release, although these are not used to form confidence intervals. | *iii. a range or an order-of-magnitude assessment* Lower down in their press release published in their website as the data is released the BLS discuss and quantify sampling errors numerically. | *iii. a range or an order-of-magnitude assessment* In separate documentation to the press release the BLS provide 90% confidence intervals [169]. These are also plotted visually at https://www.bls.gov/charts/employment-situation/otm-employment-change-by-industry-confidence-intervals.htm. |

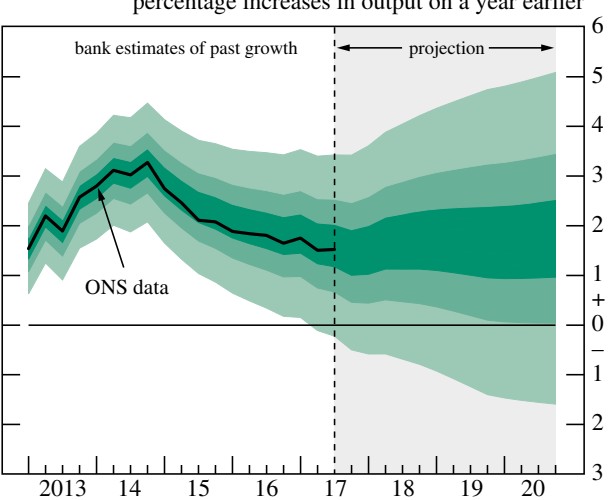

**Figure 6.** Bank of England's fan chart for GDP growth (from the November 2017 'Inflation Report' [173]). In their notes to this chart the Bank write: 'The fan chart depicts the probability of various outcomes for GDP growth... To the left of the vertical dashed line, the distribution reflects the likelihood of revisions to the data over the past; to the right, it reflects uncertainty over the evolution of GDP growth in the future...The fan chart is constructed so that outturns are also expected to lie within each pair of the lighter green areas on 30 occasions. In any particular quarter of the forecast period, GDP growth is therefore expected to lie somewhere within the fan on 90 out of 100 occasions'.

macroeconomics and finance again to acknowledge the data uncertainty. Indeed, the Bank of England form their view about data uncertainty, in part, based on the revisions properties of official GDP data.

### 7.1.3. What are the psychological effects of uncertainty communication?

Although Gilbert [176] and Clements & Galvao [177] show that US equity market participants are aware of and react to BEA's GDP data revisions, there seems to have been no research on if and/or how users of economic statistics interpret the estimates as being measured subject to uncertainties, or how different users might react to uncertainties communicated to them directly by the statistical office. The importance of undertaking such research is reinforced by Wallis [178] who shows, in the specific context of the Bank of England's fan chart (seen above), how the same probability distribution can be drawn in different ways. Moreover, the Bank itself presents the fan chart above both using the 'wide bands' shown in figure 6 and 'narrow bands'. It is not known how users react to these alternative representations of uncertainty. But the importance of undertaking such research is apparent from the lively online discussion about the usefulness, or otherwise, of the Bank of England's fan charts, such as the somewhat sarcastic discussion on the Financial Times' blog on 'Save the (in)famous fan charts!' [179].

### 7.1.4. Conclusions

According to the Code of Practice for official statistics for England and Wales [180], 'trustworthiness' forms the first of the three pillars of official statistics, and the Code emphasizes the need for clear communication of uncertainties. Nevertheless, the analysis above shows rather limited and not-so-prominent communication of uncertainty about official statistics, in spite of the strong example set by the Bank of England. Direct expressions of uncertainty may be provided, but the strength of the underlying evidence (indirect uncertainty) is communicated through caveats which may or may not be read and understood by their audiences.

As administrative, microdata data are increasingly exploited to supplement or indeed replace traditional surveys, the assessment of uncertainty will face a range of new difficulties. While administrative data are not based on a survey, they also can have inadequacies due to quality, coverage and relevance. The resulting uncertainty is difficult to quantify: Manski [28] provides a methodology to quantify non-sampling errors due to survey non-response, that derives interval estimates with no assumptions made about the values of the missing data, and this promises to become a vital area of research.

The effects of uncertainty communication on cognition, emotion, trust and behaviour in economics statistics appear not to have been studied.

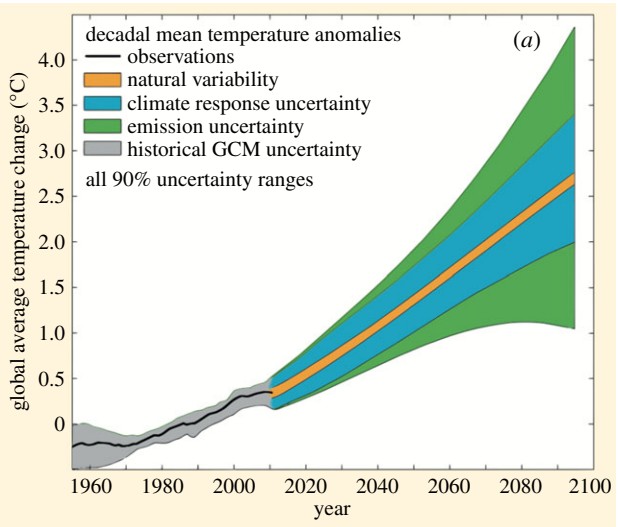

**Figure 7.** Global average temperature change (IPCC). Adapted from the IPCC [38]. Diagram showing the relative importance of different uncertainties, and their evolution in time. (*a*) Decadal mean surface temperature change (°C) from the historical record (black line), with climate model estimates of uncertainty for historical period (grey), along with future climate projections and uncertainty. Values are normalized by means from 1961 to 1980. The given uncertainty range of 90% means that the temperature is estimated to be in that range, with a probability of 90%.

## 7.2. Case study 2: intergovernmental panel on climate change (IPCC)

### 7.2.1. What epistemic uncertainties are there?

*Objects of uncertainty:* Measurable historical changes in a variety of climate system variables, such as average global surface temperature and sea-level rise.

   *Sources of uncertainty:* In the context of climate change, epistemic uncertainty often refers to uncertainty about the values of parameters in scientific models or structural uncertainty about the underlying model and its ability to accurately represent the climate system. In addition, uncertainty in model-based estimations of fluctuations in historical climate data may arise due to differences in the types of environmental data that are used as input for climate models and, for IPCC models that look at trends in the earth's surface temperature over the last 150 years, uncertainty can further arise from data gaps (proxy data) and a variety of instrumental and measurement errors [38]. Figure 7 displays uncertainty in a historical series using a simple shading of a 90% interval.

   *Levels of uncertainty:* Limitations in scientific understanding and lack of confidence in some models are expressed as caveats (indirect uncertainty), while errors which can be confidently modelled give rise to direct probabilities on the quantity of interest (absolute, direct uncertainties). Relative direct uncertainties, such as *p*-values, are rarely if ever used.

### 7.2.2. In what form are the uncertainties communicated?

*Expressions and formats of uncertainty:* The IPCC has a relatively long history of exploring how to effectively express different forms of uncertainty in their reports but has only recently started to begin incorporating insights from behavioural science. At present, uncertainty in the IPCC assessments is communicated using two metrics shown in figure 8. Firstly, quantified measures of direct (absolute) uncertainty are expressed in verbal and probabilistic terms based on statistical analyses of observations, models or expert judgement, corresponding to a pre-defined categorization—expression iv in our framework. Secondly, indirect (underlying) uncertainties are expressed through a qualitative expression of confidence in the validity of a finding based on the type, amount, quality and consistency of evidence (which can include theory, models and expert judgement) [38, p. 36].

   Likelihood provides calibrated language (panel a) for describing quantified uncertainty for a single event, a climate parameter, an observed trend or projected future change. Importantly, the likelihood table is not preferred when a full probability distribution is available instead. Confidence level (panel b) is based on the scientific evidence (robust, medium, limited) and working group agreement (high,

(a)

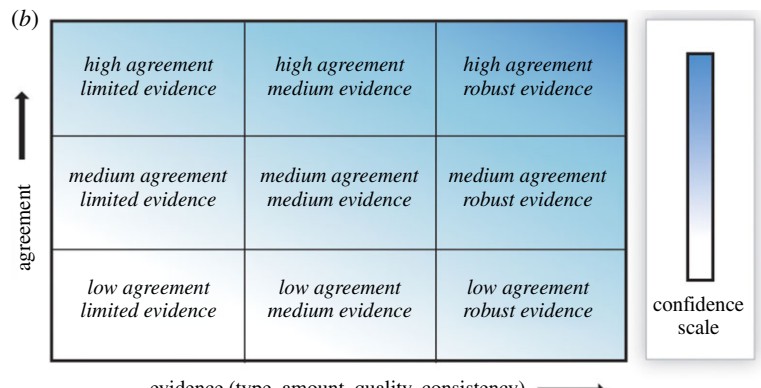

| term* | likelihood of the outcome |
|---|---|
| *virtually certain* | 99–100% probability |
| *very likely* | 99–100% probability |
| *likely* | 66–100% probability |
| *about as likely as not* | 33–66% probability |
| *unlikely* | 0–33% probability |
| *very unlikely* | 0–10% probability |
| *exceptionally unlikely* | 0–1% probability |

*additional terms (*extremely likely*: 95–100% probability, *more likely than not*: >50–100% probability, and *extremely unlikely*: 0–5% probability) may also be used when appropriate.

(b)

| high agreement limited evidence | high agreement medium evidence | high agreement robust evidence |
|---|---|---|
| medium agreement limited evidence | medium agreement medium evidence | medium agreement robust evidence |
| low agreement limited evidence | low agreement medium evidence | low agreement robust evidence |

agreement →

confidence scale

evidence (type, amount, quality, consistency) ⟶

**Figure 8.** The IPCC's two metrics for communicating the degree of certainty in their findings: (*a*) expressions of uncertainty and (*b*) confidence in the science [38, p. 36].

medium, low). Robustness of evidence is measured by the degree of consistent and independent (multiple) lines of high quality inquiry.

To illustrate the use of these tables with a written example from the fifth IPCC report: 'It is *certain* that global mean surface temperature has increased since the late 19th century. [...] For average annual Northern Hemisphere temperatures, the period 1983–2012 was *very likely* the warmest 30-year period of the last 800 years (*high confidence*) and *likely* the warmest 30-year period of the last 1400 years (*medium confidence*)' [38, p. 38].

### 7.2.3. What are the psychological effects of uncertainty communication?

In the first three reports, the IPCC generated confusion by not systematically communicating relevant uncertainty properly [181,182]. The IPCC tried to resolve this problem by verbally communicating uncertainty using seven verbal quantifiers (e.g. 'very likely', figure 8) with the probability translation (likelihood) table supplied in an appendix. Part of the reason for this is that verbal quantifiers can help bridge disagreements between authors, especially when uncertainty cannot be condensed into a single number. Yet, much research in psychology has shown that verbal quantifiers of uncertainty create an 'illusion of communication' because different people interpret such terms differently across different contexts [183]. In fact, as has already been mentioned, some research has been conducted on people's interpretation of the IPCC's uncertainty communication strategies. In a multi-national study, with 25 samples from 24 countries in 17 languages, Budescu *et al.* [110] found that people's interpretation of these verbal quantifiers were systematically inconsistent with the IPCC's intended guidelines. Specifically, the general pattern was regressive so that people underestimated high probabilities and overestimated low probabilities. Moreover, average consistency with the IPCC guidelines appeared to be higher in countries that express greater concern about climate change [110]. Subsequent research has found that a 'dual-approach' combining the verbal quantifiers with corresponding (numeric) probability ranges (e.g. 'very likely, greater than 90%') creates much stronger cross-sample homogeneity in people's interpretation of the terms in a way that is more consistent with the IPCC guidelines [110,181,184].

### 7.2.4. Conclusions

The IPCC's approach to communicating uncertainty has been developed over a long period in order to bring some standardization to a huge and diverse team, producing high-profile documents in a deeply contested area of science. Their approach to direct uncertainty through verbal descriptors calibrated with probability ranges has been subject to considerable evaluation, although their expression of confidence in the underlying science (indirect uncertainty) has not been empirically researched. It is notable that so many areas, from climate to health to education, have arrived at rather similar ways of communicating uncertainty through both direct and indirect expressions, and these common features form the basis for our structure.

## 7.3. Conclusion

Both these case studies consider potentially contested topics, where communicators may wish to portray uncertainty in different ways in order to produce different psychological effects on the reader. Whereas the official sources of information in both cases make attempts to communicate uncertainty 'neutrally', secondary users of their information (such as media organizations) often do not. In the case of official economic statistics, for example, media coverage overwhelmingly avoids mention of epistemic uncertainty. This leads to inevitable concern and discussion when revisions to statistics occur (e.g. [185]), which itself reveals that readers thought the estimates more certain than they actually were—and their comprehension, emotions and decision-making were all likely to have been altered by the change in uncertainty communication caused by a revision. In the case of climate change, there has been much discussion over whether over-emphasis or under-emphasis of uncertainty in communications has created changes in the comprehension, emotions or behaviour of different audiences—although the lack of empirical evidence specifically on this topic makes those opinions pure hypothetical, if entirely plausible [186].

## 8. Discussion and conclusion

There are many ways in which we are uncertain about the world around us, without even considering unavoidable uncertainty about the future. Not acknowledging or adequately communicating these uncertainties, leading to unwarranted degrees of weight being put on certain pieces of evidence in the process of decision-making, can have disastrous consequences (as demonstrated by the example in box 1). In this review, we therefore aimed to summarize what is known about epistemic uncertainty communication, the range of communication methods currently used, and their psychological effects on people's cognition, affect, trust, behaviour and decision-making. This work is intended as the first and necessary step towards the goal of an empirically based guidance for everyone who works with epistemic uncertainty on the forms of communication that suit their object of uncertainty, their audience and their aims.

Based on an interdisciplinary approach, we have developed an overarching framework that clarifies the components that make up the umbrella term 'uncertainty' and those that comprise the process of communication, affecting an audience's reaction to uncertainty communication (figure 1). This provides a structure both for understanding what is done and what might be investigated in the future. The framework comprises several novel and hopefully useful elements:

— the identification of three *objects* of uncertainty: categorical (facts), continuous (numbers) and hypothetical (theories or scientific models that describe the world) (§3.1)
— the distinction between two *levels* of uncertainty: direct uncertainty about specific facts, numbers and science (both absolute and relative), and indirect uncertainty: the quality of our underlying knowledge (§3.3)
— a list of alternative expressions for direct uncertainty based on an analysis of practice across many disciplines (§4.1 and figure 2).

We show how this framework could be useful to those trying to understand the communication of uncertainty by using it as the underlying basis in a review of the empirical evidence of the psychological effects of the different expressions of (direct) uncertainty on different audiences (§6), and to deconstruct how uncertainty is communicated in two example fields (official economic statistics and climate change).

Our review of the empirical evidence of the effects of epistemic uncertainty communication showed that different uncertainties, and different expressions of those uncertainties, can have varied psychological effects on their audiences. Most research has focused on cognitive effects of uncertainty communication, showing that there is considerable variability in people's interpretation of various

**Box 5(a).** Advice to communicators of uncertainty.

When communicating evidence, consider:

— What do you have uncertainty about? Do you have uncertainty about the *underlying hypothesis* behind your evidence, and/or about the *specific numbers* involved, and/or about *categorical facts* that you want to claim? For instance, you may have different levels of uncertainty about the fact that the mean global temperature has risen since 1850; that the temperature rise has been approximately 1.8°, or that the temperature rise has been the result of the 'greenhouse effect'.

— Why is there uncertainty? Is it because of unavoidable *natural variation*, because of the *difficulties of measuring*, because of *limited knowledge* about the underlying processes or because there is *disagreement* between experts? Thinking about this may help you identify more objects you are uncertain about, and about how your audience might perceive the uncertainty.

— For each object that you are uncertain about, and for each source of uncertainty, do you have both *direct* (specifically about that aspect) and *indirect* (quality of evidence) levels of uncertainty? You will probably want to, or have to, communicate them separately.

When you have identified all of the above, choose an *expression* of your uncertainties that suits the degree of precision you have (see §4.1 for a list). The media available to you will affect the formats (e.g. graphical) and expressions you can use, but also consider your *audience*, their *relationships* to you and to the subject and the *effects* you want to have on them. The psychology literature has little guidance as yet on the effects of each expression of uncertainty, but we know that stating uncertainty does not necessarily undermine trust.

Most importantly, *keep your expressions of the magnitude of uncertainty clearly separate from the magnitude of any evidence* you are trying to communicate (e.g. not confusing the *effect* of processed meat as a carcinogen, low; with the *certainty* that it is one, high. See box 4), and if you can *test the effect of your communication with your audience* at all, then do (we recommend the FDA's excellent guide when attempting this [189], and share the results—see box 5b)!

expressions of uncertainty, such as numerical ranges or verbal qualifiers. Whereas several studies have shown that uncertainty communication can affect people's affective responses and trust, more research is needed to understand under what conditions these effects can be positive or negative. To our knowledge, no systematic empirical research has been conducted about the effects of epistemic uncertainty communication on behaviour and decision-making. Nevertheless, general awareness of the potential intended and unintended effects is a first step to understanding how best to communicate uncertainty. This need is demonstrated by the examples that we found of communication schemes not based on empirical evidence, which have great potential to mislead (box 4). The reputational and real risk to life of miscommunication of uncertainty is great, as the example in box 1 highlights.

In addition, the psychological literature shows that different audiences, and even individuals within those audiences, are likely to have different reactions to various presentations of uncertainty. Many factors—from their personal characteristics and experiences to their relationship with the topic and with everyone involved in the production and communication of the information—could affect their responses. Both researchers and practitioners in this field need to recognize these factors and be aware of their potential effects.

Our scope in this paper necessarily had to be limited, not covering more general issues of user-engagement and communication design, for example the important idea of layered or progressive disclosure of information [187], or informal or interpersonal communication such as that between doctor and patient. It is worth noting that a review of research on the effects of visualizations of quantitative health risks concluded that the features of visualizations that improved accuracy of interpretation of the information are different from features that encouraged behaviour change; and features that viewers liked best may not be supportive of either of these goals [188]. These findings stress the importance of empirical evaluation of the effects of different formats of communication.

Of course, we are also only considering epistemic uncertainty about the past and present and not aleatory uncertainty about the future. However, these two are not always clearly distinct. For example, within the context of health information, a communication ostensibly about 'your risk' of a future event (apparently mostly aleatory uncertainty) could in reality be a measure of the current state of your body where the

> **Box 5(*b*).** Advice to researchers of how to communicate uncertainty around evidence.
>
> When designing your studies, it would be useful to clearly identify and state:
>
> — Is the uncertainty you are studying about the past/present or about the future—and is it likely to be perceived as past/present or future by the audience?
> — What are the psychological effects you are studying? *Cognition* (including understanding), *emotional* (affect), *trust* or *behaviour* (including decision-making)?
> — What are you varying about the uncertainty itself? (The *object*—is it a categorical or continuous object or is it a hypothesis; the *source* of the uncertainty; the *level* of the uncertainty—is it directly about the object or is it indirect: about the quality of the underlying evidence; or the *magnitude*?)
> — What are you varying about the way you communicate the uncertainty? (The form of the *expression*, the *format* or the *medium* by which you communicate?)
> — Are there aspects of the *people* doing the assessing/communicating of the uncertainty or the audience (such as their demographics, or relationship with the communicators or with the subject) that might also affect the endpoints you are interested in?

uncertainties are often purely epistemic (e.g. 'your cardiovascular risk' is actually a measure of the current health of your cardiovascular system). However, although the uncertainties are technically only about the present, the audience is likely to perceive them—framed as they are in terms of a future risk—as about the future. These are complexities that will need to be considered in empirical work.

As mentioned in the Introduction, we can foresee a time when such a framework could be used to create a practical guide to help people identify the uncertainties inherent in their work ('what' they want to communicate) and then identify the ideal form and expression that they should use to communicate those uncertainties, bearing in mind the medium, audience and desired effect. Until then, our best advice to both researchers and practitioners is summarized in boxes 5*a* and 5*b*. We hope this structured approach will aid people to communicate the epistemic uncertainty that exists about facts, numbers and science confidently and unapologetically—an approach we like to call 'muscular uncertainty'.

Because of its wide-reaching effects, uncertainty communication should be an important issue for policy makers, experts and scientists across many fields. Many of those fields carry the scars of attempts to avoid communicating uncertainty, or of poorly considered communications of uncertainty. These emphasize the need for a more considered approach to the topic, based on empirical evaluations done within an accepted framework. At present, however, this appears to be a science in its infancy. We can draw very limited conclusions from the current empirical work about the effects of communicating epistemic uncertainty and any underlying mechanisms. There is therefore a strong need for research specifically focused on communicating epistemic uncertainty and its impact on cognition, affect, trust, behaviour and decision-making. Early work needs confirming with large representative samples, and with observed or reported rather than hypothetical decision-making, as we currently have very little idea about generalizability of findings.

Most importantly, future work should try to manipulate components of communication systematically in order to unpick their effects and identify mediators. It seems extraordinary that so little is known about such an important topic, but our hope is that the framework we have set out here may help inform research in this vital and topical area.

Data accessibility. This does not apply to this review paper.

Authors' contributions. A.M.V.B. and L.Z. conducted the literature reviews with input from D.J.S., S.V.L., J.M. and A.B.G. A.M.V.B., S.V.L., A.L.J.F. and D.J.S. drafted a first version of the manuscript. J.M. and A.B.G. provided key inputs and critical feedback on the review, especially in the domain of official economic statistics. All authors contributed to the writing and approved the final version of the manuscript. D.J.S. has spent years struggling with this stuff and hopes he never has to write anything about it ever again.

Competing interests. We declare we have no competing interests.

Funding. Anne Marthe van der Bles, Sander van der Linden and Alexandra L. J. Freeman were supported by a donation from the David and Claudia Harding Foundation and by a grant from the Nuffield Foundation (OSP/43227), but the views expressed are those of the authors and not necessarily those of the Foundation. David J. Spiegelhalter and Lisa Zaval were supported by a donation from the David and Claudia Harding Foundation.

**Acknowledgements.** We thank Vivien Chopurian for her help with the formatting of the manuscript and the four reviewers for their exceptionally insightful and constructive comments. We would particularly like to thank Baruch Fischhoff and Charles Manski for their suggestions which hugely improved the manuscript.

# Technical appendix A. Formal communication of uncertainty based on statistical analysis

*Facts*: Absolute uncertainty about facts can be expressed as probabilities, such as claiming there is a 1 in 250 chance a fetus has a genetic abnormality. Such probabilities are expressing epistemic uncertainty rather than future randomness, and hence have a Bayesian interpretation in terms of reasonable betting odds. We have seen in box 2 on forensics how Bayes theorem can take into account new evidence and provide a revised absolute uncertainty, expressed as the posterior odds, based on the relative uncertainty expressed by the likelihood ratio and the prior odds. Confidence in the underlying knowledge might be expressed as a verbal expression of confidence in the likelihood ratio and the prior odds, or formally by placing intervals on these quantities.

*Numbers*: Let $\theta$ be some unknown continuous quantity. Given relevant data, $x$, a classical uncertainty statement regarding $\theta$ comprises a 95% confidence interval which can be denoted $[\theta_L(x), \theta_U(x)]$, sometimes expressed as a margin of error: for example, UK unemployment figures for January 2018 report a change on the previous quarter $-3000$, with a 95% confidence interval of $+/- 77\,000$. Formally, assuming the assumptions underlying the analysis are correct, in 95% of occasions the random interval $[\theta_L(x), \theta_U(x)]$ will include the true parameter value $\theta$.

Relative uncertainty about $\theta$ is given by the likelihood function $p(x|\theta)$, considered as a function of $\theta$. This provides the relative support for alternative possible values of $\theta$. A likelihood-based interval could be obtained from values of $\theta$ whose likelihood is not less than, say, 5% of the maximized likelihood, although this construction is rarely used.

Within a Bayesian approach, absolute uncertainty about $\theta$ is first expressed as a prior distribution $p(\theta|M)$, derived from external sources, denoted $M$. This is combined with the observed likelihood $p(x|\theta, M)$ to obtain a posterior distribution

$$p(\theta|x, M) \propto p(x|\theta, M)\, p(\theta|M).$$

Uncertainty about $\theta$ is therefore expressed as a full probability distribution, which may be summarized by say an interval containing 95% probability, known as a credible interval. In stark contrast to the classical approach, it is permissible to claim there is 95% probability that a specific credible interval contains the true parameter's value. Note that if a locally uniform prior distribution is assumed the Bayesian credible interval and the classical confidence interval will exactly, or approximately, match, but have very different interpretations.

Numerical intervals do not have to be based solely on statistical analysis, but can involve expert judgement. Metrology is defined as the 'the science of measurement, embracing both experimental and theoretical determinations at any level of uncertainty in any field of science and technology', and the standard *Guide to the Expression of Uncertainty in Measurement* (GUM [190]) specifies two sources of uncertainty: 'Type A' uncertainty arises from variation which can be reduced by taking further measurements, whereas 'Type B' uncertainty comes from non-random sources such as systematic errors and cannot be reduced simply by additional measurements. It is recommended that each type of uncertainty is quantified by a probability distribution—the Type A distribution will be based on statistical analysis, while the Type B distribution may be subjectively assessed on the basis on background knowledge. The 'combined standard uncertainty' $u_c$ is the square root of variance of the sum of these two sources of uncertainty, and the 'expanded uncertainty' is a corresponding interval of specified coverage (say 95%). This may be communicated as a measurement $+/- k\, u_c$.

*Scientific hypotheses*: Let $H_0$ be a scientific hypothesis. Then, given a summary statistic $x$ for which extreme values are unlikely if $H_0$ were true, the $p$-value for $H_0$ is the probability of observing such an extreme value of $x$, given $H_0$. The $p$-value can be considered a measure of relative uncertainty for $H_0$, as low values indicate lack of support for $H_0$ but do not provide a measure of absolute uncertainty.

A common situation is that $H_0$ represents the assumption that a parameter $\theta$ takes on a specific value $\theta_0$ while, under an alternative hypothesis $H_1$, $\theta$ can take on any value. In this situation, any observed $p$-value, say $p$, means that a $100(1 - p)\%$ confidence interval for $\theta$ just excludes $\theta_0$, thus corresponding to a direct expression of uncertainty about $\theta$.

With competing hypotheses $H_0$ and $H_1$, relative support is given by the Bayes factor identified as $B_{01} = p(x|H_0)/p(x|H_1)$, which is precisely the likelihood ratio applied to scientific hypotheses. Unknown parameters need to be integrated out assuming prior distributions within each hypothesis, so that $p(x|H_i) = \int p(x|\theta, H_i)p(\theta|H_i)\mathrm{d}\theta$.

It is possible to express absolute uncertainty about scientific hypotheses by assessing prior odds $p(H_0)/p(H_1)$, and applying Bayes theorem to give posterior odds.[5] But there is often reluctance to express probabilities for scientific hypotheses, since these are generally not directly provable as true or false.

Information criteria provide a metric for comparing the support for alternative hypotheses. Let $\theta_0$ and $\theta_1$ be the free parameters in $H_0$ and $H_1$, respectively, and suppose $L$ is the ratio of maximized likelihoods, $L = p(x|\theta_{0,\max}, H_0)/p(x|\theta_{1,\max}, H_1)$, where the $\theta_{0,\max}$ and $\theta_{1,\max}$ represent the parameter values that maximize the respective likelihoods. Then, for example, Akaike's Information Criterion (AIC) and the Bayesian Information Criterion (BIC) are defined by

$$\mathrm{AIC} = -2\log L - 2p$$
$$\mathrm{BIC} = -2\log L - p\log n,$$

where $p$ is the difference in the number of free parameters in $H_0$ and $H_1$, and $n$ is the sample size. These are only relative measures of support, allowing a ranking in terms of the confidence held in alternative models.

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
