## [Reviewer comments · Royal Society Open Science]

Review History

RSOS-181870.R0 (Original submission)

Review form: Reviewer 1 (Charles Manski)

Is the manuscript scientifically sound in its present form?

Yes

Are the interpretations and conclusions justified by the results?

Yes

Is the language acceptable?

Yes

Is it clear how to access all supporting data?

Not Applicable

Do you have any ethical concerns with this paper?

No

Have you any concerns about statistical analyses in this paper?

No

Recommendation?

Accept with minor revision (please list in comments)

Comments to the Author(s)

See attached file (Appendix A).

Review form: Reviewer 2 (Paul Han)

Is the manuscript scientifically sound in its present form?

Yes

Are the interpretations and conclusions justified by the results?

Yes

Is the language acceptable?

Yes

Is it clear how to access all supporting data?

Not Applicable

Do you have any ethical concerns with this paper?

No

Have you any concerns about statistical analyses in this paper?

No

Recommendation?

Accept with minor revision (please list in comments)

Comments to the Author(s)

See attachment (Appendix B).

Review form: Reviewer 3 (Baruch Fischhoff)

Is the manuscript scientifically sound in its present form?

Yes

Are the interpretations and conclusions justified by the results?

Yes

Is the language acceptable?

Yes

Is it clear how to access all supporting data?

Not Applicable

Do you have any ethical concerns with this paper?

No

Have you any concerns about statistical analyses in this paper?

No

Recommendation?

Major revision is needed (please make suggestions in comments)

Comments to the Author(s)

I believe that this is an important article. Hoping to help it along its way, I started to make detailed comments on the paper. After a while, though, I developed an overriding concern, namely, that it is hard to take in all that the authors are offering. That concern left me thinking that I might serve this worthy cause best by offering some strategic thoughts on the exposition:

I often found the style slow-paced. I believe that was done in the interests of making it accessible to a broad, non-specialist audience. However, with such a long paper, that style increases the commitment from readers. Here, I think that some ruthless tightening could help.

The authors have framed the paper in terms of a task analysis (Lasswell) which recognizes the complexity of communication processes. They have further elaborated it by drawing on their own ability to provide epistemological depth in areas that others have often glossed over (deepening Lasswell's "what"). However, that complexity makes it hard to get a big picture. The authors' use of exhibits should make a published version easier to navigate than the present manuscript version — where the exhibits break up the text, rather than being objects whose gist a reader could get, amplifying the text, then return to later if interested in the detail. I am afraid that the authors will lose portions of their audience unless they can find a better way of presenting the work. I have a radical suggestion at the end of these general comments.

The authors have extensive, eclectic references to other conceptual schemes, which will be a valuable resource in themselves. However, they often treat the content of those reference very briefly, seemingly dismissing their predecessors without serious attention to those efforts. One good reason is that, as the authors note, earlier work is somewhat chaotic and often incoherent, making it hard to treat systematically. As a result, it often feels as though they are giving their predecessors short shrift, for a scholarly publication. An alternative approach would be to begin each section with their own proposal (as happens with some sections), then end by briefly putting earlier approaches in that context.

There are some sections that seem just to bookmark topics in Lasswell's scheme, without reviewing the research in any depth or connecting the topic with the authors' own contribution.

My overall sense is that there are several papers lurking here, which sometimes end up working at cross purposes:

1. A motivating essay on the importance of communicating uncertainty well and the perils of doing so poorly, emphasizing the common features across domains.
2. An essay on the complexity of communication processes, as represented (and known since) the venerable Lasswell task analysis, and how failure in any component can imperil the entire effort.
3. The authors' contribution to what might be the hardest part of the effort, characterizing uncertainty (Sections 2 and 3.1), showing how to get it right.

4. A review of synthetic representations of uncertainty and how well they are understood (Section 3.2 and Section 5), interpreted in light of that scheme.
5. A critique of inept schemes (e.g., IARC, Box 4, some of the cells in Table 1, the case studies).

My recommendation is to write these articles separately, which would also free the authors from having to say something about each element in the Lasswell scheme, which leads to some unsatisfactory passages in this manuscript – while still giving Lasswell his due, in a field grown so large that people naturally focus on pieces. It would lead to dropping some things, for later, fuller treatment in a book on the topic. For this setting, I would briefly note topics 1 and 2, then focus on 3, followed by 4, emphasizing the conceptual scheme in 3. I realize that this is a presumptuous suggestion. I also suspect that the authors are exhausted at having gotten everything out in one place. (I would be.) If they chose to see the current article through to publication, it would be a fine contribution. However, I think that it would fall short of the more definitive, accessible piece that they have in them, which could structure future work in the area.

Page 2

The Abstract seems to focus more on what others haven't done than on what the authors have done. Could it be rewritten to provide the gist of their scheme (recognizing that it takes many pages to do it fuller justice).

2/8 “relatively little is known” seems at odds with what follows. Perhaps something along the lines of “what is known is widely scattered”

2/12. Wouldn't analysis be necessary for communication? It often isn't. However, I think that it would be better to frame the paper as following accepted practices, whose application is non-trivial.

2/20 “has not been systematic” is arguably true, but seems to be either vague or faulting others for not following a system that has just been introduced.

2/23 The final sentence doesn't add much. What would the authors like to have happen next?

Page 3

Given the strong suspicion in the US (at least among most of the authors' likely readers) that the books were cooked on Iraq, this does not seem like an effective example, even if Butler is held in high esteem in the UK. (“Vice” has just opened in theaters here.) The authors might consider Sherman Kent's classic “Words of Estimative Probability” (<https://www.cia.gov/library/center-for-the-study-of-intelligence/csi-publications/books-and-monographs/sherman-kent-and-the-board-of-national-estimates-collected-essays/6words.html>).

Could the authors add some surveys in the US or elsewhere, given that they want (and deserve) a broader audience than the UK (submission to the Royal Society notwithstanding).

3/8 The authors might consider the relevance of my own piece with such evidence in a somewhat analogous US publication: Fischhoff, B. (2012, Summer). Communicating uncertainty: Fulfilling the duty to inform. *Issues in Science and Technology*, 28(4), 63-70

3/22-28 See comments on Abstract.

3/34ff. See comments on Abstract (2/12)

3/48ff Without saying what predecessors did and why the authors deem it a failure, this paragraph doesn't advance the paper very much. I would cut it, adding the references to a string somewhere.

Page 4

4/7ff. To my taste, this paragraph wanders, too. I don't think that anything would be lost by cutting it and moving straight to the authors' proposal.

Using Lasswell's classing task analysis seems like an excellent move. However, I think that it needs some reference(s) to more recent scholarship, on how well it has stood the test of time and what others have done with it.

p. 5

5/24 Would a close reading of the authors' scheme show that they have not considered decision making?

page 7

7/8ff Here's one of those places where I thought that the paper's agreeable chattiness slows things down

Section 1. I don't think that this section served much purpose beyond having the authors say something about this topic, without much by way of referencing the vast (I believe) literature on source credibility.

page 8ff

Section 2. This section had the makings of a valuable standalone section on characterizing uncertainty, which approached a form that readers might apply to their own domains. It might be coupled with Section 3.1, if the two category schemes were more tightly integrated. It would be stronger by more explicit contrasts with schemes having similar aspirations. I was curious about the distinctions that the authors saw with Ravetz and Funtowicz, who have similar general aspirations. I was looking for GRADE/CONSORT here (and pleased to find it later). I suggest moving the references to behavioral research, to a later section where they could be treated more systematically.

p. 11

Box 3. Is there a place here for a reference to non-Bayesian treatments of evidence?

p. 12

12/6 I don't think that the implicit references to risk comparisons here did that topic justice, and would leave it to an empirical section.

12/16 This paragraph felt out of place without more systematic treatment of how uncertainty is to be used.

Section 3. It seemed like there should be a stronger connection between this section and the previous one. Given how much thought the authors have given to these issues, I was interested in knowing what constraints they saw what was being communicated (Section 2) imposed on how it was communicated (Section 3). Some of the two seemed to be intertwined (e.g., the top of p.13). I have the feeling that both sections would be strengthened by moving some of the commentary in Section 3 to Section 2. Alternatively, Section 2 might be folded into Section 3, eliminating the seeming (to me at least) disconnect between the two.

p. 13

Some of the material here seemed to belong to Section 2.

p. 14

I loved this table, with so many good and bad examples brought together. However, as suggested in my general comments, absorbing both simultaneously is difficult, for a reader wondering "what should I don?"

p. 19

As mentioned, although I agree that GRADE/CONSORT is a powerful communication tool (and have advocated it as such), I think that its value arises from its practical approach to characterizing uncertainty — hence belonging to Section 2, then reaping its benefits in Section 3. Perhaps I am missing something fundamental here, however, it seems to me that that the two sections could be combined — at the price of abandoning the Lasswell scheme.

bottom. This seems like a good example of a bad example, which muddles the normative exposition here (of what to do right).

p. 22

Section 3.2 wasn't really connected to what preceded it and seems part of the story of Section 5. The representations presented here, like those in Section 5, are compromises designed for practical purposes, rather than having essential, epistemological properties, like the core topics of Sections 2 and 3.1.

p. 24

Section 4, like Section 1, didn't seem to serve any purpose beyond having something to say about this element in the Lasswell scheme. Its few references seemed somewhat haphazardly selected. It has less anti-public tone than most brief summaries. However, it doesn't do justice to the empirical literature or the ways in which the authors' proposal would lead to reinterpreting its contents. I would cut it.

p. 25-26

This section got off to a slow start.

p. 27-29

Section 5.1 builds nicely on the scheme introduced in Section 3.1. The studies that I knew were very nicely summarized. However, it was often unclear how authoritative the authors intended their summaries to be. It seemed like they sometimes picked illustrative studies, suggesting what research might find, and other times offered their interpretation of the results of more comprehensive reviews. As a result, the section felt uneven and not as effective an explication of the scheme in Section 3.1 as it might have been. For example, where fundamental categories are combined, is that because the authors were trying to cram a lot into an already long article or because there was no prior research? If the latter, then I think that they would advance the cause by explicating the research program.

pp. 29-31

Section 5.2 nicely applies the scheme from 3.1 to an emerging topic (which I know less well). Here, it feels though the authors have cited every study that they could find. If so, then does the topic merit the same length as Section 5.1?

pp. 31-33

An important topic. Some of the results seemed hard to interpret absent the context provided by the issues in the preceding sections. Here, too, I wondered why some topics in the authors' scheme were not addressed.

pp. 33-34

Section 5.4 was less satisfying than its predecessors, which looked at one aspect of individuals' responses. Dealing with decisions, which involve multiple aspects, would require more systematic treatment. For example, what standards did [149] and [150] apply when determining rationality? The examples here seem to be mostly hypothetical decisions, in any case. There are also so few that the authors' scheme is abandoned. I suggest cutting the section and folding anything relevant into the preceding sections.

p. 35ff

I liked the case studies, which were very nicely written. I've seen parts of some of them before. Manski has a recent piece in PNAS interpreting his work from a communication perspective: <https://www.pnas.org/content/early/2018/11/21/1722389115>.

pp. 43ff

I thought that the statement of the scheme was useful and might have been well-placed at the beginning of the article, replacing the one on p. 6 which, to my mind, promised detail on some topics that the article didn't deliver (e.g., Sections 1 and 4). The rest of the section, though, seemed to say mostly that the issues were complicated and the situation murky. I didn't feel like I got very much from it. My suggestion would be to frame the empirical accounts in Section 5 more explicitly as illustrative. Use them to make the case that empirical research is essential – supported by the ill-conceived schemes that appear throughout the article.

Baruch Fischhoff

Review form: Reviewer 4 (Andrew Przybylski)

Is the manuscript scientifically sound in its present form?

Yes

Are the interpretations and conclusions justified by the results?

Yes

Is the language acceptable?

Yes

Is it clear how to access all supporting data?

Not Applicable

Do you have any ethical concerns with this paper?

No

Have you any concerns about statistical analyses in this paper?

No

Recommendation?

Accept with minor revision (please list in comments)

Comments to the Author(s)

In this paper the author(s) advance a meta theoretical framework for communicating scientific uncertainty. This is a topic that I as a scientist find extremely challenging (in practice) and have largely not felt satisfied with many of the proposed ways of addressing the topic as they're often siloed within specific disciplines which I find it difficult to relate. With that in mind I was largely positive on this paper as I learned a lot as a reader. Some comments highlight areas where things might have been missing.

1. There are areas where I could see competing interests feeding into the framework here (e.g. at the who and what stages). Could the authors consider cases where this has been handled, to some extent, e.g. pharmaceuticals, and areas where this is less clear, e.g. social data science?
2. I think the repetitional costs of miscommunication around uncertainty are something that the paper could expand on. This is a prominent aspect of the paper at the start and can be looped back in via the argumentation in an additional paragraph in the conclusion.

3. I think a synthetic example could support the narrative and draw the reader in more fully. The examples in the discussion (p.44) could be expanded to this end. In other words, consider presenting how to different scientific programs on topic x would play out differently in the future given they take different paths through the framework the authors outline. I understand that this pivots the system from descriptive to proscriptive but this could be very useful.

this review is signed,
Andrew K. Przybylski
University of Oxford

Decision letter (RSOS-181870.R0)

25-Jan-2019

Dear Dr van der Bles,

The editors assigned to your paper ("Communicating uncertainty about facts, numbers and science") have now received comments from reviewers. We would like you to revise your paper in accordance with the referee and Associate Editor suggestions which can be found below (not including confidential reports to the Editor). Please note this decision does not guarantee eventual acceptance.

Please submit a copy of your revised paper before 17-Feb-2019. Please note that the revision deadline will expire at 00.00am on this date. If we do not hear from you within this time then it will be assumed that the paper has been withdrawn. In exceptional circumstances, extensions may be possible if agreed with the Editorial Office in advance. We do not generally allow multiple rounds of revision so we urge you to make every effort to fully address all of the comments at this stage. If deemed necessary by the Editors, your manuscript will be sent back to one or more of the original reviewers for assessment. If the original reviewers are not available, we may invite new reviewers.

If your study uses humans or animals please include details of the ethical approval received, including the name of the committee that granted approval. For human studies please also detail

whether informed consent was obtained. For field studies on animals please include details of all permissions, licences and/or approvals granted to carry out the fieldwork.

- Data accessibility

If you wish to submit your supporting data or code to Dryad (<http://datadryad.org/>), or modify your current submission to dryad, please use the following link:
<http://datadryad.org/submit?journalID=RSOS&manu=RSOS-181870>

- Competing interests

- Authors' contributions

- Acknowledgements

- Funding statement

Kind regards,
Andrew Dunn
Senior Publishing Editor

on behalf of Professor Chris Chambers (Associate Editor) and Essi Viding (Subject Editor)
 openscience@royalsociety.org

Associate Editor's comments (Professor Chris Chambers):

Four expert reviewers have now assessed your manuscript. As you will see, all find merit in the article while also offering a wide range of constructive suggestions to clarify (and potentially simplify) the structure of the paper, clarify key concepts, and provide additional working examples of your main arguments (eg. Reviewer 4, point 3). A major revision is therefore recommended.

Comments to Author:

Reviewers' Comments to Author:
 Reviewer: 1

Comments to the Author(s)
 See attached file

Reviewer: 2

Comments to the Author(s)
 See attachment

Reviewer: 3

Comments to the Author(s)

I believe that this is an important article. Hoping to help it along its way, I started to make detailed comments on the paper. After a while, though, I developed an overriding concern, namely, that it is hard to take in all that the authors are offering. That concern left me thinking that I might serve this worthy cause best by offering some strategic thoughts on the exposition:

I often found the style slow-paced. I believe that was done in the interests of making it accessible to a broad, non-specialist audience. However, with such a long paper, that style increases the commitment from readers. Here, I think that some ruthless tightening could help.

The authors have framed the paper in terms of a task analysis (Lasswell) which recognizes the complexity of communication processes. They have further elaborated it by drawing on their own ability to provide epistemological depth in areas that others have often glossed over (deepening Lasswell's "what"). However, that complexity makes it hard to get a big picture. The authors' use of exhibits should make a published version easier to navigate than the present manuscript version – where the exhibits break up the text, rather than being objects whose gist a reader could get, amplifying the text, then return to later if interested in the detail. I am afraid that the authors will lose portions of their audience unless they can find a better way of presenting the work. I have a radical suggestion at the end of these general comments.

The authors have extensive, eclectic references to other conceptual schemes, which will be a valuable resource in themselves. However, they often treat the content of those references very briefly, seemingly dismissing their predecessors without serious attention to those efforts. One

good reason is that, as the authors note, earlier work is somewhat chaotic and often incoherent, making it hard to treat systematically. As a result, it often feels as though they are giving their predecessors short shrift, for a scholarly publication. An alternative approach would be to begin each section with their own proposal (as happens with some sections), then end by briefly putting earlier approaches in that context.

There are some sections that seem just to bookmark topics in Lasswell's scheme, without reviewing the research in any depth or connecting the topic with the authors' own contribution.

My overall sense is that there are several papers lurking here, which sometimes end up working at cross purposes:

1. A motivating essay on the importance of communicating uncertainty well and the perils of doing so poorly, emphasizing the common features across domains.
2. An essay on the complexity of communication processes, as represented (and known since) the venerable Lasswell task analysis, and how failure in any component can imperil the entire effort.
3. The authors' contribution to what might be the hardest part of the effort, characterizing uncertainty (Sections 2 and 3.1), showing how to get it right.
4. A review of synthetic representations of uncertainty and how well they are understood (Section 3.2 and Section 5), interpreted in light of that scheme.
5. A critique of inept schemes (e.g., IARC, Box 4, some of the cells in Table 1, the case studies).

My recommendation is to write these articles separately, which would also free the authors from having to say something about each element in the Lasswell scheme, which leads to some unsatisfactory passages in this manuscript – while still giving Lasswell his due, in a field grown so large that people naturally focus on pieces. It would lead to dropping some things, for later, fuller treatment in a book on the topic. For this setting, I would briefly note topics 1 and 2, then focus on 3, followed by 4, emphasizing the conceptual scheme in 3. I realize that this is a presumptuous suggestion. I also suspect that the authors are exhausted at having gotten everything out in one place. (I would be.) If they chose to see the current article through to publication, it would be a fine contribution. However, I think that it would fall short of the more definitive, accessible piece that they have in them, which could structure future work in the area.

Page 2

The Abstract seems to focus more on what others haven't done than on what the authors have done. Could it be rewritten to provide the gist of their scheme (recognizing that it takes many pages to do it fuller justice).

2/8 “relatively little is known” seems at odds with what follows. Perhaps something along the lines of “what is known is widely scattered”

2/12. Wouldn't analysis be necessary for communication? It often isn't. However, I think that it would be better to frame the paper as following accepted practices, whose application is non-trivial.

2/20 “has not been systematic” is arguably true, but seems to be either vague or faulting others for not following a system that has just been introduced.

2/23 The final sentence doesn't add much. What would the authors like to have happen next?

Page 3

Given the strong suspicion in the US (at least among most of the authors' likely readers) that the books were cooked on Iraq, this does not seem like an effective example, even if Butler is held in high esteem in the UK. (“Vice” has just opened in theaters here.) The authors might consider Sherman Kent's classic “Words of Estimative Probability” (<https://www.cia.gov/library/center-for-the-study-of-intelligence/csi-publications/books-and-monographs/sherman-kent-and-the-board-of-national-estimates-collected-essays/6words.html>).

Could the authors add some surveys in the US or elsewhere, given that they want (and deserve) a broader audience than the UK (submission to the Royal Society notwithstanding).

3/8 The authors might consider the relevance of my own piece with such evidence in a somewhat analogous US publication: Fischhoff, B. (2012, Summer). Communicating uncertainty: Fulfilling the duty to inform. *Issues in Science and Technology*, 28(4), 63-70

3/22-28 See comments on Abstract.

3/34ff. See comments on Abstract (2/12)

3/48ff Without saying what predecessors did and why the authors deem it a failure, this paragraph doesn't advance the paper very much. I would cut it, adding the references to a string somewhere.

Page 4

4/7ff. To my taste, this paragraph wanders, too. I don't think that anything would be lost by cutting it and moving straight to the authors' proposal.

Using Lasswell's classing task analysis seems like an excellent move. However, I think that it needs some reference(s) to more recent scholarship, on how well it has stood the test of time and what others have done with it.

p. 5

5/24 Would a close reading of the authors' scheme show that they have not considered decision making?

page 7

7/8ff Here's one of those places where I thought that the paper's agreeable chattiness slows things down

Section 1. I don't think that this section served much purpose beyond having the authors say something about this topic, without much by way of referencing the vast (I believe) literature on source credibility.

page 8ff

Section 2. This section had the makings of a valuable standalone section on characterizing uncertainty, which approached a form that readers might apply to their own domains. It might be coupled with Section 3.1, if the two category schemes were more tightly integrated. It would be stronger by more explicit contrasts with schemes having similar aspirations. I was curious about the distinctions that the authors saw with Ravetz and Funtowicz, who have similar general aspirations. I was looking for GRADE/CONSORT here (and pleased to find it later). I suggest moving the references to behavioral research, to a later section where they could be treated more systematically.

p. 11

Box 3. Is there a place here for a reference to non-Bayesian treatments of evidence?

p. 12

12/6 I don't think that the implicit references to risk comparisons here did that topic justice, and would leave it to an empirical section.

12/16 This paragraph felt out of place without more systematic treatment of how uncertainty is to be used.

Section 3. It seemed like there should be a stronger connection between this section and the previous one. Given how much thought the authors have given to these issues, I was interested in knowing what constraints they saw what was being communicated (Section 2) imposed on how it was communicated (Section 3). Some of the two seemed to be intertwined (e.g., the top of p.13). I have the feeling that both sections would be strengthened by moving some of the

commentary in Section 3 to Section 2. Alternatively, Section 2 might be folded into Section 3, eliminating the seeming (to me at least) disconnect between the two.

p. 13

Some of the material here seemed to belong to Section 2.

p. 14

I loved this table, with so many good and bad examples brought together. However, as suggested in my general comments, absorbing both simultaneously is difficult, for a reader wondering "what should I do?"

p. 19

As mentioned, although I agree that GRADE/CONSORT is a powerful communication tool (and have advocated it as such), I think that its value arises from its practical approach to characterizing uncertainty — hence belonging to Section 2, then reaping its benefits in Section 3. Perhaps I am missing something fundamental here, however, it seems to me that that the two sections could be combined — at the price of abandoning the Lasswell scheme. bottom. This seems like a good example of a bad example, which muddles the normative exposition here (of what to do right).

p. 22

Section 3.2 wasn't really connected to what preceded it and seems part of the story of Section 5. The representations presented here, like those in Section 5, are compromises designed for practical purposes, rather than having essential, epistemological properties, like the core topics of Sections 2 and 3.1.

p. 24

Section 4, like Section 1, didn't seem to serve any purpose beyond having something to say about this element in the Lasswell scheme. Its few references seemed somewhat haphazardly selected. It has less anti-public tone than most brief summaries. However, it doesn't do justice to the empirical literature or the ways in which the authors' proposal would lead to reinterpreting its contents. I would cut it.

p. 25-26

This section got off to a slow start.

p. 27-29

Section 5.1 builds nicely on the scheme introduced in Section 3.1. The studies that I knew were very nicely summarized. However, it was often unclear how authoritative the authors intended their summaries to be. It seemed like they sometimes picked illustrative studies, suggesting what research might find, and other times offered their interpretation of the results of more comprehensive reviews. As a result, the section felt uneven and not as effective an explication of the scheme in Section 3.1 as it might have been. For example, where fundamental categories are combined, is that because the authors were trying to cram a lot into an already long article or because there was no prior research? If the latter, then I think that they would advance the cause by explicating the research program.

pp. 29-31

Section 5.2 nicely applies the scheme from 3.1 to an emerging topic (which I know less well). Here, it feels though the authors have cited every study that they could find. If so, then does the topic merit the same length as Section 5.1?

pp. 31-33

An important topic. Some of the results seemed hard to interpret absent the context provided by the issues in the preceding sections.. Here, too, I wondered why some topics in the authors' scheme were not addressed.

pp. 33-34

Section 5.4 was less satisfying than its predecessors, which looked at one aspect of individuals' responses. Dealing with decisions, which involve multiple aspects, would require more systematic treatment. For example, what standards did [149] and [150] apply when determining rationality? The examples here seem to be mostly hypothetical decisions, in any case. There are also so few that the authors' scheme is abandoned. I suggest cutting the section and folding anything relevant into the preceding sections.

p. 35ff

I liked the case studies, which were very nicely written. I've seen parts of some of them before. Manski has a recent piece in PNAS interpreting his work from a communication perspective: <https://www.pnas.org/content/early/2018/11/21/1722389115>.

pp. 43ff

I thought that the statement of the scheme was useful and might have been well-placed at the beginning of the article, replacing the one on p. 6 which, to my mind, promised detail on some topics that the article didn't deliver (e.g., Sections 1 and 4). The rest of the section, though, seemed to say mostly that the issues were complicated and the situation murky. I didn't feel like I got very much from it. My suggestion would be to frame the empirical accounts in Section 5 more explicitly as illustrative. Use them to make the case that empirical research is essential — supported by the ill-conceived schemes that appear throughout the article.

Baruch Fischhoff

Reviewer: 4

Comments to the Author(s)

In this paper the author(s) advance a meta theoretical framework for communicating scientific uncertainty. This is a topic that I as a scientist find extremely challenging (in practice) and have largely not felt satisfied with many of the proposed ways of addressing the topic as they're often siloed within specific disciplines which I find it difficult to relate. With that in mind I was largely positive on this paper as I learned a lot as a reader. Some comments highlight areas where things might have been missing.

1. There are areas where I could see competing interests feeding into the framework here (e.g. at the who and what stages). Could the authors consider cases where this has been handled, to some extent, e.g. pharmaceuticals, and areas where this is less clear, e.g. social data science?

2. I think the repetitional costs of miscommunication around uncertainty are something that the paper could expand on. This is a prominent aspect of the paper at the start and can be looped back in via the argumentation in an additional paragraph in the conclusion.

3. I think a synthetic example could support the narrative and draw the reader in more fully. The examples in the discussion (p.44) could be expanded to this end. In other words, consider presenting how to different scientific programs on topic x would play out differently in the future

given they take different paths through the framework the authors outline. I understand that this pivots the system from descriptive to proscriptive but this could be very useful.

this review is signed,
Andrew K. Przybylski
University of Oxford

Author's Response to Decision Letter for (RSOS-181870.R0)

See Appendix C.

RSOS-181870.R1 (Revision)

Review form: Reviewer 2 (Paul Han)

Is the manuscript scientifically sound in its present form?

Yes

Are the interpretations and conclusions justified by the results?

Yes

Is the language acceptable?

Yes

Is it clear how to access all supporting data?

Not Applicable

Do you have any ethical concerns with this paper?

No

Have you any concerns about statistical analyses in this paper?

No

Recommendation?

Accept with minor revision (please list in comments)

Comments to the Author(s)

General Comments

I reiterate my conclusion from my prior review: this is an impressive, well-done, well-written review. I believe the authors have carefully addressed my critiques and those of the other reviewers, and I commend their responsiveness overall. I do believe there are lingering problems, which mainly reflect my disagreement with some of the authors' conceptual decisions,

which I outline below. There can certainly be legitimate disagreement on these issues, but I think some added consideration of these issues would strengthen the manuscript.

Specific Comments

1. Pp 3-5: I appreciate the authors' re-working of the introduction and their inclusion of more explicit discussion of the definition of uncertainty. However, I still have a problem with the term "future" uncertainty and its conceptual distinction from epistemic uncertainty. I maintain that as described by the authors, these are not mutually exclusive categories, and the term "future" just confuses matters. To me, this distinction is not useful and in fact misleading. Epistemic uncertainty can encompass issues pertaining to the future as well as the present and past. The issues comes down to definitions, or course, but I believe that temporality is not the factor that determines whether uncertainty is "epistemic" or not, and believe I'm not alone in this view.

Compounding the confusion, in my mind, is that the authors simultaneously introduce a dichotomy between unknowability (can't know) and unknownness (don't know), and place it alongside the distinction between future and epistemic uncertainty. This juxtaposition implies that the defining feature of "future" uncertainty is unknowability, while the defining feature of "epistemic" uncertainty is mere unknownness. This distinction, too, is problematic. Even if an aspect of reality is in principle knowable, one can still be uncertain if it is unknown for the time being – and this uncertainty is clearly epistemic (pertaining to one's knowledge). Furthermore, the author's conflation of all these issues also implies that uncertainties about the present and past are knowable – and this is clearly not true. There are many things about the present and past that we not only don't know, but can't know, for many reasons.

I think what the authors really want to do is to distinguish between uncertainty that arises from or pertains to limits to knowledge (epistemic) vs. the fundamental indeterminacy or randomness of the world – this 2nd uncertainty is captured by the term aleatory/aleatoric, which the authors used in the previous version. I would much prefer they use that term instead of the word "future": it is much less ambiguous and confusing, and more logically coherent and consistent with established terminology in the literature.

2. Pp 11-12: I appreciate the authors' attempt to rework the section previously entitled "type of uncertainty". They have now changed the focus to the idea of "levels of uncertainty" with 2 main ones: direct and indirect. However, I believe this change has created new conceptual difficulties that again are raised by the choice of specific words that have specific connotations. I believe what they are now referring to is a distinction that decision theorists since Knight (whom the authors cite) have described variously as 1st-order vs 2nd-order uncertainty, known vs unknown probabilities, risk vs. uncertainty, probability vs. ambiguity (Ellsberg). The essence of this distinction is a metacognitive reflexiveness: a thinking about thinking – in this case, an uncertainty about one's uncertainty. The distinction captures a secondary mental state of uncertainty focused back upon a prior uncertainty about some issue – in the authors' words, "Caveats" about the "quality of the underlying knowledge" regarding "facts, numbers, and hypotheses." The authors' example is a case of direct uncertainty as concerning "the absolute probability of guilt," which is compounded by indirect uncertainty concerning the "credibility to be given to an individual's testimony concerning this item of evidence." This distinction is conceptually equivalent to the existing, well-established distinction between 1st vs 2nd order uncertainty, or probability vs ambiguity; the underlying phenomenon boils down to higher-order uncertainty about uncertainty.

The question is which set of words is most useful to express a concept. To me, the words "direct" and "indirect" do not hit the mark with their connotations, and do not clarify but instead obscure the essence of the underlying phenomenon. To me, these words do not capture the concept of reflexiveness, and don't even match up to the notion of "levels." It is certainly the authors'

prerogative to apply a new terminology, but then I believe they should acknowledge the other more established terms, and make explicit how their terms differ or not.

I also recognize that some scholars do not believe in 2nd-order uncertainty; Morgan and others, for example, have argued that the notion of uncertainty about uncertainty is incoherent and not useful – that everything is uncertainty, and it is not necessary to distinguish levels. I personally do not agree with this view in theory, and believe there is ample empirical evidence showing that people behave as if 2nd-order uncertainty did exist. However, if this is the theoretical rationale for the author's choice of terms, then they should at least make it explicit and defend it.

3. P 15, table: a related fundamental disagreement I have with the authors is their treatment of representations of imprecision as expressions of 1st-order (what they call "direct") rather than 2nd-order ("indirect") uncertainty ("ambiguity" in Ellsberg's terms). I believe, as do many decision theorists, that imprecision signifies 2nd-order uncertainty (uncertainty about uncertainty). From a statistical modeling standpoint, imprecision does manifest uncertainty in probability estimates; confidence intervals are wider when either the evidence used in estimating probabilities or our modeling methods themselves are more inadequate or unreliable. And from a psychological standpoint, imprecision is perceived as signifying uncertainty: people respond differently to probabilities when imprecision is expressed. They perceive imprecise ranges as more uncertain than point estimates, and display ambiguity aversion in response to them. For all these reasons I believe that representations of imprecision in probability estimates such as probability distributions and ranges do not belong in the same conceptual category as point estimates of probability.

4. P 20: for this reason I also disagree with the assertion that "Methods for communicating the quality of the underlying evidence do not give quantitative information about absolute values or facts..." I think probability distributions, risk ranges, and confidence intervals do just that, by signifying with quantitative precision the imprecision of our estimates. That is not to say they are ideal or necessary or sufficient representations of 2nd-order uncertainty, only that they constitute quantitative expressions of it.

5. p 25: again, I think the use of the term "future" uncertainty is misleading, and also do not agree with the assertion in lines 12-13 that "ambiguity aversion is mostly about people's aversion to using this information for making decisions about a future event." To me, that statement is much too strong. It's true that the classic experimental paradigm for demonstrating ambiguity aversion has consisted of gambling experiments using balls and urns, where the outcome was the willingness to bet on an unknown probability. However, this phenomenon generalizes to all kinds of judgments and decisions as well as not only behavioral but cognitive and emotional responses, and is thought to specifically reflect the effects of epistemic uncertainty (2nd-order uncertainty about one's uncertainty) arising from limitations in the reliability, credibility, or adequacy of one's information. This distinguishes it from risk aversion. A large body of empirical evidence has also shown that ambiguity aversion also consists not only of decisions about a future event, but current risk perceptions, judgments, preferences, and emotional responses.

6. P 43: I appreciate and agree with the authors' discussion at the bottom of the page on the difficulty of distinguishing between past, present, and future uncertainty, and think it would be good to discuss and highlight this important caveat much earlier in the paper.

Review form: Reviewer 3 (Baruch Fischhoff)

Is the manuscript scientifically sound in its present form?

Yes

Are the interpretations and conclusions justified by the results?

Yes

Is the language acceptable?

Yes

Is it clear how to access all supporting data?

Not Applicable

Do you have any ethical concerns with this paper?

No

Have you any concerns about statistical analyses in this paper?

No

Recommendation?

Accept with minor revision (please list in comments)

Comments to the Author(s)

I appreciate the authors' thoughtful, and interesting, responses to my review and those of the other reviewers. I gave such detailed comments, in part, because I believe this to be an important project, which I was hoping to help along its way, and, in part, because it did not flow for me, despite being locally very clear. I decided going through it slowly was the best way to think about what might be redone. This time, I just read for pleasure.

I have three residual comments, two minor and one major, along with a few references that might be of use.

Minor Comment 1: I think that the risks with indirect uncertainty practices (p. 20ff) are greater than the text implies, in cases where audience members are not privy to professional conventions. The authors might consider the conventions that John Cohen introduced as part of his campaign to get psychologists to perform power analyses: small, medium, and large effect sizes. Those terms and their statistical equivalents reflected his intuition of what kind of results impressed psychologists and, I am guessing, what would make them feel good enough to adopt his methods. What do non-psychologists think if psychologists say, or act like, they have large effects – not knowing that it is large for psychologists. Similarly, do people interpret “high quality” as “high quality, given the challenges of clinical trials”?

Minor Comment 2: The authors repeatedly say that there is very little empirical research on these topics, but then cite a lot of it (witness the references to empirical papers). Indeed, in the spirit of their paper, it communicated to me that, on some topics, there is enough evidence for it to be inconclusive. They might revisit the wording on this topic,

Major Comment: I was disappointed not to see a strong concluding statement calling for the empirical evaluation of communications. To my mind, its absence substantially undermines the value of the entire article. The authors have demonstrated the dangers of the amateur-hour approach that is the norm in scientific communication. They have established that the empirical results are mixed on those issues where there have been studies. Their framework demonstrates

the complexity of real-world communication. Their summary suggestions could be very helpful in designing communications that are worth testing. However, they only provide better guesses at what might work. If readers believe that following these suggestions will guarantee success, then they are being set up for failure. We structured the FDA guide so that evaluation is central to the effort and most chapters end with suggestions for how to evaluate for no money at all, a little money, or money commensurate with the health, economic, and political stakes riding on the communications – framed so that even amateurs could do it.

(<http://www.fda.gov/AboutFDA/ReportsManualsForms/Reports/ucm268078.htm>)

The authors may disagree on this question. If so, I think that it would be appropriate to end the paper with a clear statement of that position, rather than ignoring it.

References:

The first of the references below is an empirical attempt to apply both Bayesian and non-Bayesian methods, recognizing that the authors do not want to get into the latter.

The second and third references are, I believe, empirical studies of some of the issues that the authors raise and might be worthy of citation.

Curley SP. 2007. The application of Dempster-Shafer theory demonstrated with justification provided by legal evidence. *Judgm. Dec. Making* 2(5):25-276.

Üklümen G, Fox CR, Malle BF. 2016. Two dimensions of subjective uncertainty: Clues from natural language. *J. Exp. Psychol.:Gen.* 145(10):1280-1297.

Walters DJ, Ferbach PM, Fox CR, Sloman SA. 2017. Known unknowns: A critical determinant of confidence and calibration. *Manag. Sci.* 63(12):4298-4307.

Baruch

Review form: Reviewer 4 (Andrew Przybylski)

Is the manuscript scientifically sound in its present form?

Yes

Are the interpretations and conclusions justified by the results?

Yes

Is the language acceptable?

Yes

Is it clear how to access all supporting data?

Not Applicable

Do you have any ethical concerns with this paper?

No

Have you any concerns about statistical analyses in this paper?

No

Recommendation?

Accept as is

Comments to the Author(s)

I am satisfied with these changes. Thank you for addressing my points 1 and 2 so directly. I believe the additions to the communication framework will also be helpful to readers.

Decision letter (RSOS-181870.R1)

01-Apr-2019

Dear Dr van der Bles:

On behalf of the Editors, I am pleased to inform you that your Manuscript RSOS-181870.R1 entitled "Communicating uncertainty about facts, numbers, and science" has been accepted for publication in Royal Society Open Science subject to minor revision in accordance with the referee suggestions. Please find the referees' comments at the end of this email.

The reviewers and Subject Editor have recommended publication, but also suggest some minor revisions to your manuscript. Therefore, I invite you to respond to the comments and revise your manuscript.

- Ethics statement

- Data accessibility

<http://datadryad.org/submit?journalID=RSOS&manu=RSOS-181870.R1>

- Competing interests

- Authors' contributions

All submissions, other than those with a single author, must include an Authors' Contributions section which individually lists the specific contribution of each author. The list of Authors

should meet all of the following criteria; 1) substantial contributions to conception and design, or acquisition of data, or analysis and interpretation of data; 2) drafting the article or revising it critically for important intellectual content; and 3) final approval of the version to be published.

- Acknowledgements

- Funding statement

Because the schedule for publication is very tight, it is a condition of publication that you submit the revised version of your manuscript before 10-Apr-2019. Please note that the revision deadline will expire at 00.00am on this date. If you do not think you will be able to meet this date please let me know immediately.

- 1) A text file of the manuscript (tex, txt, rtf, docx or doc), references, tables (including captions) and figure captions. Do not upload a PDF as your "Main Document".
- 2) A separate electronic file of each figure (EPS or print-quality PDF preferred (either format should be produced directly from original creation package), or original software format)
- 3) Included a 100 word media summary of your paper when requested at submission. Please ensure you have entered correct contact details (email, institution and telephone) in your user account

4) Included the raw data to support the claims made in your paper. You can either include your data as electronic supplementary material or upload to a repository and include the relevant doi within your manuscript

5) All supplementary materials accompanying an accepted article will be treated as in their final form. Note that the Royal Society will neither edit nor typeset supplementary material and it will be hosted as provided. Please ensure that the supplementary material includes the paper details where possible (authors, article title, journal name).

on behalf of Professor Chris Chambers (Associate Editor) and Professor Essi Viding (Subject Editor)
openscience@royalsociety.org

Associate Editor Comments to Author (Professor Chris Chambers):

The revised manuscript was returned to three of the original reviewers. All are broadly positive about the revision, however two reviewers point to several remaining issues to address, especially concerning justification and clarification of key arguments (including the conclusion). A final revision is therefore invited, and provided the next version of the manuscript thoroughly addresses (or rebuts) these points, full acceptance should be forthcoming without requiring further in-depth review.

Reviewer comments to Author:

Reviewer: 3

Comments to the Author(s)

I appreciate the authors' thoughtful, and interesting, responses to my review and those of the other reviewers. I gave such detailed comments, in part, because I believe this to be an important project, which I was hoping to help along its way, and, in part, because it did not flow for me, despite being locally very clear. I decided going through it slowly was the best way to think about what might be redone. This time, I just read for pleasure.

I have three residual comments, two minor and one major, along with a few references that might be of use.

Minor Comment 1: I think that the risks with indirect uncertainty practices (p. 20ff) are greater than the text implies, in cases where audience members are not privy to professional conventions.

The authors might consider the conventions that John Cohen introduced as part of his campaign to get psychologists to perform power analyses: small, medium, and large effect sizes. Those terms and their statistical equivalents reflected his intuition of what kind of results impressed psychologists and, I am guessing, what would make them feel good enough to adopt his methods. What do non-psychologists think if psychologists say, or act like, they have large effects – not knowing that it is large for psychologists. Similarly, do people interpret “high quality” as “high quality, given the challenges of clinical trials”?

Minor Comment 2: The authors repeatedly say that there is very little empirical research on these topics, but then cite a lot of it (witness the references to empirical papers). Indeed, in the spirit of their paper, it communicated to me that, on some topics, there is enough evidence for it to be inconclusive. They might revisit the wording on this topic,

Major Comment: I was disappointed not to see a strong concluding statement calling for the empirical evaluation of communications. To my mind, its absence substantially undermines the value of the entire article. The authors have demonstrated the dangers of the amateur-hour approach that is the norm in scientific communication. They have established that the empirical results are mixed on those issues where there have been studies. Their framework demonstrates the complexity of real-world communication. Their summary suggestions could be very helpful in designing communications that are worth testing. However, they only provide better guesses at what might work. If readers believe that following these suggestions will guarantee success, then they are being set up for failure. We structured the FDA guide so that evaluation is central to the effort and most chapters end with suggestions for how to evaluate for no money at all, a little money, or money commensurate with the health, economic, and political stakes riding on the communications – framed so that even amateurs could do it.
(<http://www.fda.gov/AboutFDA/ReportsManualsForms/Reports/ucm268078.htm>)

The authors may disagree on this question. If so, I think that it would be appropriate to end the paper with a clear statement of that position, rather than ignoring it.

References:

The first of the references below is an empirical attempt to apply both Bayesian and non-Bayesian methods, recognizing that the authors do not want to get into the latter.

The second and third references are, I believe, empirical studies of some of the issues that the authors raise and might be worthy of citation.

Curley SP. 2007. The application of Dempster-Shafer theory demonstrated with justification provided by legal evidence. *Judgm. Dec. Making* 2(5):25-276.

Üklümen G, Fox CR, Malle BF. 2016. Two dimensions of subjective uncertainty: Clues from natural language. *J. Exp. Psychol.:Gen.* 145(10):1280-1297.

Walters DJ, Ferbach PM, Fox CR, Sloman SA. 2017. Known unknowns: A critical determinant of confidence and calibration. *Manag. Sci.* 63(12):4298-4307.

Baruch

Reviewer: 2

Comments to the Author(s)
General Comments

I reiterate my conclusion from my prior review: this is an impressive, well-done, well-written

review. I believe the authors have carefully addressed my critiques and those of the other reviewers, and I commend their responsiveness overall. I do believe there are lingering problems, which mainly reflect my disagreement with some of the authors' conceptual decisions, which I outline below. There can certainly be legitimate disagreement on these issues, but I think some added consideration of these issues would strengthen the manuscript.

Specific Comments

1. Pp 3-5: I appreciate the authors' re-working of the introduction and their inclusion of more explicit discussion of the definition of uncertainty. However, I still have a problem with the term "future" uncertainty and its conceptual distinction from epistemic uncertainty. I maintain that as described by the authors, these are not mutually exclusive categories, and the term "future" just confuses matters. To me, this distinction is not useful and in fact misleading. Epistemic uncertainty can encompass issues pertaining to the future as well as the present and past. The issues comes down to definitions, or course, but I believe that temporality is not the factor that determines whether uncertainty is "epistemic" or not, and believe I'm not alone in this view.

Compounding the confusion, in my mind, is that the authors simultaneously introduce a dichotomy between unknowability (can't know) and unknownness (don't know), and place it alongside the distinction between future and epistemic uncertainty. This juxtaposition implies that the defining feature of "future" uncertainty is unknowability, while the defining feature of "epistemic" uncertainty is mere unknownness. This distinction, too, is problematic. Even if an aspect of reality is in principle knowable, one can still be uncertain if it is unknown for the time being – and this uncertainty is clearly epistemic (pertaining to one's knowledge). Furthermore, the author's conflation of all these issues also implies that uncertainties about the present and past are knowable – and this is clearly not true. There are many things about the present and past that we not only don't know, but can't know, for many reasons.

I think what the authors really want to do is to distinguish between uncertainty that arises from or pertains to limits to knowledge (epistemic) vs. the fundamental indeterminacy or randomness of the world – this 2nd uncertainty is captured by the term aleatory/aleatoric, which the authors used in the previous version. I would much prefer they use that term instead of the word "future": it is much less ambiguous and confusing, and more logically coherent and consistent with established terminology in the literature.

2. Pp 11-12: I appreciate the authors' attempt to rework the section previously entitled "type of uncertainty". They have now changed the focus to the idea of "levels of uncertainty" with 2 main ones: direct and indirect. However, I believe this change has created new conceptual difficulties that again are raised by the choice of specific words that have specific connotations. I believe what they are now referring to is a distinction that decision theorists since Knight (whom the authors cite) have described variously as 1st-order vs 2nd-order uncertainty, known vs unknown probabilities, risk vs. uncertainty, probability vs. ambiguity (Ellsberg). The essence of this distinction is a metacognitive reflexiveness: a thinking about thinking – in this case, an uncertainty about one's uncertainty. The distinction captures a secondary mental state of uncertainty focused back upon a prior uncertainty about some issue – in the authors' words, "Caveats" about the "quality of the underlying knowledge" regarding "facts, numbers, and hypotheses." The authors' example is a case of direct uncertainty as concerning "the absolute probability of guilt," which is compounded by indirect uncertainty concerning the "credibility to be given to an individual's testimony concerning this item of evidence." This distinction is conceptually equivalent to the existing, well-established distinction between 1st vs 2nd order uncertainty, or probability vs ambiguity; the underlying phenomenon boils down to higher-order uncertainty about uncertainty.

The question is which set of words is most useful to express a concept. To me, the words “direct” and “indirect” do not hit the mark with their connotations, and do not clarify but instead obscure the essence of the underlying phenomenon. To me, these words do not capture the concept of reflexivity, and don’t even match up to the notion of “levels.” It is certainly the authors’ prerogative to apply a new terminology, but then I believe they should acknowledge the other more established terms, and make explicit how their terms differ or not.

I also recognize that some scholars do not believe in 2nd-order uncertainty; Morgan and others, for example, have argued that the notion of uncertainty about uncertainty is incoherent and not useful – that everything is uncertainty, and it is not necessary to distinguish levels. I personally do not agree with this view in theory, and believe there is ample empirical evidence showing that people behave as if 2nd-order uncertainty did exist. However, if this is the theoretical rationale for the author’s choice of terms, then they should at least make it explicit and defend it.

3. P 15, table: a related fundamental disagreement I have with the authors is their treatment of representations of imprecision as expressions of 1st-order (what they call “direct”) rather than 2nd-order (“indirect”) uncertainty (“ambiguity” in Ellsberg’s terms). I believe, as do many decision theorists, that imprecision signifies 2nd-order uncertainty (uncertainty about uncertainty). From a statistical modeling standpoint, imprecision does manifest uncertainty in probability estimates; confidence intervals are wider when either the evidence used in estimating probabilities or our modeling methods themselves are more inadequate or unreliable. And from a psychological standpoint, imprecision is perceived as signifying uncertainty: people respond differently to probabilities when imprecision is expressed. They perceive imprecise ranges as more uncertain than point estimates, and display ambiguity aversion in response to them. For all these reasons I believe that representations of imprecision in probability estimates such as probability distributions and ranges do not belong in the same conceptual category as point estimates of probability.

4. P 20: for this reason I also disagree with the assertion that “Methods for communicating the quality of the underlying evidence do not give quantitative information about absolute values or facts...” I think probability distributions, risk ranges, and confidence intervals do just that, by signifying with quantitative precision the imprecision of our estimates. That is not to say they are ideal or necessary or sufficient representations of 2nd-order uncertainty, only that they constitute quantitative expressions of it.

5. p 25: again, I think the use of the term “future” uncertainty is misleading, and also do not agree with the assertion in lines 12-13 that “ambiguity aversion is mostly about people’s aversion to using this information for making decisions about a future event.” To me, that statement is much too strong. It’s true that the classic experimental paradigm for demonstrating ambiguity aversion has consisted of gambling experiments using balls and urns, where the outcome was the willingness to bet on an unknown probability. However, this phenomenon generalizes to all kinds of judgments and decisions as well as not only behavioral but cognitive and emotional responses, and is thought to specifically reflect the effects of epistemic uncertainty (2nd-order uncertainty about one’s uncertainty) arising from limitations in the reliability, credibility, or adequacy of one’s information. This distinguishes it from risk aversion. A large body of empirical evidence has also shown that ambiguity aversion also consists not only of decisions about a future event, but current risk perceptions, judgments, preferences, and emotional responses.

6. P 43: I appreciate and agree with the authors’ discussion at the bottom of the page on the difficulty of distinguishing between past, present, and future uncertainty, and think it would be good to discuss and highlight this important caveat much earlier in the paper.

Reviewer: 4

Comments to the Author(s)

I am satisfied with these changes. Thank you for addressing my points 1 and 2 so directly. I believe the additions to the communication framework will also be helpful to readers.

Author's Response to Decision Letter for (RSOS-181870.R1)

See Appendix D.

Decision letter (RSOS-181870.R2)

11-Apr-2019

Dear Dr van der Bles,

I am pleased to inform you that your manuscript entitled "Communicating uncertainty about facts, numbers, and science" is now accepted for publication in Royal Society Open Science.

on behalf of Professor Chris Chambers (Associate Editor) and Professor Essi Viding (Subject Editor)
openscience@royalsociety.org

Appendix A

Comments for *Royal Society Open Science* on van der Bles *et al*, “Communicating Uncertainty about Facts, Numbers, and Science”

Charles F. Manski, Northwestern University

General Comments

This paper usefully summarizes and interprets the state of knowledge regarding the subject conveyed by its title. To this reader, the most important remarks made in the paper occur in its section on Discussion and Conclusions, where the authors write

“To our knowledge, no systematic empirical research has been conducted about the effects of epistemic uncertainty communication on behaviour and decision-making.”

“Because of its wide-reaching effects, uncertainty communication should be an important issue for policy makers, experts, and scientists across many fields. At present, however, this appears to be a science in its infancy.”

These statements are accurate. I have reached similar conclusions as I have pursued my own research on communication of scientific uncertainty. It has pained me that the importance of these matters has not been appreciated widely. This paper has the opportunity to make a contribution by increasing awareness and helping to stimulate new research.

In addition to its summary and interpretation of the state of knowledge, the authors make an effort to organize thinking on the subject by proposing what they view as a novel framework based on what they call the “Laswell model of communication.” I did not find this framework innovative. To the extent that it is helpful, it simply expresses common sense. I found the presentation more discursive than necessary.

Specific Comments

1. I do not understand why the authors conflate aleatory uncertainty with uncertainty about the future. The term “aleatory” is generally used to mean “statistical” or “stochastic,” in the sense of frequentist statistics. Uncertainty about the future may have some such elements, but it may also have epistemic foundations.

2. The authors mention or at least cite a wide spectrum of relevant research. Perhaps inevitably, however, they miss some work that is directly relevant to the paper. I particularly have in mind three items:

Morgan, M and M. Henrion (1990), *Uncertainty: A Guide to Dealing with Uncertainty in Quantitative Risk and Policy Analysis* (Cambridge Univ Press, Cambridge, UK).

Fischhoff, B (2012) “Communicating uncertainty: Fulfilling the duty to inform,” *Issues in Science and Technology*, 28, 63–70.

Manski, C. (2018), "Communicating Uncertainty in Policy Analysis," *Proceedings of the National Academy of Sciences*, <https://doi.org/10.1073/pnas.1722389115>.

3. The authors reference Wynne for the term “indeterminacy” to mean uncertainty about scientific knowledge. Multiple other terms are used across various disciplines. Some scientists may refer to “deep uncertainty” or “model uncertainty.” There exists an extensive literature in econometrics on “partial identification.”

4. The authors define epistemic uncertainty as

“uncertainty due to lack of knowledge about current and past facts, numbers, or scientific models and hypotheses – all of which are, at least in theory, verifiable or falsifiable.”

It is not correct that such knowledge is necessarily verifiable or falsifiable, even in theory. In particular, knowledge regarding counterfactual events is logically not verifiable or falsifiable. The distinction between models/hypotheses that are and are not falsifiable has been important in econometric research on partial identification, which has used the terms “refutable” or “testable.” See the textbook exposition of

Manski, C. (2007), *Identification for Prediction and Decision*, Harvard University Press.

The index directs one to multiple discussions of “nonrefutable assumptions” and “refutable assumptions.”

5. The authors write that confidence in a scientific model is “something not readily reduced to a numerical expression.” Bayesian researchers perform what they call “Bayesian model averaging,” which places subjective probabilities on the correctness of alternative models. I think there is often reason to be critical of these exercises, but Bayesians find them useful. It would be good for the paper to address the question in some manner.

6. The authors cite in a positive manner the qualitative methods used by the GRADE system for evaluating knowledge regarding medical interventions. I suggest that some skepticism is warranted. There may exist substantial interpersonal differences in the way that the members of a GRADE panel interpret the verbal expressions that the system uses. It is not clear what emerges when the system aggregates these expressions through the voting system that it uses.

Later in the paper, Box 4 calls attention to the problems that can arise with verbal descriptors of uncertainty. I think that Box 4 is right on target and that these problems apply to GRADE.

RSOS-181870

General comments

This is an impressive and well-done, well-written review of the topic of uncertainty communication. It is a welcome addition to the field and will be useful to diverse readers. The authors have synthesized a large body of evidence and done an admirable job pulling out key themes and also identifying important knowledge gaps. I have only a few comments and suggestions on conceptual issues that I believe the authors could address to strengthen the paper. It is an ambitious paper, and that is also its challenge. It tends to gloss over whole literatures—an unavoidable problem with any attempt at synthesis, but the authors acknowledge this issue and do an admirable job. Nevertheless, I believe there are issues that the authors could clarify and areas where a little more attention would strengthen the paper.

Specific comments

Pp 3-5: The authors rightly begin the paper discussing definitions, and acknowledge that the term “uncertainty” is “used in a myriad of ways” and proceed to mention a few examples of the conceptual confusion and imprecision of terms that are out there—e.g., in Wynne’s distinction. However, they do not come down off the fence to really define the term itself. They proceed directly to trying to define a “more general framework” or “structure.” I believe that their analysis would be strengthened if they came down off the fence and actually suggested a provisional definition of the term uncertainty.

P 5, top: related to this, I find the authors’ distinction between future uncertainty and epistemic uncertainty problematic. One could argue that future uncertainty is epistemic in nature, as well as being traceable to indeterminacy and inherent unpredictability. I think they need to defend this distinction; I believe it is somewhat unusual and stems from a lack of clarity on what is meant by “uncertainty” in the first place.

P 5: I like the tie to the Lasswell communication model—it makes for a nice orienting structure for conceptualizing the problem of communicating uncertainty

P 9: I believe the discussion of sources of uncertainty, as the authors note, could be further expanded, but also collapsed within higher-order concepts. The question is, what is the right level of complexity of a conceptual framework. Again, it also ties back to how the authors are defining “uncertainty” in the first place, what is the intended use of the framework, and also whether there are other objective reasons for discriminating between different sources—that is, is there empirical evidence that different sources of uncertainty have different effects when communicated. Here I think they could also reference other published conceptual taxonomies of ignorance, uncertainty, and scientific uncertainty put forth by investigators from different disciplinary perspectives, and that have divided these concepts in different ways. In what sense is the authors’ framework more or less useful or conceptually justified? I believe some further discussion of this question is needed.

p. 10: I found the discussion at the top of this page hard to follow. I’m confused about what is meant by “objects” of uncertainty, could this be defined more clearly? In the top para it seems like “what the coin shows” is what is meant by object, but then this could be more precisely labelled as “actual outcome” or state of reality. In the 2nd para it seems like “object” refers to the numerical representation of uncertainty, that is the “number 1/2”. And in the 3rd para it is the “scientific model.” Another confusing issues here is that these distinctions conflate sources of uncertainty. That is, the “number” (object 2)

integrates epistemic uncertainty from ambiguity in decision theory terms (uncertainty arising from limitations in the evidence at hand). I'm not certain about the usefulness of introducing this other dimension of "objects" of uncertainty, and think this section needs to be expanded and explained more clearly.

p. 10 middle: another semantic and conceptual problem lies in the author's category of "type" of uncertainty. Here it would again be helpful to clarify what a word means. To me, the distinction between absolute and relative uncertainty does not signify different "types" of uncertainty as much as different representations of it: they are simply different ways of operationalizing, using different summary statistics, the same underlying uncertainty. To me, this makes them different representations, not different types: these are just semantic distinctions but they need to be made clearer by defining basic terms more precisely. Some further discussion of this issue would clarify this section.

P 12: I like the list of different forms or expressions. I think it could also be useful to put the list under Type A into a table where it is also indicated somehow that they are falling on a continuum of precision or explicitness of expression.

Here again, though, I think the authors have put forth conceptual distinctions that are debatable, and that would benefit from more clarification. To me, the terms "absolute" and "relative" signify something different than what the authors are getting at. Under "absolute" they are talking about estimates of the magnitude of uncertainty, which they explicitly acknowledge. To me, however, it seems odd to equate "weight of evidence" with the term "relative." It may be true that from the standpoint of statistical hypothesis testing, one can use p values or other summary statistics to operationalize the concept of weight of evidence. However, from a conceptual standpoint, it does not make sense to call weight of evidence a relative phenomenon rather than absolute. To me, weight of evidence is an added, higher-order of with its own sources, and that has the ultimate effect of qualifying the magnitude of one's uncertainty estimates themselves, and making us rethink our degree of belief in them. It does not matter if the statistical methodology used to represent weight of evidence depends on some relative comparison. The deeper meaning of "weight of evidence" is that there is a higher level of uncertainty that has to do with the adequacy of our estimates of the magnitude of (lower-order) uncertainty.

Again, this may be a semantic issue, but it is important because words have connotations. I think the precision of the authors' use of these concepts could be increased, and that additional explanation/justification of the current scheme would help the paper.

Pp 14-18: I like the table and examples, very helpful.

P 26: I'm a little confused about the distinction between "scientific uncertainty and the (psychological) effects of communicating this..." Is there a psychological distinction between the psychological effects of communicating scientific vs. other forms of uncertainty? This is an empirical question, but I believe that the empirical literature suggests that many effects of uncertainty are not domain-specific. In any case, the authors should clarify this assertion and its justifications. It serves the instrumental value for the authors of confining the scope of their already ambitious review and making it more tractable, but it is an artificial barrier.

I also do not agree with the author's conceptual equation of epistemic uncertainty with uncertainty about the "past and present", and its distinction from aleatoric uncertainty (and the equation of the

latter with uncertainty about the future). I think the authors need to defend this conceptualization, as it departs from other thinking and writing on this subject. If the authors choose to maintain the underlying distinction (and there may be good reasons to do so), I would recommend abandoning the terms epistemic and aleatory and just simply saying past/present vs. future as the focus of uncertainty. This comment applies elsewhere in the paper, where this distinction is made, and particularly in the discussion and conclusion section on pp 43-44.

Pp 35 onward: I very much like the case studies, very helpful.

Appendix C

UNIVERSITY OF
CAMBRIDGE

Winton Centre for
Risk and Evidence Communication

16th February 2019,

Dear Prof Chambers,

Thank you very much indeed for your letter of 25th January regarding the manuscript **“Communicating uncertainty about facts, numbers, and science”**. We would also like to pass on our thanks to all four reviewers of the article. We found their comments exceptionally helpful and insightful and have reworked the manuscript in the light of them.

Because our revisions to the manuscript have been so extensive we have not highlighted changes as is often customary to do, but below we detail our response to and actions we have taken as a result of each of the points made by each of the reviewers. We are happy for these responses to be shared with the reviewers.

We feel that the revisions in the light of the reviewers have made significant improvements to the manuscript and hope that you now feel that it is suitable for publication. Do not hesitate to come back to us with any further comments or queries and we will respond as quickly as possible.

Yours sincerely,

Dr Anne Marthe van der Bles
Dr Sander van der Linden
Dr Alexandra Freeman
Prof James Mitchell
Prof Ana Galvao
Dr Lisa Zaval
Prof David Spiegelhalter

Reviewer 1 (Charles F. Manski)

General Comments

This paper usefully summarizes and interprets the state of knowledge regarding the subject conveyed by its title. To this reader, the most important remarks made in the paper occur in its section on Discussion and Conclusions, where the authors write

“To our knowledge, no systematic empirical research has been conducted about the effects of epistemic uncertainty communication on behaviour and decision-making.”

“Because of its wide-reaching effects, uncertainty communication should be an important issue for policy makers, experts, and scientists across many fields. At present, however, this appears to be a science in its infancy.”

These statements are accurate. I have reached similar conclusions as I have pursued my own research on communication of scientific uncertainty. It has pained me that the importance of these matters has not been appreciated widely. This paper has the opportunity to make a contribution by increasing awareness and helping to stimulate new research.

We are very glad to hear that Prof Manski shares our mission and in this revision we have emphasised that work by others (including Manski 2018, PNAS) reaches similar conclusions.

In addition to its summary and interpretation of the state of knowledge, the authors make an effort to organize thinking on the subject by proposing what they view as a novel framework based on what they call the “Laswell model of communication.” I did not find this framework innovative. To the extent that it is helpful, it simply expresses common sense. I found the presentation more discursive than necessary.

The ‘Laswell model of communication’ is indeed an expression of ‘common sense’ (he himself did not perceive it as being novel, merely helpful), and we would certainly not describe our use of it to help organise thinking around the communication of uncertainty as ‘innovative’. We have rewritten the Abstract and Introduction to be careful to describe our intentions with the work:

“In this paper we present a cohesive framework that aims to provide clarity and structure to the issues surrounding such communication, combining a statistical approach to quantifying uncertainty, with a psychological perspective that stresses the importance of the effects of communication on the audience. Our aim is to provide guidance on how best to communicate uncertainty honestly and transparently without losing trust and credibility, to the benefit of everyone who subsequently uses the information to form an opinion or make a decision.”

And

“Based on an interdisciplinary approach, we have developed an overarching framework that clarifies the components that make up the umbrella term ‘uncertainty’ and those that comprise the process of communication, affecting an audience’s reaction to uncertainty communication (see Figure 1). This provides a structure both for understanding what is done and what might be investigated in the future. The framework comprises several novel and hopefully useful elements:

- The identification of three *objects* of uncertainty: categorical (facts), continuous (numbers) and hypothetical (theories or scientific models that describe the world) (Section 2.1).
- The distinction between two *levels* of uncertainty: direct uncertainty about specific facts, numbers and science (both absolute and relative), and indirect uncertainty: the quality of our underlying knowledge (Section 2.3).

- A list of alternative expressions for direct uncertainty based on an analysis of practice across many disciplines (Section 3.1 and Figure 2).”

We hope that this makes clear the purpose of this work in reviewing and bringing together work that has been scattered across many different domains and disciplines into a cohesive framework that is designed to be practical and useful for both communicators and researchers. We also believe that this framework has some novel elements, which we specifically highlight.

Our writing style may be more discursive than is normal in academic writing. This was a deliberate decision because we are writing for readers from multiple disciplines, as well as non-academics (professional communicators), and so find ourselves needing to avoid terminology that can be misunderstood. However, we have edited and shortened the manuscript in the light of all the reviewers’ comments and hope that this has made it easier to digest.

Specific Comments

1. I do not understand why the authors conflate aleatory uncertainty with uncertainty about the future. The term “aleatory” is generally used to mean “statistical” or “stochastic,” in the sense of frequentist statistics. Uncertainty about the future may have some such elements, but it may also have epistemic foundations.

In the light of the comments from all reviewers we have now started the manuscript with a definition of the terms we use:

“Our unavoidable uncertainty about the future is characterised by what we *can’t know* for certain, often couched in terms of luck or chance. In contrast, our uncertainty about the world around us is characterised by what we *don’t know* for certain. This is due to limited knowledge or ignorance, and is the focus of this paper. We shall use the term ‘epistemic uncertainty’ for uncertainty about facts, numbers or science that arises because of limits to our knowledge about them; when it is knowledge that we could have, at least in principle, but in practice we do not.”

And we add the footnote to this:

“We therefore do not consider concepts that are neither in the future nor theoretically knowable, such as non-identifiable parameters in statistical models, knowledge about counterfactual events or the existence of God. We refer the reader to Manski (2007) for a discussion of “nonrefutable” and “refutable” (or testable) assumptions in econometrics.”

2. The authors mention or at least cite a wide spectrum of relevant research. Perhaps inevitably, however, they miss some work that is directly relevant to the paper. I particularly have in mind three items:

Morgan, M and M. Henrion (1990), *Uncertainty: A Guide to Dealing with Uncertainty in Quantitative Risk and Policy Analysis* (Cambridge Univ Press, Cambridge, UK).

Fischhoff, B (2012) “Communicating uncertainty: Fulfilling the duty to inform,” *Issues in Science and Technology*, 28, 63–70.

Manski, C. (2018), “Communicating Uncertainty in Policy Analysis,” *Proceedings of the National Academy of Sciences*, <https://doi.org/10.1073/pnas.1722389115>.

We are grateful for having these references drawn to our attention. We have incorporated the last into the manuscript. The others we appreciate as excellent commentaries on the topic, but have not found them necessary to reference.

3. The authors reference Wynne for the term “indeterminacy” to mean uncertainty about scientific knowledge. Multiple other terms are used across various disciplines. Some scientists may refer to “deep uncertainty” or “model uncertainty.” There exists an extensive literature in econometrics on “partial identification.”

We have added more to the manuscript acknowledging the breadth of terms used across different disciplines. We have added a section in the introduction on ‘other frameworks’ mentioning work done and terms used in several fields, have now added footnotes in the introduction referring to the work of Manski on partial identification and a reference to ‘deep uncertainty’ in section 2.3; and discuss partial identification and the “bounds” work of Manski in section 2.2 and in the case study on economics statistics.

4. The authors define epistemic uncertainty as “uncertainty due to lack of knowledge about current and past facts, numbers, or scientific models and hypotheses – all of which are, at least in theory, verifiable or falsifiable.” It is not correct that such knowledge is necessarily verifiable or falsifiable, even in theory. In particular, knowledge regarding counterfactual events is logically not verifiable or falsifiable. The distinction between models/hypotheses that are and are not falsifiable has been important in econometric research on partial identification, which has used the terms “refutable” or “testable.” See the textbook exposition of

Manski, C. (2007), Identification for Prediction and Decision, Harvard University Press.

The index directs one to multiple discussions of “nonrefutable assumptions” and “refutable assumptions.”

We believe our response above to point 1 covers this point as well.

5. The authors write that confidence in a scientific model is “something not readily reduced to a numerical expression.” Bayesian researchers perform what they call “Bayesian model averaging,” which places subjective probabilities on the correctness of alternative models. I think there is often reason to be critical of these exercises, but Bayesians find them useful. It would be good for the paper to address the question in some manner.

We have added a footnote at this point:

“However, Bayesian researchers perform “Bayesian model averaging” which places subjective probabilities on the correctness of alternative, candidate scientific models; see the Technical Appendix for further discussion.”

In the Technical appendix we say:

“Bayes theorem can also be applied across competing scientific models, in the face of what is often called “model uncertainty”: Bayesian Model Averaging involves treating the set of models, S , as an additional parameter and then integrating over S ; e.g. see Draper (1995).”

6. The authors cite in a positive manner the qualitative methods used by the GRADE system for evaluating knowledge regarding medical interventions. I suggest that some skepticism is warranted. There may exist substantial interpersonal differences in the way that the members of a GRADE panel

interpret the verbal expressions that the system uses. It is not clear what emerges when the system aggregates these expressions through the voting system that it uses. Later in the paper, Box 4 calls attention to the problems that can arise with verbal descriptors of uncertainty. I think that Box 4 is right on target and that these problems apply to GRADE.

We agree. In our previous version of the manuscript we had unwittingly conflated the checklist systems used to evaluate uncertainties in underlying evidence (such as CONSORT) with the communication schemes devised by organisations such as GRADE to communicate the results of these evaluations. We have firstly separated these two components (as suggested by another reviewer), and we have also checked our wording to ensure that it is purely descriptive and does not offer any opinion on the effectiveness of either component.

Reviewer: 2

General comments

This is an impressive and well-done, well-written review of the topic of uncertainty communication. It is a welcome addition to the field and will be useful to diverse readers. The authors have synthesized a large body of evidence and done an admirable job pulling out key themes and also identifying important knowledge gaps. I have only a few comments and suggestions on conceptual issues that I believe the authors could address to strengthen the paper. It is an ambitious paper, and that is also its challenge. It tends to gloss over whole literatures—an unavoidable problem with any attempt at synthesis, but the authors acknowledge this issue and do an admirable job. Nevertheless, I believe there are issues that the authors could clarify and areas where a little more attention would strengthen the paper.

We thank the reviewer very much for their positive words about our paper and helpful comments below.

Specific comments

Pp 3-5: The authors rightly begin the paper discussing definitions, and acknowledge that the term “uncertainty” is “used in a myriad of ways” and proceed to mention a few examples of the conceptual confusion and imprecision of terms that are out there—e.g., in Wynne’s distinction. However, they do not come down off the fence to really define the term itself. They proceed directly to trying to define a “more general framework” or “structure.” I believe that their analysis would be strengthened if they came down off the fence and actually suggested a provisional definition of the term uncertainty.

P 5, top: related to this, I find the authors’ distinction between future uncertainty and epistemic uncertainty problematic. One could argue that future uncertainty is epistemic in nature, as well as being traceable to indeterminacy and inherent unpredictability. I think they need to defend this distinction; I believe it is somewhat unusual and stems from a lack of clarity on what is meant by “uncertainty” in the first place.

We entirely agree, and as mentioned in response to Prof Manski’s review (point 1) we have added definitions of each of the terms at the start of the Introduction:

“Uncertainty: a situation in which something is not known, or something that is not known or certain (Cambridge Dictionary)

Uncertainty is all-pervasive in the world, and we regularly communicate this in everyday life. We might say we are uncertain when we are unable to predict the future, we cannot decide what to do, there is ambiguity about what something means, we are ignorant of what has happened, or simply for a general feeling of doubt or unease. The broad definition above from the Cambridge dictionary reflects these myriad ways the term ‘uncertainty’ is used in normal speech.

In the scientific context, a vast literature has focused on uncertainty about the future, represented by research on the assessment, communication and management of both quantifiable and unquantifiable risks. Prominent examples include uncertain economic forecasts, climate change models, and actuarial survival curves. Our unavoidable uncertainty about the future is characterised by what we *can’t know* for certain, often couched in terms of luck or chance.

In contrast, our uncertainty about the world around us is characterised by what we *don’t know* for certain. This is due to limited knowledge or ignorance, and is the focus of this paper. We shall use the term ‘epistemic uncertainty’ for uncertainty about facts, numbers or science that arises because

of limits to our knowledge about them; when it is knowledge that we could have, at least in principle, but in practice we do not. Such epistemic uncertainty is an integral part of every stage of the scientific process: from the assumptions we have, the observations we note, to the extrapolations and the generalisations that we make. This means that all knowledge on which decisions and policies are based — from medical evidence to government statistics — is shrouded with epistemic uncertainty of different types and degrees.”

P 5: I like the tie to the Lasswell communication model—it makes for a nice orienting structure for conceptualizing the problem of communicating uncertainty

We thank the reviewer!

P 9: I believe the discussion of sources of uncertainty, as the authors note, could be further expanded, but also collapsed within higher-order concepts. The question is, what is the right level of complexity of a conceptual framework. Again, it also ties back to how the authors are defining “uncertainty” in the first place, what is the intended use of the framework, and also whether there are other objective reasons for discriminating between different sources—that is, is there empirical evidence that different sources of uncertainty have different effects when communicated. Here I think they could also reference other published conceptual taxonomies of ignorance, uncertainty, and scientific uncertainty put forth by investigators from different disciplinary perspectives, and that have divided these concepts in different ways. In what sense is the authors’ framework more or less useful or conceptually justified? I believe some further discussion of this question is needed.

We thank the reviewer for this thoughtful point. We have added definitions (as discussed above) and a section in the introduction discussing other frameworks and that “In spite of all this activity, no general consensus has emerged as to a general framework, perhaps due to the wide variety of contexts and tasks being considered, and the complexity of many of the proposals. Our structure, with its more restricted aim of communicating epistemic uncertainty, attempts to be a pragmatic cross-disciplinary compromise between applicability and generality. The individual elements of it are those factors which we believe (either through direct empirical evidence or suggestive evidence from other fields) could affect the communication of uncertainty and thus should be considered individually.”

We also state more directly the purpose of Table 1 as a partial test of the practicality of part of our framework: “In order to explore whether each in this list of 9 expressions of absolute, direct uncertainty could be applied to all three objects of uncertainty in our framework: categorical or binary *facts*, continuous variables (*numbers*) and *models* we set out to find real examples of each in use. The result of our search is shown in Table 1. We were not able to find examples for each cell in the table, illustrating where some usages are rare at best. However, our intention was both to test the comprehensiveness of our framework and to illustrate it to help others identify how it can be applied.”

The empirical evidence on the degree to which each of the different elements that we identify in our framework affects the communication is so far lacking, and so it is inevitably at this point based on informed speculation of which elements we might consider to have a role, and hence need further study. We hope to stimulate that further study.

p. 10: I found the discussion at the top of this page hard to follow. I’m confused about what is meant by “objects” of uncertainty, could this be defined more clearly? In the top para it seems like “what the coin shows” is what is meant by object, but then this could be more precisely labelled as “actual outcome” or state of reality. In the 2nd para it seems like “object” refers to the numerical representation of uncertainty, that is the “number 1/2”. And in the 3rd para it is the “scientific model.” Another confusing issues here is that these distinctions conflate sources of uncertainty. That

is, the “number” (object 2) integrates epistemic uncertainty from ambiguity in decision theory terms (uncertainty arising from limitations in the evidence at hand). I’m not certain about the usefulness of introducing this other dimension of “objects” of uncertainty, and think this section needs to be expanded and explained more clearly.

We have now moved the coin example up into the previous section (Section 2.1) on ‘objects’ of uncertainty to make it clear that it is illustrating the concept of there being three different kinds of ‘objects’ of uncertainty as we define in that section.

p. 10 middle: another semantic and conceptual problem lies in the author’s category of “type” of uncertainty. Here it would again be helpful to clarify what a word means. To me, the distinction between absolute and relative uncertainty does not signify different “types” of uncertainty as much as different representations of it: they are simply different ways of operationalizing, using different summary statistics, the same underlying uncertainty. To me, this makes them different representations, not different types: these are just semantic distinctions but they need to be made clearer by defining basic terms more precisely. Some further discussion of this issue would clarify this section.

We agree that this was unclear and have changed and clarified our classification in section 2.3. We have removed the ‘three types’ and instead simplified it to ‘two levels’ (with the direct uncertainty having two possible forms of expression: absolute and relative) as suggested by the reviewer.

P 12: I like the list of different forms or expressions. I think it could also be useful to put the list under Type A into a table where it is also indicated somehow that they are falling on a continuum of precision or explicitness of expression.

We have followed the reviewer’s suggestion and placed an arrow next to the list (now Fig 2) to indicate decreasing precision.

Here again, though, I think the authors have put forth conceptual distinctions that are debatable, and that would benefit from more clarification. To me, the terms “absolute” and “relative” signify something different than what the authors are getting at. Under “absolute” they are talking about estimates of the magnitude of uncertainty, which they explicitly acknowledge. To me, however, it seems odd to equate “weight of evidence” with the term “relative.” It may be true that from the standpoint of statistical hypothesis testing, one can use p values or other summary statistics to operationalize the concept of weight of evidence. However, from a conceptual standpoint, it does not make sense to call weight of evidence a relative phenomenon rather than absolute. To me, weight of evidence is an added, higher-order of with its own sources, and that has the ultimate effect of qualifying the magnitude of one’s uncertainty estimates themselves, and making us rethink our degree of belief in them. It does not matter if the statistical methodology used to represent weight of evidence depends on some relative comparison. The deeper meaning of “weight of evidence” is that there is a higher level of uncertainty that has to do with the adequacy of our estimates of the magnitude of (lower-order) uncertainty.

Again, this may be a semantic issue, but it is important because words have connotations. I think the precision of the authors’ use of these concepts could be increased, and that additional explanation/justification of the current scheme would help the paper.

We agree, and have clarified this. We have removed our three ‘Types’ of uncertainty and instead used a simpler structure with two ‘levels’, and we have removed the use of the phrase ‘weight of evidence’ which has different meanings in different domains and was therefore potentially confusing.

Pp 14-18: I like the table and examples, very helpful.

P 26: I'm a little confused about the distinction between "scientific uncertainty and the (psychological) effects of communicating this..." Is there a psychological distinction between the psychological effects of communicating scientific vs. other forms of uncertainty? This is an empirical question, but I believe that the empirical literature suggests that many effects of uncertainty are not domain-specific. In any case, the authors should clarify this assertion and its justifications. It serves the instrumental value for the authors of confining the scope of their already ambitious review and making it more tractable, but it is an artificial barrier.

We apologise – we must have been unclear in what we meant by the distinction, and have tried to make this both clearer and briefer. We did not mean that there was a psychological distinction between the effects of communicating scientific or 'non scientific' forms of uncertainty, but that there was a difference between the uncertainties that we consider the objects being communicated and 'uncertainty' as a subjective emotion that might be evoked as a result of communication (and which psychologists often refer to as 'uncertainty' and on which there is a very large literature). We have clarified the wording:

"First, it is important to distinguish epistemic or scientific uncertainty from the subjective psychological experience of uncertainty – the feeling which might be the result of an ambiguous communication. Psychological uncertainty is a human experience, usually defined as an aversive psychological state in which an individual lacks information. In other words, it describes the subjective feeling of "not knowing". The psychological experience of uncertainty has been extensively investigated: the fact that people are averse to ambiguous information has been referred to as "one of the most robust phenomena in the decision-making literature". That is not the subject of our reviewing; we focus on uncertainty that is the property of a fact, number, or model that is being communicated."

I also do not agree with the author's conceptual equation of epistemic uncertainty with uncertainty about the "past and present", and its distinction from aleatoric uncertainty (and the equation of the latter with uncertainty about the future). I think the authors need to defend this conceptualization, as it departs from other thinking and writing on this subject. If the authors choose to maintain the underlying distinction (and there may be good reasons to do so), I would recommend abandoning the terms epistemic and aleatory and just simply saying past/present vs. future as the focus of uncertainty. This comment applies elsewhere in the paper, where this distinction is made, and particularly in the discussion and conclusion section on pp 43-44.

We considered the reviewer's suggestion of abandoning the term 'epistemic' but felt that 'uncertainty about the past and present' would become unwieldy when used throughout the manuscript, so instead have chosen to set out our definitions at the very start of the paper and hope that this clarifies our meaning throughout.

Pp 35 onward: I very much like the case studies, very helpful.

We are grateful for the positive comments about the tables and case studies!

Reviewer: 3 (Baruch Fischhoff)

I believe that this is an important article. Hoping to help it along its way, I started to make detailed comments on the paper. After a while, though, I developed an overriding concern, namely, that it is hard to take in all that the authors are offering. That concern left me thinking that I might serve this worthy cause best by offering some strategic thoughts on the exposition:

I often found the style slow-paced. I believe that was done in the interests of making it accessible to a broad, non-specialist audience. However, with such a long paper, that style increases the commitment from readers. Here, I think that some ruthless tightening could help.

We agree with these comments and have tightened throughout, especially sections which are not the main focus of the paper (discussed in more detail later), and have indeed now highlighted the sections that we focus on in Fig 1.

The authors have framed the paper in terms of a task analysis (Lasswell) which recognizes the complexity of communication processes. They have further elaborated it by drawing on their own ability to provide epistemological depth in areas that others have often glossed over (deepening Lasswell's "what"). However, that complexity makes it hard to get a big picture. The authors' use of exhibits should make a published version easier to navigate than the present manuscript version — where the exhibits break up the text, rather than being objects whose gist a reader could get, amplifying the text, then return to later if interested in the detail. I am afraid that the authors will lose portions of their audience unless they can find a better way of presenting the work. I have a radical suggestion at the end of these general comments.

We very much appreciate Prof Fischhoff's deeply thoughtful approach to our manuscript. We agree that careful print layout should help the readability of the paper, moving the Boxes and Tables out of the main flow of the text. We take on board the serious concerns that the manuscript was too long and wordy. We have not only edited it for style (trying to keep it suitable for a general readership but tightening throughout) but have also focussed the manuscript around the areas where we have concentrated our work — highlighting on Fig 1 the three parts of the Lasswell model where we believe the paper's contributions particularly lie, and shortening the other sections.

The authors have extensive, eclectic references to other conceptual schemes, which will be a valuable resource in themselves. However, they often treat the content of those references very briefly, seemingly dismissing their predecessors without serious attention to those efforts. One good reason is that, as the authors note, earlier work is somewhat chaotic and often incoherent, making it hard to treat systematically. As a result, it often feels as though they are giving their predecessors short shrift, for a scholarly publication. An alternative approach would be to begin each section with their own proposal (as happens with some sections), then end by briefly putting earlier approaches in that context.

This is a fair comment. We have now added a section in the introduction which specifically addresses in more depth some of the other models of uncertainty that previous scholars have devised. However, as Prof Fischhoff acknowledges, these are generally not attempting the same task as we are in this paper and so we end with "In spite of all this activity, no general consensus has emerged as to a general framework, perhaps due to the wide variety of contexts and tasks being considered, and the complexity of many of the proposals. Our structure, with its more restricted aim of communicating epistemic uncertainty, attempts to be a pragmatic cross-disciplinary compromise between applicability and generality."

It may still seem that we are giving our predecessors short shrift as we admittedly still do treat all previous frameworks briefly, but this is because we do not feel that any have been attempting to do the same task as we are, and that reviewing them in greater depth will only add to the length of the manuscript without adding greater understanding or changing its conclusions. We do apologise if this seems harsh treatment for authors who have each done excellent work with different aims.

There are some sections that seem just to bookmark topics in Lasswell's scheme, without reviewing the research in any depth or connecting the topic with the authors' own contribution.

Indeed, this is true and we now address this head-on by specifically highlighting in Fig 1 the sections we concentrate on and shortening all the others.

My overall sense is that there are several papers lurking here, which sometimes end up working at cross purposes:

- 1. A motivating essay on the importance of communicating uncertainty well and the perils of doing so poorly, emphasizing the common features across domains.**
- 2. An essay on the complexity of communication processes, as represented (and known since) the venerable Lasswell task analysis, and how failure in any component can imperil the entire effort.**
- 3. The authors' contribution to what might be the hardest part of the effort, characterizing uncertainty (Sections 2 and 3.1), showing how to get it right.**
- 4. A review of synthetic representations of uncertainty and how well they are understood (Section 3.2 and Section 5), interpreted in light of that scheme.**
- 5. A critique of inept schemes (e.g., IARC, Box 4, some of the cells in Table 1, the case studies).**

My recommendation is to write these articles separately, which would also free the authors from having to say something about each element in the Lasswell scheme, which leads to some unsatisfactory passages in this manuscript — while still giving Lasswell his due, in a field grown so large that people naturally focus on pieces. It would lead to dropping some things, for later, fuller treatment in a book on the topic. For this setting, I would briefly note topics 1 and 2, then focus on 3, followed by 4, emphasizing the conceptual scheme in 3. I realize that this is a presumptuous suggestion. I also suspect that the authors exhausted at having gotten everything out in one place. (I would be.) If they chose to see the current article through to publication, it would be a fine contribution. However, I think that it would fall short of the more definitive, accessible piece that they have in them, which could structure future work in the area.

We are extremely grateful to Prof Fischhoff for his careful analysis of our attempts with this manuscript, and he has – of course – seen exactly our intentions and our failings. We feel that 1, 2 & 5 are indeed worthy aims in their own right, and note that Prof Fischhoff has written very eloquent articles already on these topics, but agree that our aim in this manuscript was to concentrate on 3 and 4 – forming a conceptual understanding of the elements of uncertainty communication in the light of the current state of empirical knowledge of its effects on human psychology – with both feeding into each other. The framework was affected by the empirical knowledge and our review of the empirical knowledge then structured by the framework we had developed.

Prof Fischhoff's suggestion of separating the elements is not at all presumptuous, and he may well be right that the original manuscript was too big to serve its purpose. We have taken his suggestions to heart, and have tried to shorten the parts that are less core to the aims of the paper. We have kept in our critiques of inept schemes because we feel that they are important for professional communicators to read about, and that since they are in the form of Boxes/Tables/Case Studies they will – we hope – be more separated from the main text in print layout and hence not disrupt the main narrative so much as at present.

Page 2

The Abstract seems to focus more on what others haven't done than on what the authors have done. Could it be rewritten to provide the gist of their scheme (recognizing that it takes many pages to do it fuller justice).

2/8 "relatively little is known" seems at odds with what follows. Perhaps something along the lines of "what is known is widely scattered"

2/12. Wouldn't analysis be necessary for communication? It often isn't. However, I think that it would be better to frame the paper as following accepted practices, whose application is non-trivial.

2/20 "has not been systematic" is arguably true, but seems to be either vague or faulting others for not following a system that has just been introduced.

2/23 The final sentence doesn't add much. What would the authors like to have happen next?

We have rewritten the abstract bearing these comments in mind. We have emphasised the cross-disciplinary nature of this work as one of its important features and drawn attention more to the important conceptual points of our framework, as well as ending with a sentence drawing attention to the (new) boxes that summarise our guidance for practitioners and researchers in the field:

"Uncertainty is an inherent part of knowledge, and yet in an era of contested expertise, many shy away from openly communicating their uncertainty about what they know, fearful of their audience's reaction. But what effect does communication of such epistemic uncertainty have? Empirical research is widely scattered across many disciplines. This interdisciplinary review structures and summarises current practice and research across domains, combining a statistical and psychological perspective. Within a framework for uncertainty communication, we identify three objects of uncertainty - facts, numbers, and science - and two levels of uncertainty: direct and indirect. An examination of current practices provides a scale of nine expressions of direct uncertainty. We discuss attempts to codify indirect uncertainty in terms of the quality of the underlying evidence. We review the limited literature about the effects of communicating epistemic uncertainty on cognition, affect, trust, and decision-making. While there is some evidence that communicating epistemic uncertainty does not necessarily affect audiences negatively, impact can vary between individuals and communication formats. Case studies in economic statistics and climate change illustrate our framework in action. We conclude with advice to guide both communicators and future researchers in this important but so far rather neglected field."

Page 3

Given the strong suspicion in the US (at least among most of the authors' likely readers) that the books were cooked on Iraq, this does not seem like an effective example, even if Butler is held in high esteem in the UK. ("Vice" has just opened in theaters here.) The authors might consider Sherman Kent's classic "Words of Estimative Probability" (<https://www.cia.gov/library/center-for-the-study-of-intelligence/csi-publications/books-and-monographs/sherman-kent-and-the-board-of-national-estimates-collected-essays/6words.html>).

We have read and referenced now the analogous reports from the US. We have acknowledged the fact that the US Senate Committee's first conclusion was indeed that the facts were stretched and also cited their second conclusion about the removal of uncertainties:

"In the US, it was the Intelligence Community's October 2002 National Intelligence Estimate (NIE) called "Iraq's Continuing Programs for Weapons of Mass Destruction" that was the analogous document pre-invasion. A US Senate Select Committee investigation was even more critical of it than the Butler Review in the UK, but its second conclusion was similar:

“Conclusion 2. The Intelligence Community did not accurately or adequately explain to policymakers the uncertainties behind the judgments in the October 2002 National Intelligence Estimate.”

The removal of considerable expressions of uncertainty from both documents had a dramatic effect on the opinions of the public and governments, and in the UK at least the removal of the uncertainties was considered key to paving the way to war.”

Although we are big fans of ‘Words of Estimative Probability’, here doesn’t seem quite the right place to cite it as we were looking for a specific example of the consequences of poor uncertainty communication.

Could the authors add some surveys in the US or elsewhere, given that they want (and deserve) a broader audience than the UK (submission to the Royal Society notwithstanding).

We have added the Pew Research Center and US National Science Foundation surveys from 2015 & 2018.

3/8 The authors might consider the relevance of my own piece with such evidence in a somewhat analogous US publication: Fischhoff, B. (2012, Summer). Communicating uncertainty: Fulfilling the duty to inform. Issues in Science and Technology, 28(4), 63-70

We thank Prof Fischhoff for drawing our attention to this article, which was also pointed out by another reviewer. We have included it in our introduction: “Anecdotal experience suggests a tacit assumption among many scientists and policy makers that communicating uncertainty might have negative consequences, such as signalling incompetence, encouraging critics, and decreasing trust (for example, see [REF]).”

3/22-28 See comments on Abstract.

3/34ff. See comments on Abstract (2/12)

3/48ff Without saying what predecessors did and why the authors deem it a failure, this paragraph doesn’t advance the paper very much. I would cut it, adding the references to a string somewhere.

We have rewritten the introductory paragraphs, now using them more usefully: to define what we mean by the terms we use throughout.

Page 4

4/7ff. To my taste, this paragraph wanders, too. I don’t think that anything would be lost by cutting it and moving straight to the authors’ proposal.

We felt that this paragraph had some merit in stating the particular timeliness of the work and the opposing hypotheses about the role of uncertainty communication in trust, but we have shortened it.

Using Lasswell’s classing task analysis seems like an excellent move. However, I think that it needs some reference(s) to more recent scholarship, on how well it has stood the test of time and what others have done with it.

We now explicitly acknowledge the age of the Lasswell model (in describing it as ‘venerable’), and point out that we have specifically added ‘relevant context of the message’ to the original model (a feature often added to later models of communication). We didn’t feel there was any later scholarship that specifically needed referencing on the topic of Lasswell’s model and that doing so would add to the weight of the introductions without adding useful knowledge.

p. 5

5/24 Would a close reading of the authors' scheme show that they have not considered decision making?

Thank you for pointing this semantic issue out. We considered decision-making as part of 'behaviour', and now make this explicitly clear in our summary of the factors in this introduction and also throughout Section 5.

page 7

7/8ff Here's one of those places where I thought that the paper's agreeable chattiness slows things down

Section 1. I don't think that this section served much purpose beyond having the authors say something about this topic, without much by way of referencing the vast (I believe) literature on source credibility.

We take this point on board and have reduced the section to a short paragraph. We hope that this is one of several major shortenings that brings the manuscript closer to Prof Fischhoff's vision of it as a slimmer and more to-the-point piece.

page 8ff

Section 2. This section had the makings of a valuable standalone section on characterizing uncertainty, which approached a form that readers might apply to their own domains. It might be coupled with Section 3.1, if the two category schemes were more tightly integrated. It would be stronger by more explicit contrasts with schemes having similar aspirations. I was curious about the distinctions that the authors saw with Ravetz and Funtowicz, who have similar general aspirations. I was looking for GRADE/CONSORT here (and pleased to find it later). I suggest moving the references to behavioral research, to a later section where they could be treated more systematically.

We have addressed these valuable points in slightly different ways. As mentioned before, we have added a section in the introduction explicitly dealing with some previous models of uncertainty, and have discussed Funtowicz and Ravetz' NUSAP scheme there.

We have moved our discussion of CONSORT and other schemes developed to help characterise indirect uncertainty up into section 2.3 where Prof Fischhoff quite rightly spotted that they should have been.

We have also moved the references to behavioural research as suggested.

Finally, we have – partly through our simplification of what we previous called 'three types of uncertainty' into 'two levels', and partly through better wording (discussed later) – attempted to make Sections 2 and 3 more synergistic.

p. 11

Box 3. Is there a place here for a reference to non-Bayesian treatments of evidence?

We don't think there is a place here for it, our apologies.

p. 12

12/6 I don't think that the implicit references to risk comparisons here did that topic justice, and would leave it to an empirical section.

12/16 This paragraph felt out of place without more systematic treatment of how uncertainty is to be used.

Thank you - we have moved these two paragraphs into Section 5 where they feel more in context.

Section 3. It seemed like there should be a stronger connection between this section and the previous one. Given how much thought the authors have given to these issues, I was interested in knowing what constraints they saw what was being communicated (Section 2) imposed on how it was communicated (Section 3). Some of the two seemed to be intertwined (e.g., the top of p.13). I have the feeling that both sections would be strengthened by moving some of the commentary in Section 3 to Section 2. Alternatively, Section 2 might be folded into Section 3, eliminating the seeming (to me at least) disconnect between the two.

We feel that the overall structure of the paper, sticking with our framework as described in Fig 1, is worth keeping. However, we have moved some material from Section 3 up to Section 2 (specifically the bits suggested below), and have also made more explicit the connection between the object, level and expression of uncertainty.

p. 13

Some of the material here seemed to belong to Section 2.

We agree, and have moved it.

p. 14

I loved this table, with so many good and bad examples brought together. However, as suggested in my general comments, absorbing both simultaneously is difficult, for a reader wondering "what should I do?"

We have clarified the table's purpose in the text:

"In order to explore whether each in this list of 9 expressions of absolute, direct uncertainty could be applied to all three objects of uncertainty in our framework: categorical or binary *facts*, continuous variables (*numbers*) and *models* we set out to find real examples of each in use. The result of our search is shown in Table 1. We were not able to find examples for each cell in the table, illustrating where some usages are rare at best. However, our intention was both to test the comprehensiveness of our framework and to illustrate it to help others identify how it can be applied. We fully admit that some of the entries are ambiguous: for example, as we shall see in Box 4, the IARC's claim of a 'probable carcinogen' is more an indirect summary of the quality of evidence for carcinogenicity, rather than a direct expression of probability and so may not belong in the table at all."

We have added boxes of take-home bullet points in the conclusion for the reader wondering 'what should I do?'

p. 19

As mentioned, although I agree that GRADE/CONSORT is a powerful communication tool (and have advocated it as such), I think that its value arises from its practical approach to characterizing uncertainty — hence belonging to Section 2, then reaping its benefits in Section 3. Perhaps I am missing something fundamental here, however, it seems to me that that the two sections could be combined — at the price of abandoning the Lasswell scheme.

This was a mistake on our behalf which led to confusion. We have now made things clearer by separating the concept of a checklist (eg CONSORT) which is used to **characterize** the uncertainty from the **communication format** for that characterization (eg GRADE's categorizations, or the EEF's padlock ratings). At this point in the paper we should only have been talking about the

communication format of the uncertainty. We have moved the section describing the means of assessing indirect uncertainty up into Section 2 as suggested.

We think that it is very common to fail to separate the methods of characterising uncertainty from the methods of communicating it (as indeed we did in our previous draft!) and that actually they need to be considered and evaluated separately – hence the value of the separation of sections 2 & 3. Hopefully this is now much more explicit and clear in our new version of the manuscript.

bottom. This seems like a good example of a bad example, which muddles the normative exposition here (of what to do right).

We have tried to be purely descriptive rather than proscriptive in our text, not giving examples only of ‘what to do right’. However, to make our intentions clear, we have added a sentence before the Box explicitly stating that “We cite these examples as a useful warning to practitioners considering constructing a ‘simplified’ method of communicating the uncertainties in their field.”

p. 22

Section 3.2 wasn’t really connected to what preceded it and seems part of the story of Section 5. The representations presented here, like those in Section 5, are compromises designed for practical purposes, rather than having essential, epistemological properties, like the core topics of Sections 2 and 3.1.

We have tried to make this more clearly ‘the next part of the story’ (as defined by the Lasswell model in Fig 1) and also clarified that format is not simply a design choice – they contain different types of information, and related these directly to our list in Section 3.1.

p. 24

Section 4, like Section 1, didn’t seem to serve any purpose beyond having something to say about this element in the Lasswell scheme. Its few references seemed somewhat haphazardly selected. It has less anti-public tone than most brief summaries. However, it doesn’t do justice to the empirical literature or the ways in which the authors’ proposal would lead to reinterpreting its contents. I would cut it.

We understand the (fair) point that Prof Fischhoff makes here, but we would like to keep all elements of the Lasswell scheme in the manuscript for overall consistency. We strongly prefer to avoid a situation in which we selectively report on only some sections of the Lasswell scheme. However, to accommodate the reviewer, we have shortened this section by at least 50% but we felt that it was important to mention some of the well-known psychological phenomena such as confirmation bias and motivated reasoning that many readers may be unfamiliar with and so have kept a brief 2-paragraph section here, explicitly described as a warning that this is an important factor consider. In other words, this section serves a reminder that “knowing your audience” matters and we now end this section by eluding to the important relationship between the audience and the communicator, a key factor which we do cover in detail in section 5.3.

p. 25-26

This section got off to a slow start.

We have shortened this section considerably.

p. 27-29

Section 5.1 builds nicely on the scheme introduced in Section 3.1. The studies that I knew were very nicely summarized. However, it was often unclear how authoritative the authors intended their summaries to be. It seemed like they sometimes picked illustrative studies, suggesting what research

might find, and other times offered their interpretation of the results of more comprehensive reviews. As a result, the section felt uneven and not as effective an explication of the scheme in Section 3.1 as it might have been. For example, where fundamental categories are combined, is that because the authors were trying to cram a lot into an already long article or because there was no prior research? If the latter, then I think that they would advance the cause by explicating the research program.

Thank you. We have now been explicit in the introduction to this section that we have not completed an exhaustive systematic review of this widely scattered field, but have cited all studies that we found that were of relevance. In terms of the “uneven” feeling, we now also make clear upfront that we have gone into detail with the description of those studies which we deemed particularly insightful for illustrative purposes (as we don’t think that balancing the length of every study will do the review justice). As a result, inevitably, some summaries are more detailed than others, but we hope that the general explanation and rationale behind our approach makes this much clearer now and we thank the reviewer for pointing out the need to make this explicit.

In addition, we have now also explained that we have combined categories only when conclusions from studies appeared to be equally applicable.

pp. 29-31

Section 5.2 nicely applies the scheme from 3.1 to an emerging topic (which I know less well). Here, it feels though the authors have cited every study that they could find. If so, then does the topic merit the same length as Section 5.1?

We have tried to cite every study we could find throughout the review, but Section 5.2 is about 1/3rd shorter than 5.1, mainly because it is an emerging topic and so less is known overall. However, we have aimed to offer a similar level of detail in coverage about what is known, especially because it is an important emerging topic. We do feel that the issue of affect/emotion deserves some attention, especially in response to frequent non-empirical claims about how uncertainty is typically predicted to elicit negative emotions.

pp. 31-33

An important topic. Some of the results seemed hard to interpret absent the context provided by the issues in the preceding sections.. Here, too, I wondered why some topics in the authors’ scheme were not addressed.

Thank you. We have now made it explicit that we are only able to write about the three forms of expression which we found sufficiently studied in the literature.

pp. 33-34

Section 5.4 was less satisfying than its predecessors, which looked at one aspect of individuals’ responses. Dealing with decisions, which involve multiple aspects, would require more systematic treatment. For example, what standards did [149] and [150] apply when determining rationality? The examples here seem to be mostly hypothetical decisions, in any case. There are also so few that the authors’ scheme is abandoned. I suggest cutting the section and folding anything relevant into the preceding sections.

Yes, thank you for this important comment. We would prefer not to cut this section (as per our other note to maintain overall consistency of Lasswell’s framework). We have now been clearer about using the term decision-making (not all decisions involve or lead to a behaviour of course). In addition, we have substantially expanded our explanation of the studies mentioned for conceptual

clarity. Upon re-reading, we have found the term “rationality” not really meaningful in this context, and so simply abandoned this terminology to avoid confusion. Instead, we now plainly explain what these studies were about with better context. We do have to abandon our scheme given the overall lack of research on real behaviour (which is a problem in many literatures) but we don’t view this as a weakness, as it provides an opportunity for future research to identify and fill this gap.

p. 35ff

I liked the case studies, which were very nicely written. I’ve seen parts of some of them before. Manski has a recent piece in PNAS interpreting his work from a communication perspective: <https://www.pnas.org/content/early/2018/11/21/1722389115>.

We have added this new reference (thank you!) and integrated it into our discussion.

pp. 43ff

I thought that the statement of the scheme was useful and might have been well-placed at the beginning of the article, replacing the one on p. 6 which, to my mind, promised detail on some topics that the article didn’t deliver (e.g., Sections 1 and 4). The rest of the section, though, seemed to say mostly that the issues were complicated and the situation murky. I didn’t feel like I got very much from it. My suggestion would be to frame the empirical accounts in Section 5 more explicitly as illustrative. Use them to make the case that empirical research is essential — supported by the ill-conceived schemes that appear throughout the article.

We have rewritten the introduction to the manuscript and changed Fig 1 to highlight the sections where the article now focusses.

We have also added two boxes to the discussion giving a more positive summary for both researchers and practitioners to give take-home points (hopefully useful ‘at a glance’), and used the ill-conceived schemes in Box 4 as an explicit warning on the dangers of designing communication schemes without empirical testing.

We hope that these changes – along with the general tightening and the specific alterations suggested by the reviewers – have moved the paper closer to that which Prof Fischhoff envisages.

Reviewer: 4 (Andrew K. Przybylski)

In this paper the author(s) advance a meta theoretical framework for communicating scientific uncertainty. This is a topic that I as a scientist find extremely challenging (in practice) and have largely not felt satisfied with many of the proposed ways of addressing the topic as they're often siloed within specific disciplines which I find it difficult to relate. With that in mind I was largely positive on this paper as I learned a lot as a reader. Some comments highlight areas where things might have been missing.

1. There are areas where I could see competing interests feeding into the framework here (e.g. at he who and what stages). Could the authors consider cases where this has been handled, to some extent, e.g. pharmaceuticals, and areas where this is less clear, e.g. social data science?

We have actually shortened considerably the 'who' section of the manuscript (section 1) following the suggestions of Prof Fischhoff (and overall suggestions from all reviewers that the paper could be improved by being tighter). However we have kept in the relevant reference to different intentions and added the following example of competing interests:

“Communicators may intend to have very different effects on their audiences, from strategically-deployed uncertainty (also known as “merchants of doubt”) to transparent informativeness. For example, in the Butler report on the document “Iraq’s Weapons of Mass Destruction: The Assessment of the British Government” discussed in Box 1 it was noted that the differences in uncertainty communication were in part because:

“The Government wanted a document on which is could draw in its *advocacy* of its policy. The JIC sought to offer a dispassionate *assessment* of intelligence and other material...”

We have also added a section at the end of the Case Studies discussing in a little more detail how competing interests have (potentially) influenced the choice of communication of uncertainty and the resulting psychological impact:

“Both these case studies consider potential contested topics, where communicators may wish to portray uncertainty in different ways in order to produce different psychological effects on the reader. Whereas the official sources of information in both cases make attempts to communicate uncertainty ‘neutrally’, secondary users of their information (such as media organisations) often don’t. In the case of official economic statistics, for example, media coverage overwhelmingly avoids mention of epistemic uncertainty. This leads to inevitable concern and discussion when revisions to statistics occur, which itself reveals that readers thought the estimates more certain than they actually were – and their comprehension, emotions and decision-making were all likely to have been altered by the change in uncertainty communication caused by a revision. In the case of climate change, there has been much discussion over whether over-emphasis or under-emphasis of uncertainty in communications has created changes in the comprehension, emotions or behaviour of different audiences – although the lack of empirical evidence specifically on this topic makes those opinions pure hypothetical, if entirely plausible.”

We decided to stick to the existing examples we had in the manuscript (intelligence, climate change).

2. I think the repetitional costs of miscommunication around uncertainty are something that the paper could expand on. This is a prominent aspect of the paper at the start and can be looped back in via the argumentation in an additional paragraph in the conclusion.

conclusion.

Thank you - we have now added a specific warning as to the reputational risks in the Discussion (“The reputational and real risk to life of miscommunication of uncertainty is great, as the example in Box 1 highlights.”). We feel that this back-reference to a strong example should underline this important point.

3. I think a synthetic example could support the narrative and draw the reader in more fully. The examples in the discussion (p.44) could be expanded to this end. In other words, consider presenting how to different scientific programs on topic x would play out differently in the future given they take different paths through the framework the authors outline. I understand that this pivots the system from descriptive to proscriptive but this could be very useful.

Rather than create a synthetic example we have added a paragraph at the end of the case studies (cited above) to use these more effectively to illustrate how, particularly in these two contested topics, different parties could use different paths through the communication framework to achieve different end effects. We also add two boxes to the Discussion which are proscriptive, but hopefully more useful to readers.

Appendix D

9th April 2019,

REF: Manuscript RSOS-181870.R1

Dear Editors,

Thank you very much indeed for your letter of 1st April regarding this manuscript ('Communicating uncertainty about facts, numbers, and science'). We are delighted at your acceptance for publication (pending minor revisions) and also wish to thank the reviewers once again for their deeply helpful and perceptive comments.

We have revised the manuscript according to their comments, as listed below.

Baruch Fischhoff

Minor Comment 1: I think that the risks with indirect uncertainty practices (p. 20ff) are greater than the text implies, in cases where audience members are not privy to professional conventions. The authors might consider the conventions that John Cohen introduced as part of his campaign to get psychologists to perform power analyses: small, medium, and large effect sizes. Those terms and their statistical equivalents reflected his intuition of what kind of results impressed psychologists and, I am guessing, what would make them feel good enough to adopt his methods. What do non-psychologists think if psychologists say, or act like, they have large effects — not knowing that it is large for psychologists. Similarly, do people interpret “high quality” as “high quality, given the challenges of clinical trials”?

We entirely agree that the context-specificity of words is important and have made an adjustment on p20 such that it now reads:

“These broad categorical ratings are used when the impact of poorer quality evidence is difficult to quantify. One issue with such broad categorical ratings or verbal descriptions (“high quality”) is that their meaning is in part dependent on the context of their use: at what threshold evidence is classified as high quality or low quality might depend on the research field or topic. The audience, especially if they are non-experts, might not be aware of this. In addition, research has shown that there is considerable variation in people’s interpretation of verbal probability and uncertainty words such as “likely” [69–72]. There might be a similar variability in what people interpret “high quality” or “low quality” to mean, which might make such broad categorical ratings or verbal descriptions less effective.”

Minor Comment 2: The authors repeatedly say that there is very little empirical research on these topics, but then cite a lot of it (witness the references to empirical papers). Indeed, in the spirit of their paper, it communicated to me that, on some topics, there is enough evidence for it to be inconclusive. They might revisit the wording on this topic,

We agree that overall there are many papers that in some way address the effects of communicating uncertainty and sometimes there is enough work to say it has been done but is inconclusive, with the main problems being that it is often low quality (small sample sizes, not representative populations) and not done systematically (the type of uncertainty is not controlled, or sometimes even specified).

We have searched the manuscript for places to make this clear. In the introduction we feel it is adequately described (“the existing empirical research on the effects of communicating epistemic uncertainty is limited.”) In Section 4 we have added the word ‘systematic’ to make it clear what the limitations of existing work are (“Unfortunately, there is very little systematic empirical work studying these effects on the communication of epistemic uncertainty.”)

In Section 5 we feel we generally describe the literature accurately (5.2.3: *“In short, the limited research described above [3 studies] reports inconsistent results: it appears that communicating uncertainty can have*

an impact on people's emotions, but that the nature of the impact might be dependent on how emotions are defined and measured as well as how uncertainty interacts with other characteristics of the communication." 5.3: "Research into how the communication of scientific uncertainty impacts trust and credibility is very sparse, and we found examples from only three of forms of expression of uncertainty").

In 5.3.2 we have removed the adjective 'limited' to leave the summary sentence as "In sum, until more research is conducted, it is difficult to make firm conclusions about these mixed findings across domains about the way and extent to which communicating uncertainty affects the perceived credibility of and trust in both the message and the communicator."

In the concluding Section 5.5 we have replaced the word 'limited' with 'scattered' so that the opening sentence reads: "*Although the scattered evidence available suggests that communicating direct epistemic uncertainty does affect people's cognition, emotion, trust, and behaviour and decision-making, little has been done within a systematic framework — identifying the aspects of the communication that are being manipulated and therefore delineating their precise effects.*" The next paragraph opens with a sentence we believe accurate ("*Considering the literature on the psychological effects of different expressions of uncertainty, however, suggests several interesting preliminary findings.*"), and the next we believe is an appropriate use of the term limited ("*The limited research that has investigated the effects of epistemic uncertainty communication on emotions has found mixed results...*")

In the concluding discussion we have removed the word 'limited' ("Our review of the empirical evidence of the effects of epistemic uncertainty communication showed that different uncertainties, and different expressions of those uncertainties, can have varied psychological effects on their audiences.")

Major Comment: I was disappointed not to see a strong concluding statement calling for the empirical evaluation of communications. To my mind, its absence substantially undermines the value of the entire article. The authors have demonstrated the dangers of the amateur-hour approach that is the norm in scientific communication. They have established that the empirical results are mixed on those issues where there have been studies. Their framework demonstrates the complexity of real-world communication. Their summary suggestions could be very helpful in designing communications that are worth testing. However, they only provide better guesses at what might work. If readers believe that following these suggestions will guarantee success, then they are being set up for failure. We structured the FDA guide so that evaluation is central to the effort and most chapters end with suggestions for how to evaluate for no money at all, a little money, or money commensurate with the health, economic, and political stakes riding on the communications — framed so that even amateurs could do it. (<http://www.fda.gov/AboutFDA/ReportsManualsForms/Reports/ucm268078.htm>)

The authors may disagree on this question. If so, I think that it would be appropriate to end the paper with a clear statement of that position, rather than ignoring it.

We entirely agree, are fans of the FDA guide, and have strengthened this aspect of our conclusions. We have added a reference to the guide and emboldened the final sentence of our conclusions box for communicators, which now states:

*"if you can **test the effect of your communication with your audience** at all, then do (we recommend the FDA's excellent guide when attempting this[189], and please share the results – see Box 5b)"*

We have also added two sentences (underlined below) to our penultimate paragraph:

"Because of its wide-reaching effects, uncertainty communication should be an important issue for policy makers, experts, and scientists across many fields. Many of those fields carry the scars of attempts to avoid communicating uncertainty, or of poorly considered communications of uncertainty. These emphasise the need for a more considered approach to the topic, based on empirical evaluations done within an accepted framework. At present, however, this appears to be a science in its infancy. We can draw very limited conclusions from the current empirical work about the effects of communicating epistemic uncertainty and any underlying mechanisms. There is therefore a strong need for research specifically focused on communicating epistemic uncertainty and its impact on cognition, affect, trust, behaviour and decision-making. Early work needs confirming with large representative samples, and with observed or reported rather than hypothetical decision-making, as we currently have very little idea about generalisability of findings."

References:

The first of the references below is an empirical attempt to apply both Bayesian and non-Bayesian methods, recognizing that the authors do not want to get into the latter.

The second and third references are, I believe, empirical studies of some of the issues that the authors raise and might be worthy of citation.

Curley SP. 2007. The application of Dempster-Shafer theory demonstrated with justification provided by legal evidence. *Judgm. Dec. Making* 2(5):25-276.

This is an interesting paper, but we think might distract from our main arguments about Levels of uncertainty – Curley’s ‘evidential weights’ are still used to produce direct statements.

Üklümen G, Fox CR, Malle BF. 2016. Two dimensions of subjective uncertainty: Clues from natural language. *J. Exp. Psychol.:Gen.* 145(10):1280-1297.

Walters DJ, Ferbach PM, Fox CR, Sloman SA. 2017. Known unknowns: A critical determinant of confidence and calibration. *Manag. Sci.* 63(12):4298-4307.

We greatly thank Prof Fischhoff for suggesting these important references. In particular, Üklümen, Fox, and Malle (2016) led us to another paper by Üklümen & Fox (2011), which jointly make the important case that people psychologically distinguish between epistemic uncertainty about facts and aleatoric uncertainty about possible future outcomes. Moreover, they describe important differences in cognitive attribution and expressions in natural language that follows from this intuitive distinction. We have now incorporated these references in detail on p.26, which greatly strengthens our point that this distinction is of high psychological relevance, and also, that the literature up until now, has largely failed to explicitly recognize this distinction (a point which Üklümen et al. also buttress). We feel these references were extremely useful additions to the manuscript. Thank you.

Charles Manski

1. Pp 3-5: I appreciate the authors’ re-working of the introduction and their inclusion of more explicit discussion of the definition of uncertainty. However, I still have a problem with the term “future” uncertainty and its conceptual distinction from epistemic uncertainty. I maintain that as described by the authors, these are not mutually exclusive categories, and the term “future” just confuses matters. To me, this distinction is not useful and in fact misleading. Epistemic uncertainty can encompass issues pertaining to the future as well as the present and past. The issues comes down to definitions, or course, but I believe that temporality is not the factor that determines whether uncertainty is “epistemic” or not, and believe I’m not alone in this view.

Compounding the confusion, in my mind, is that the authors simultaneously introduce a dichotomy between unknowability (can’t know) and unknownness (don’t know), and place it alongside the distinction between future and epistemic uncertainty. This juxtaposition implies that the defining feature of “future” uncertainty is unknowability, while the defining feature of “epistemic” uncertainty is mere unknownness. This distinction, too, is problematic. Even if an aspect of reality is in principle knowable, one can still be uncertain if it is unknown for the time being—and this uncertainty is clearly epistemic (pertaining to one’s knowledge). Furthermore, the author’s conflation of all these issues also implies that uncertainties about the present and past are knowable—and this is clearly not true. There are many things about the present and past that we not only don’t know, but can’t know, for many reasons.

I think what the authors really want to do is to distinguish between uncertainty that arises from or pertains to limits to knowledge (epistemic) vs. the fundamental indeterminacy or randomness of the world—this 2nd uncertainty is captured by the term aleatory/aleatoric, which the authors used in the previous version. I would much prefer they use that term instead of the word “future”: it is much less ambiguous and confusing, and more logically coherent and consistent with established terminology in the literature.

We agree, and have gone back to the basic distinction between aleatory and epistemic, pointing out that these are often, but not exclusively, related to future / past-present, and can’t/don’t know. So we explicitly say that our contrast is with fundamental indeterminacy or randomness, and that future generally contains

epistemic components. In a sense we are focussing on ‘pure’ epistemic uncertainty, and hence in later sections make a contrast with predictions of the future.

Our paragraphs now read (p. 3, introduction):

“In the scientific context, a large literature has focused on what is often termed ‘aleatory uncertainty’ due to the fundamental indeterminacy or randomness in the world, often couched in terms of luck or chance. This generally relates to future events, which we can’t know for certain. This form of uncertainty is an essential part of the assessment, communication and management of both quantifiable and unquantifiable future risks, and prominent examples include uncertain economic forecasts, climate change models, and actuarial survival curves.

In contrast, our focus in this paper is uncertainties about facts, numbers and science due to limited knowledge or ignorance – so-called ‘epistemic’ uncertainty. Epistemic uncertainty generally, but not always, concerns past or present phenomena that we currently don’t know but could, at least in theory, know or establish.¹ Such epistemic uncertainty is an integral part of every stage of the scientific process: from the assumptions we have, the observations we note, to the extrapolations and the generalisations that we make. This means that all knowledge on which decisions and policies are based — from medical evidence to government statistics — is shrouded with epistemic uncertainty of different types and degrees.

Risk assessment and communication about possible future events are well-established academic and professional disciplines. Apart from the pure aleatory uncertainty of, say, roulette, the assessment of future risks generally also contains a strong element of epistemic uncertainty, in that further knowledge would revise our predictions: see the later example of climate change. However there has been comparatively little study of communicating ‘pure’ epistemic uncertainty, even though failure to do so clearly can seriously compromise decisions – see Box 1.”

Footnote:

¹ *We may, for example, have epistemic uncertainty about future events that have no randomness attached to them but that we currently don’t know (for example, presents that we might receive on our birthday that have already been bought: there is no aleatory uncertainty, only uncertainty caused by our lack of information, which will updated when our birthday arrives). In this paper we do not consider concepts that are not even theoretically knowable, such as non-identifiable parameters in statistical models, knowledge about counterfactual events, or the existence of God. We refer the reader to Manski [27] for a discussion of “nonrefutable” and “refutable” (or testable) assumptions in econometrics*

2. Pp 11-12: I appreciate the authors’ attempt to rework the section previously entitled “type of uncertainty”. They have now changed the focus to the idea of “levels of uncertainty” with 2 main ones: direct and indirect. However, I believe this change has created new conceptual difficulties that again are raised by the choice of specific words that have specific connotations. I believe what they are now referring to is a distinction that decision theorists since Knight (whom the authors cite) have described variously as 1st-order vs 2nd-order uncertainty, known vs unknown probabilities, risk vs. uncertainty, probability vs. ambiguity (Ellsberg). The essence of this distinction is a metacognitive reflexiveness: a thinking about thinking—in this case, an uncertainty about one’s uncertainty. The distinction captures a secondary mental state of uncertainty focused back upon a prior uncertainty about some issue—in the authors’ words, “Caveats” about the “quality of the underlying knowledge” regarding “facts, numbers, and hypotheses.” The authors’ example is a case of direct uncertainty as concerning “the absolute probability of guilt,” which is compounded by indirect uncertainty concerning the “credibility to be given to an individual’s testimony concerning this item of evidence.” This distinction is conceptually equivalent to the existing, well-established distinction between 1st vs 2nd order uncertainty, or probability vs ambiguity; the underlying phenomenon boils down to higher-order uncertainty about uncertainty.

The question is which set of words is most useful to express a concept. To me, the words “direct” and “indirect” do not hit the mark with their connotations, and do not clarify but instead obscure the essence of the underlying phenomenon. To me, these words do not capture the concept of reflexiveness, and don’t even match up to the notion of “levels.” It is certainly the authors’ prerogative to apply a new terminology, but then I believe they should acknowledge the other more established terms, and make explicit how their terms differ or not.

I also recognize that some scholars do not believe in 2nd-order uncertainty; Morgan and others, for example,

have argued that the notion of uncertainty about uncertainty is incoherent and not useful—that everything is uncertainty, and it is not necessary to distinguish levels. I personally do not agree with this view in theory, and believe there is ample empirical evidence showing that people behave as if 2nd-order uncertainty did exist. However, if this is the theoretical rationale for the author’s choice of terms, then they should at least make it explicit and defend it.

This is such a good point and led to great discussions amongst the authors! We have rewritten 2.3 to introduce ideas of 1st and 2nd order, but then argue that this is not the most useful division when discussing the communication of uncertainty (otherwise everything that’s not a full probability distribution, ie. anything below the top row of the Table 1, would be lumped together as 2nd order). In all the contexts we have examined, the most natural division happens between direct and indirect expressions, and so we have adopted this structure.

Our paragraphs now read (p.11-12):

“A vital consideration in communication is what we have termed the level of uncertainty: whether the uncertainty is directly about the object, or a form of indirect ‘meta-uncertainty’ – how sure we are about the underlying evidence upon which our assessments are based. This differs from the common distinction made between situations where probabilities are, or are not, assumed known. In the context of uncertainty quantification, the former is known as 1st order uncertainty and the latter 2nd order uncertainty, often expressed as a probability distribution over 1st order probability distributions or alternative models. An alternative categorisation derives from Knight [29] and Keynes [30], who distinguish quantifiable risks from deeper (unquantifiable) uncertainties.

In contrast to both these approaches, we have observed that the major division in practical examples of communication comes between statements about uncertainty around the object of interest, which may or may not comprise precise 1st-order probabilities, and a ‘meta-level’ reflection on the adequacy of evidence upon which to make any judgement whatever. We therefore consider that, when communicating, it is most appropriate to distinguish two fundamental levels of uncertainty:

Direct uncertainty about the fact, number, or scientific hypothesis. This can be communicated either in absolute quantitative terms, say a probability distribution or confidence interval, or expressed relative to alternatives, such as likelihood ratios, or given an approximate quantitative form, verbal summary and so on.

Indirect uncertainty in terms of the quality of the underlying knowledge that forms a basis for any claims about the fact, number or hypothesis. This will generally be communicated as a list of caveats about the underlying sources of evidence, possibly amalgamated into a qualitative or ordered categorical scale.

This division neither matches the traditional split into 1st/2nd order nor quantified/unquantified uncertainty. Direct uncertainty may be assessed through modelling or through expert judgement, involving aspects of both 1st and 2nd order uncertainty, and may be quantified to a greater or lesser extent, whereas indirect uncertainty is a reflexive summary of our confidence in the models or the experts.² An example of a system designed to communicate indirect uncertainty is the GRADE system of summarising overall quality of evidence, which we discuss further in Section 3.

Box 3 demonstrates the difference between direct and indirect uncertainty within a legal context where we hope the distinction between the two levels is particularly clear.”

Footnote:

² *If we feel we ‘know’ the probabilities (pure 1st order uncertainty), for example when we have an unbiased coin, then in a sense there is no indirect uncertainty, since there are no caveats except for our assumptions. But as soon as assumptions are expressed, there is the possibility of someone else questioning them, and so they may have caveats. This reinforces the fact that epistemic uncertainty is always subjective and depends on the knowledge and judgements of the people assessing the uncertainty.*

3. P 15, table: a related fundamental disagreement I have with the authors is their treatment of representations of imprecision as expressions of 1st-order (what they call “direct”) rather than 2nd-order (“indirect”) uncertainty (“ambiguity” in Ellsberg’s terms). I believe, as do many decision theorists, that

imprecision signifies 2nd-order uncertainty (uncertainty about uncertainty). From a statistical modeling standpoint, imprecision does manifest uncertainty in probability estimates; confidence intervals are wider when either the evidence used in estimating probabilities or our modeling methods themselves are more inadequate or unreliable. And from a psychological standpoint, imprecision is perceived as signifying uncertainty: people respond differently to probabilities when imprecision is expressed. They perceive imprecise ranges as more uncertain than point estimates, and display ambiguity aversion in response to them. For all these reasons I believe that representations of imprecision in probability estimates such as probability distributions and ranges do not belong in the same conceptual category as point estimates of probability.

We see a natural progression, demonstrated in Table 1, from precise probabilities through increasing imprecision, all of which are direct statements about an object. Of course it is rather arbitrary where a line is drawn, but our experience is that this formulation resonates with people who need to express their uncertainty, and reflects what they have arrived at through a lot of work.

4. P 20: for this reason I also disagree with the assertion that “Methods for communicating the quality of the underlying evidence do not give quantitative information about absolute values or facts...” I think probability distributions, risk ranges, and confidence intervals do just that, by signifying with quantitative precision the imprecision of our estimates. That is not to say they are ideal or necessary or sufficient representations of 2nd-order uncertainty, only that they constitute quantitative expressions of it.

Same point – all the imprecise qualifiers in Table 1 (although technically 2nd order), reflect directly on the number. Whereas a qualitative statement about the quality of the underlying evidence does not lead directly to, say, an appropriate widening of the interval. However, we now add a comment on page 23 about how this might be a good thing, and reference to Turner et al 2009:

“Methods have been proposed for turning indirect into direct uncertainty. In the context of a meta-analysis of health-care interventions, Turner et al. [75] demonstrate that experts can take caveats about lower-quality studies and express their impact in terms of subjective probability distributions of potential biases. When these are added to the nominal confidence intervals, the intervals appropriately widen and the heterogeneity of the studies explained. These techniques have been tried in a variety of applications [76,77] and show promise, although they do require acceptance of quantified expert judgement.”

5. p 25: again, I think the use of the term “future” uncertainty is misleading, and also do not agree with the assertion in lines 12-13 that “ambiguity aversion is mostly about people’s aversion to using this information for making decisions about a future event.” To me, that statement is much too strong. It’s true that the classic experimental paradigm for demonstrating ambiguity aversion has consisted of gambling experiments using balls and urns, where the outcome was the willingness to bet on an unknown probability. However, this phenomenon generalizes to all kinds of judgments and decisions as well as not only behavioral but cognitive and emotional responses, and is thought to specifically reflect the effects of epistemic uncertainty (2nd-order uncertainty about one’s uncertainty) arising from limitations in the reliability, credibility, or adequacy of one’s information. This distinguishes it from risk aversion. A large body of empirical evidence has also shown that ambiguity aversion also consists not only of decisions about a future event, but current risk perceptions, judgments, preferences, and emotional responses.

This is a great point. We mostly agree and appreciate the importance of these nuances. Accordingly, we have changed our terminology from “future” uncertainty back to “aleatory” uncertainty throughout the manuscript. On the point about ambiguity aversion, the most cited study (Fox & Tversky, 1995) consists primarily of judgment scenarios that involve predictions about a future event (e.g. the chance of winning a lottery and how to allocate the winnings). We do feel that this lines up nicely with Craig Fox’s own psychological distinction between epistemic and aleatory uncertainty (where he clearly classifies possible future outcomes as aleatoric uncertainty). Nonetheless, we do agree and want to recognize that not all ambiguity aversion is about the future, some also concerns present judgments, and sometimes the two forms of uncertainty interact or one informs the other. We have toned down our language on p.26 (instead of “mostly” we now say “often, but not exclusively”) and acknowledge that these two forms of uncertainty are sometimes entangled and that as a result we must draw on the ambiguity aversion literature from time to time to inform our discussions about epistemic uncertainty.

6. P 43: I appreciate and agree with the authors' discussion at the bottom of the page on the difficulty of distinguishing between past, present, and future uncertainty, and think it would be good to discuss and highlight this important caveat much earlier in the paper.

Point made earlier, and forward reference to climate change example added.

We hope that you find these changes acceptable.

Thank you again for a constructive review process.

Yours faithfully,

Dr Anne Marthe van der Bles and the authorship team